# Discrepancy Minimization in Input-Sparsity Time

Yichuan Deng [1]  Xiaoyu Li [2]  Zhao Song [3]  Omri Weinstein [4]

## Abstract

A recent work by [Larsen, SODA 2023] introduced a faster combinatorial alternative to Bansal's SDP algorithm for finding a coloring $x \in \{-1, 1\}^n$ that approximately minimizes the discrepancy $\mathrm{disc}(A, x) := \|Ax\|_\infty$ of a *real-valued* $m \times n$ matrix $A$. Larsen's algorithm runs in $\widetilde{O}(mn^2)$ time compared to Bansal's $\widetilde{O}(mn^{4.5})$-time algorithm, with a slightly weaker logarithmic approximation ratio in terms of the *hereditary discrepancy* of $A$ [Bansal, FOCS 2010]. We present a combinatorial $\widetilde{O}(\mathrm{nnz}(A) + n^3)$-time algorithm with the same approximation guarantee as Larsen's, optimal for tall matrices where $m = \mathrm{poly}(n)$. Using a more intricate analysis and fast matrix multiplication, we further achieve a runtime of $\widetilde{O}(\mathrm{nnz}(A) + n^{2.53})$, breaking the cubic barrier for square matrices and surpassing the limitations of linear-programming approaches [Eldan and Singh, RS&A 2018]. Our algorithm relies on two key ideas: (i) a new sketching technique for finding a projection matrix with a short $\ell_2$-basis using implicit leverage-score sampling, and (ii) a data structure for efficiently implementing the iterative Edge-Walk *partial-coloring* algorithm [Lovett and Meka, SICOMP 2015], and using an alternative analysis to enable "lazy" batch updates with low-rank corrections. Our results nearly close the computational gap between real-valued and binary matrices, for which input-sparsity time coloring was recently obtained by [Jain, Sah and Sawhney, SODA 2023].

## 1. Introduction

Discrepancy theory is a fundamental subject in combinatorics and theoretical computer science, studying how to color elements of a finite set-system $S_1, \ldots, S_m \subseteq \{1, \ldots, n\}$ with two colors (e.g., red and blue) to minimize the maximum imbalance in color distribution across all sets. It finds diverse applications in fields such as computational geometry (Matousek, 1999b; De Berg, 2000), probabilistic algorithms (Spencer, 1985; Chazelle, 2000), machine learning (Vapnik & Chervonenkis, 1971; Talagrand, 1995; Karnin & Liberty, 2019; Bechavod et al., 2022; Han et al., 2025), differential privacy (Muthukrishnan & Nikolov, 2012; Nikolov et al., 2013), and optimization (Beck & Fiala, 1981; Bansal, 2010; 2012; Nikolov, 2015).

**Definition 1.1** (Discrepancy). *The discrepancy of a real matrix $A \in \mathbb{R}^{m \times n}$ with respect to a "coloring" vector $x \in \{\pm 1\}^n$ is defined as*

$$\mathrm{disc}(A, x) := \|Ax\|_\infty = \max_{j \in [m]} |(Ax)_j|.$$

*The discrepancy of a real matrix $A \in \mathbb{R}^{m \times n}$ is defined as*

$$\mathrm{disc}(A) := \min_{x \in \{\pm 1\}^n} \mathrm{disc}(A, x).$$

This is a natural generalization of the classic combinatorial notion of discrepancy of set systems, corresponding to *binary* matrices $A \in \{0, 1\}^{m \times n}$ where rows represent (the indicator vector of) $m$ sets over a ground set of $[n]$ elements, and the goal is to find a coloring $x \in \{\pm 1\}^n$ which is as "balanced" as possible simultaneously on *all* sets.

Most of the history of this problem was focused on the existential question of understanding the minimum possible discrepancy achievable for various classes of matrices. For general set systems of size $n$ (arbitrary $n \times n$ binary matrices $A$), the classic result of Spencer (1985) states that $\mathrm{disc}(A) \leq 6\sqrt{n}$, which is asymptotically better than random coloring ($\Theta(\sqrt{n \log n})$). More recent works have focused on restricted matrix families, such as *sparse* matrices (Banaszczyk, 1998; Beck & Fiala, 1981), showing that it is possible to achieve the $o(\sqrt{n})$ discrepancy in these restricted cases. For example, the minimum-discrepancy coloring of $k$-sparse binary matrices (sparse set systems) turns out to have discrepancy at most $O(\sqrt{k \log n})$ (Banaszczyk, 1998).

All of these results, however, do not provide polynomial time algorithms since they are non-constructive—they only

[1]The University of Washington [2]University of New South Wales [3]University of California, Berkeley [4]Columbia University. Correspondence to: Xiaoyu Li <7.xiaoyu.li@gmail.com>, Zhao Song <magic.linuxkde@gmail.com>.

*Proceedings of the 42$^{nd}$ International Conference on Machine Learning*, Vancouver, Canada. PMLR 267, 2025. Copyright 2025 by the author(s).

argue about the *existence* of low-discrepancy colorings, which prevents their use in algorithmic applications that rely on low-discrepancy (partial) coloring, such as *bin-packing* problems (Rothvoß, 2013; Hoberg & Rothvoss, 2017).[1] Indeed, the question of efficiently finding a low-discrepancy coloring, i.e., computing $\mathrm{disc}(A)$, was less understood until recently and is more nuanced: Charikar et al. (2011) showed that it is NP-hard to distinguish whether a matrix has $\mathrm{disc}(A) = 0$ or $\mathrm{disc}(A) = \Omega(\sqrt{n})$, suggesting that ($\omega(1)$) approximation is inevitable to achieve fast runtimes. The celebrated work of Bansal (2010) gave the first polynomial-time algorithm which achieves an *additive* $O(\sqrt{n})$-approximation to the optimal coloring of general $n \times n$ matrices, matching Spencer's non-constructive result. Bansal's algorithm has approximation guarantees in terms of the *hereditary discrepancy* (Lovász et al., 1986).

**Definition 1.2** (Hereditary discrepancy). *Given a matrix $A \in \mathbb{R}^{m \times n}$, its hereditary discrepancy is defined as*

$$\mathrm{herdisc}(A) := \max_{B \in \mathcal{A}} \mathrm{disc}(B),$$

*where $\mathcal{A}$ is the set of all matrices obtained from $A$ by deleting some columns from $A$.*

Bansal (2010) gave an SDP-based algorithm that finds a coloring $\widetilde{x}$ satisfying $\mathrm{disc}(A, \widetilde{x}) = O(\log(mn) \cdot \mathrm{herdisc}(A))$ for any real matrix $A$, in time $\widetilde{O}(mn^{4.5})$ assuming state-of-the-art SDP solvers (Jiang et al., 2020b; Huang et al., 2022b). In other words, if *all* submatrices of $A$ have low-discrepancy colorings, then it is in fact possible (yet still quite expensive) to find an almost-matching overall coloring.

Building on Bansal's work, Lovett & Meka (2015) designed a simpler algorithm for *set-systems* (binary matrices $A$), running in $\widetilde{O}((n + m)^3)$ time. The main new idea of their algorithm, which is also central to our work, was a subroutine for repeatedly finding a *partial-coloring* via random-walks (a.k.a EDGE-WALK), which in every iteration "rounds" a constant-fraction of the coordinates of a fractional-coloring to an integral one in $\{-1, 1\}$ (more on this in the next section). Followup works by Rothvoss (2017) and Eldan & Singh (2018) extended these ideas to the *real-valued* case, and developed faster convex optimization frameworks for obtaining low-discrepancy coloring, the latter requiring $O(\log n)$ *linear programs* instead of a semidefinite program (Bansal, 2010), assuming a *value oracle* to $\mathrm{herdisc}(A)$.[2] In this model, Eldan & Singh (2018)

---

[1] If we don't have a polynomial constructive algorithm, then we can't approximate solution for bin packing in polynomial. This is common in integer programming where we first relax it to linear programming and then round it back to an integer solution.

[2] Eldan & Singh (2018)'s LP requires an upper-bound estimate on $\mathrm{herdisc}(A)$ *for each* of the $O(\log n)$ sequential LPs, hence standard exponential-guessing seems too expensive: the number of possible "branches" is $> 2^{O(\log n)}$.

yields an $O^*(\max\{mn + n^3, (m + n)^w\})$ time approximate discrepancy algorithm via state-of-art (tall) LP solvers (Brand et al., 2020). This line of work, however, has a fundamental setback for achieving *input-sparsity* time, which is a major open problem for (high-accuracy) LP solvers (Bubeck et al., 2018). Sparse matrices are often the realistic case for discrepancy problems, and have been widely studied in this context as discussed earlier in the introduction. Another drawback of convex-optimization based algorithms is that they are far from being practical due to the complicated nature of fast LP solvers.

Interestingly, in the *binary* (set-system) case, these limitations have been very recently overcome in the breakthrough work of Jain et al. (2023), who gave an $\widetilde{O}(n + \mathrm{nnz}(A))$-time coloring algorithm for binary matrices $A \in \{-1, 1\}^{m \times n}$, with near optimal $(O(\sqrt{n \log(m/n + 2)}))$ discrepancy. While their approach, too, was based on convex optimization, their main observation was that an *approximate* LP solver, using *first-order* methods, in fact suffices for a logarithmic approximation. Unfortunately, the input-sparsity running time of the algorithm in Jain et al. (2023) does not extend to real-valued matrices, as their LP is based on a "heavy-light" decomposition of the rows of binary matrices based on their support size. More precisely, generalizing the argument of Jain et al. (2023) to matrices with entries in range, say, $[-R, R]$, guarantees that a uniformly random vector would only have discrepancy $\widetilde{O}(\mathrm{poly}(R) \cdot \sqrt{n})$ on the "heavy" rows, and this term would also govern the approximation ratio achieved by their algorithm.

By contrast, a concurrent result of Larsen (2023) gave a purely *combinatorial* (randomized) algorithm, which is not as fast, but handles general real matrices and makes a step toward *practical* coloring algorithms. Larsen's algorithm improves Bansal's SDP from $O(mn^{4.5})$ to $\widetilde{O}(mn^2 + n^3)$ time, at the price of a slightly weaker approximation guarantee: For any $A \in \mathbb{R}^{m \times n}$, Larsen (2023) finds a coloring $\widetilde{x} \in \{-1, +1\}^n$ such that $\mathrm{disc}(A, \widetilde{x}) = O(\mathrm{herdisc}(A) \cdot \log n \cdot \log^{1.5} m)$. Recent work by Jambulapati et al. (2024) introduced a new framework for efficient computing of the discrepancy, with which they successfully recovered Spencer's result in time $\widetilde{O}(\mathrm{nnz}(A) \cdot \log^5(n))$ for matrix $A$ with entries restricted within 1.

The recent exciting developments naturally raise the following question:

*Is it possible to achieve input-sparsity time for discrepancy minimization with general (real-valued) matrices?*

In fact, this was one of the main open questions raised in 2018 workshop on Discrepancy Theory and Integer Programming (Dadush, 2018).

*Table 1.* Progress on approximate discrepancy-minimization algorithms of real-valued $m \times n$ matrices ($m \geq n$). For simplicity, we ignore $n^{o(1)}$ and poly$(\log n)$ factors in the table. "(Eldan & Singh, 2018)*" refers to a (black-box) combination of our Theorem 1.5 with (Eldan & Singh, 2018) and using state-of-art LP solvers for square and tall matrices (Lee & Sidford, 2014; Brand et al., 2020; Jiang et al., 2021).

| References | Methods | Running Time |
|---|---|---|
| Bansal (2010) | SDP (Jiang et al., 2020c; Huang et al., 2022b) | $mn^{4.5}$ |
| Eldan & Singh (2018)* | (Step 1: Theorem 1.5) + (Step 2: LP (Jiang et al., 2021)) | $m^{\omega} + m^{2+1/18}$ |
| Eldan & Singh (2018)* | (Step 1: Theorem 1.5) + (Step 2: LP (Brand et al., 2020)) | $mn + n^3$ |
| Eldan & Singh (2018)* | (Step 1: Theorem 1.5) + (Step 2: LP (Lee & Sidford, 2014)) | $\mathrm{nnz}(A)\sqrt{n} + n^{2.5}$ |
| Larsen (2023) | Combinatorial | $mn^2 + n^3$ |
| Ours (Theorem 1.3) | Combinatorial | $\mathrm{nnz}(A) + n^3$ |
| Ours (Theorem 1.3) | (Step 1: Theorem 1.5) + (Step 2: Lemma E.12) | $\mathrm{nnz}(A) + n^{2.53}$ |

## 1.1. Our Results

We answer this question in the affirmative. To this end, we develop an algorithm that achieves a *near-optimal* runtime for tall matrices with $m = \mathrm{poly}(n)$ and a *subcubic* runtime for square matrices, while maintaining the same approximation guarantees as Larsen's algorithm, up to constant factors. We state our main result in the following theorem.

**Theorem 1.3** (Main result, informal version of Theorem D.1 and Theorem D.2)**.** *For any parameter $a \in [0, 1]$, there is a randomized algorithm that, given a real matrix $A \in \mathbb{R}^{m \times n}$, finds a coloring $x \in \{-1, +1\}^n$ such that*

$$\mathrm{disc}(A, x) = O(\mathrm{herdisc}(A) \cdot \log n \cdot \log^{1.5} m).$$

*Moreover, it runs in time*

$$\widetilde{O}(\mathrm{nnz}(A) + n^{\omega} + n^{2+a} + n^{1+\omega(1,1,a)-a}).$$

*Here, $\omega(a, b, c)$ denotes the time for multiplying an $n^a \times n^b$ matrix with an $n^b \times n^c$ matrix, and $\omega := \omega(1, 1, 1)$ denotes the exponent of fast matrix multiplication (FMM).*

**Remark 1.4.** *With the current value of $\omega \approx 2.371$, according to Table 1 in Alman et al. (2025), we choose the parameter $a \approx 0.53$ to balance the terms $n^{2+a}$ and $n^{1+\omega(1,1,a)-a}$. Consequently, the running time in Theorem 1.3 simplifies to $\widetilde{O}(\mathrm{nnz}(A) + n^{2.53})$. Without FMM, our algorithm runs in $\widetilde{O}(\mathrm{nnz}(A) + n^3)$ time and is purely combinatorial.*

Let $\alpha$ denote the dual exponent of matrix multiplication, i.e., $2 = \omega(1, 1, \alpha)$, and let $a \in [0, 1]$ be the tunable parameter of Theorem 1.3. Currently, $\alpha \approx 0.32$ (Williams et al., 2024). Note that the running time of our algorithm (when using FMM) has a tradeoff between the additive terms $n^{2+a}$ and $n^{1+\omega(1,1,a)-a}$, so it is never better than $n^{2.5}$, as it is never beneficial to set $a < 1/2$. This tradeoff also means that our runtime is barely sensitive to future improvements in the value of the dual exponent (even $\alpha \approx 1$ would improve the exponent of the additive term by merely 0.03). Curiously, a similar phenomenon occurs in recent FMM-based LP

solvers (Cohen et al., 2019; Jiang et al., 2021), dynamic attention (Brand et al., 2024) and weight pruning (Li et al., 2024) in large language models.

A central technical component of Theorem 1.3 is the following theorem, which allows us to quickly find a "hereditary projection" matrix (i.e., a subspace such that the projection of a constant fraction of the rows of $A$ to its orthogonal complement has small $\ell_2$-norm), and is of independent interest in randomized linear algebra. We state it as follows.

**Theorem 1.5** (Fast hereditary projection, informal version of Theorem C.2 and Theorem C.3)**.** *Let $A \in \mathbb{R}^{m \times n}$ with $m \geq n$, and let $d = n/4$. There is a randomized algorithm that outputs a $d \times n$ matrix $V$ such that with probability $1 - \delta$, the followings hold simultaneously:*

- *For all $l \in [d]$, we have $\|V_{l,*}\|_2 = 1$,*

- *It holds that*

$$\max_{j \in [m]} \|(A(I - V^{\top}V))_{j,*}\|_2 = O(\mathrm{herdisc}(A) \log(m/n)),$$

*where $V_{l,*}$ denotes the $l$-th row of $V$ for any $l \in [d]$.*

*Moreover, it runs in time $\widetilde{O}(\mathrm{nnz}(A) + n^{\omega})$, where the $\widetilde{O}$-notation hides a $\log(m/\delta)$ factor.*

Theorem 1.5 directly improves Theorem 3 in Larsen (2023) which runs in $\widetilde{O}(mn^2)$ time without using FMM, and in $O(mn^{\omega-1})$ time using FMM. In either case, Larsen's algorithm pays at least $mn^{\omega-1}$ time, even when $A$ is *sparse* (see more discussion in Section 2.1). Theorem 1.5 is in some sense best possible: Reading the input matrix $A$ requires $O(\mathrm{nnz}(A))$ time, and explicitly computing the projection matrix $P = V^{\top}V$ in Theorem 1.5 requires $n^{\omega}$ time. We note that, in the case of binary matrices, the *projection-free* algorithm of Jain et al. (2023) avoids this bottleneck (and hence the $n^{\omega}$ term), but for real-valued matrices, all known discrepancy algorithms involve projections (Bansal, 2010; Lovett & Meka, 2015; Larsen, 2023; Rothvoss, 2017).

## 1.2. Related Work

**Algorithmic discrepancy theory** Constructive discrepancy theory has become a pivotal area of research, focusing on efficiently finding low-discrepancy solutions. Bansal's seminal work introduced a semidefinite programming (SDP)-based algorithm that achieves additive $O(\sqrt{n})$ discrepancy (Bansal, 2010), matching the nonconstructive bounds but incurring a significant computational cost of $O(mn^{4.5})$. Subsequent contributions, such as Lovett & Meka (2015) and Eldan & Singh (2018), developed faster algorithms leveraging random walks and convex optimization techniques, respectively. Recent advances include Larsen's combinatorial approach (Larsen, 2023), which attains a runtime of $O(mn^2 + n^3)$, and the near-input-sparsity algorithms for binary matrices proposed by Jain et al. (2023). Furthermore, Jambulapati et al. (2024) introduced an efficient framework for computing discrepancy, recovering Spencer's result in $\widetilde{O}(\mathrm{nnz}(A) \cdot \log^5(n))$ time for matrices $A$ with entries bounded by 1. These innovations have substantially narrowed the computational gap between theoretical results and practical applications, advancing the field's capabilities within polynomial time. For further details, we refer readers to related works (Cohen, 2016b; Bansal et al., 2018; 2019; Dadush et al., 2018; Alweiss et al., 2021; Bansal et al., 2020; 2022; Pesenti & Vladu, 2023; Jambulapati et al., 2024). Most recently, Han et al. (2025) uses the discrepancy theory to approximate the attention computation in the streaming model.

**Sketching and leverage-score sampling** Sketching is a versatile technique employed across numerous fundamental problems, including linear programming (Jiang et al., 2021; Brand et al., 2020; Song & Yu, 2021; Liu et al., 2023), empirical risk minimization (Lee et al., 2019; Qin et al., 2023; Gu et al., 2025), and semidefinite programming (Jiang et al., 2020a; Huang et al., 2022a; Song et al., 2023b). It is particularly prominent in randomized linear algebra, where it has been applied to a wide range of tasks (Clarkson & Woodruff, 2013; Nelson & Nguyên, 2013; Razenshteyn et al., 2016; Boutsidis et al., 2016; Song et al., 2017; Xiao et al., 2018; Song et al., 2019b; Lee et al., 2019; Jiang et al., 2021; Song & Yu, 2021; Brand et al., 2021; Song et al., 2022a; Hu et al., 2022; Gu & Song, 2022). Sketching frequently serves as an effective tool for oblivious dimension reduction (Clarkson & Woodruff, 2013; Nelson & Nguyên, 2013). The use of sampling matrices to enhance computational efficiency is a well-established approach in numerical linear algebra (see Clarkson & Woodruff (2013); Razenshteyn et al. (2016); Boutsidis et al. (2016); Song et al. (2017); Cohen et al. (2019); Song et al. (2019b); Brand et al. (2020); Li et al. (2023); Gao et al. (2023); Deng et al. (2023)). In this paper, we employ leverage score sampling as a non-oblivious dimension reduction method, in line with previous works

such as Spielman & Srivastava (2011); Batson et al. (2012); Zhang (2022); Song et al. (2022b).

## 2. Technical Overview

In this section, we give the overview of the techniques used to prove the main results, and the formal proofs are in Appendix. In Section 2.1, we first give an overview and discuss the barriers of Larsen's algorithm. In Section 2.2, we introduce our techniques used to improve the Larsen's algorithm and overcome the barriers.

### 2.1. Overview and Barriers of Larsen's Algorithm

Larsen's algorithm (Larsen, 2023) is a clever re-implementation of the iterated *partial-coloring* subroutine of Lovett & Meka (2015): In each iteration, with constant probability, this subroutine "rounds" at least half of the coordinates of a fractional coloring $x \in \mathbb{R}^n$ to $\{-1, 1\}$. As in Lovett & Meka (2015), this is done by performing a random-walk in the *orthogonal complement* subspace $V_\perp$ spanned by a set of rows from $A$ (a.k.a "Edge-Walk" (Lovett & Meka, 2015)). The key idea in Larsen (2023) lies in a clever choice of $V$: Using a connection between the eigenvalues of $A^\top A$ and $\mathrm{herdisc}(A)$ (Larsen, 2017), Larsen shows that there is a subspace $V$ spanned by $\leq n/4$ rows of $A$, so that the projection of $A$ onto the complement $V_\perp$ has small $\ell_2$ norm, i.e., every row of $A(I - V^\top V)$ has norm less than $O(\mathrm{herdisc}(A) \log(m/n))$. Larsen shows that computing this projection operator (henceforth $B^\top B$) can be done in $O(mn^2)$ time combinatorially, or in $\mathcal{T}_{\mathrm{mat}}(m, n, n) = \widetilde{O}(mn^{\omega-1})$ time[3] using FMM.

The second part (PARTIALCOLORING) of Larsen's algorithm is to repeatedly apply the above subroutine (PROJECTTOSMALLROWS) to implement the Edge-Walk of (Lovett & Meka, 2015): Starting from a partial coloring $x \in \mathbb{R}^n$, the algorithm first generates a subspace $V_1$ by calling the afformentioned PROJECTTOSMALLROWS subroutine. Then it samples a fresh random Gaussian vector $\mathbf{g}_t$ in each iteration $t \in [N] = O(n)$, and projects it to obtain $g_t = (I - V_t^\top V_t)\mathbf{g}_t$. The algorithm then decides whether or not to update $V$; More precisely, it maintains a vector $u_t$, and gradually updates it to account for large entries of $x + u_{t+1}$ that have reached $\approx 1$ in absolute value (as in Lovett & Meka (2015)). When a coordinate $i \in [n]$ reaches this threshold at some iteration, the corresponding unit vector $e_i$ is added to $V_t$ and $V_t$ is updated to $V_{t+1}$. This update is necessary to ensure that in future iterations, no amount will be added to the $i$-th entry of $u_t$. At the end of the loop, the algorithm outputs a vector $x^{\mathrm{new}} = x + u_{N+1}$ with property that for each row $a_i$ of $A$, the difference between $\langle a_i, x \rangle$ and

---

[3]$\mathcal{T}_{\mathrm{mat}}(m, n, k)$ represents the time required to multiply an $m \times n$ matrix by an $n \times k$ matrix.

$\langle a_i, x^{\text{new}} \rangle$ is less than $O(\text{herdisc}(A) \cdot \log^{1.5}(m))$. Since $V_t$ is constantly changing thoughout iterations, each iteration requires an *online* Matrix-Vector multiplication $(I - V_t^\top V_t)g_t$. Hence, since there are $N = O(n)$ iterations, $O(mn^2 + n^3)$ time is required to implement this part, *even if fast-matrix multiplication is allowed* – Indeed, the Online Matrix-Vector conjecture (Henzinger et al., 2015) postulates that $mn^2$ is essentially best possible for such online problem. Beating this barrier for the PARTIALCOLORING subroutine therefore requires to somehow avoid this online problem, as we discuss in the next subsection.

Below we summarize the computational bottlenecks in Larsen's algorithm, and then explain the new ideas required to overcome them and achieve the claimed overall runtime of Theorem 1.3.

Implementing the first part of Larsen's algorithm (PROJECTTOSMALLROWS) incurs the following computational bottlenecks, which we overcome in Theorem 1.5:

- *Barrier 1.* Computing the projection matrix $B_t = A(I - V_t^\top V_t) \in \mathbb{R}^{m \times n}$ explicitly (exactly) already takes $\mathcal{T}_{\text{mat}}(m, n, n)$ time. [4]

- *Barrier 2.* Computing the $j$th-row's norm $\|e_j^\top B_t\|_2$ requires $\mathcal{T}_{\text{mat}}(m, n, n)$ time.

- *Barrier 3.* Computing $B_t^\top B_t \in \mathbb{R}^{n \times n}$ requires multiplying an $n \times m$ with a $m \times n$ matrix, which also takes $\mathcal{T}_{\text{mat}}(n, m, n)$ time.

Implementing the second part of Larsen's algorithm (PARTIALCOLORING) incurs the following two (main) computational bottlenecks:

- *Barrier 4.* The coloring algorithm requires computing $\eta = \max_{j \in [m]} \|e_j^\top A(I - V^\top V)\|_2$, which takes $\mathcal{T}_{\text{mat}}(m, n, n)$ time.

- *Barrier 5.* In each of the $N = O(n)$ iterations, the algorithm first chooses a Gaussian vector $g \sim \mathcal{N}(0, I_d)$ and projects it to the orthogonal span of $V_t$, i.e., $g_{V_t} = (I - V_t^\top V_t)g$. Next, it finds a rescaling factor $g$ by solving a single variable maximization problem.[5] Finally, the algorithm checks whether $|\langle a_j, v+g \rangle| \geq \tau$, $|\langle a_j, v \rangle| < \tau$ for each $j \in [m]$. The overall runtime is therefore $mn^2$.

As mentioned above, the last step is conceptually challenging, as it must be done *adaptively* ($V_t$ is being updated

throughout iterations), and cannot be batched via FMM (assuming the OMv Conjecture (Henzinger et al., 2015))[6]. We circumvent this step by slightly modifying the algorithm and analysis of Larsen, and using the fact that, while the subspace $V_t$ is changing throughout iterations, the projected random Gaussian vectors $g_t$ themselves are *independent* – We show this enables to *batch* the projections and then perform *low-rank corrections* as needed in the last step. We now turn to explain our technical approach and the main ideas for overcoming these computational bottlenecks.

## 2.2. Our Techniques

A natural approach to accelerate the first part of the above algorithm is to use linear sketching techniques (Clarkson & Woodruff, 2013) as they enable working with much smaller matrices in the aforementioned steps. However, sketching techniques naturally introduce (spectral) error to the algorithm, which is exacerbated in iterative algorithms. Indeed, a nontrivial challenge is showing that Larsen's algorithm can be made robust to noise, which requires to modify his analysis in several parts of the algorithm. The main technical obstacle (not present in vanilla "sketch-and-solve" problems such as linear-regression, low-rank, tensor, inverse problems (Clarkson & Woodruff, 2013; Nelson & Nguyên, 2013; Razenshteyn et al., 2016; Song et al., 2017; 2019a; Lee et al., 2019; Jiang et al., 2021)) is that we can never afford to *explicitly* store the projection matrix $B_t$, as even writing it would already require $O(mn)$ time. This constraint makes it more challenging to apply non-oblivious sketching tools, in particular *approximate leverage-score sampling* (LSS, (Spielman & Srivastava, 2011)), which are key to our algorithm.

Breaking the $n^3$ runtime of Larsen's partial-coloring algorithm (which is important for near-square matrices $m \approx n$) requires a different idea, as this bottleneck stems from the Online Matrix-Vector Conjecture (Henzinger et al., 2015). To circumvent this bottleneck, we modify the analysis and implementation of the Edge-Walk subroutine (dropping certain verification steps via concentration arguments), and then design a "guess-and-correct" data structure (inspired by "lookahead" algorithms (Brand et al., 2019)) that batches the prescribed gaussian projections, with low-rank amortizes corrections. We now turn to formalize these three main ideas.

### 2.2.1. ROBUST ANALYSIS OF LARSEN'S ALGORITHM

**Approximate norm-estimation suffices.** The original algorithm (Larsen, 2023) explicitly calculates the exact norm of each row of $B_t = A(I - V_t^\top V_t)$. Since the JL lemma

---

[4]We remark that, $\mathcal{T}_{\text{mat}}(m, n, n) = O(mn^2)$ without using FMM, and $\mathcal{T}_{\text{mat}}(m, n, n) = O(mn^{\omega-1})$ with using FMM

[5]The naive computation here would take $O(n^2)$ time. Later we show how to do it in $O(n)$ time.

[6]The study of OMv originates from (Henzinger et al., 2015), the (Larsen & Williams, 2017; Chakraborty et al., 2018) provide surprising upper bound for the problem.

guarantees $\|e_j^\top \widehat{B}_t\|_2 \in (1 \pm \epsilon_0)\|e_j^\top B_t\|_2, \forall j \in [m]$ except with polynomially-small probability $\delta_0$ (as the sketch dimension is logarithmic in $1/\delta_0$), it is not hard to show that, even though our approximation could potentially miss some "heavy" rows, this has a minor effect on the correctness of the algorithm: the rows our algorithm selects will be larger than $(1 - \epsilon_0)$ times a pre-specified threshold, and the ones that are not chosen will be smaller than $(1 + \epsilon_0)$ times the threshold. This allows to set $\epsilon_0 = \Omega(1)$.

**Approximate SVD suffices.** Recall that in the PROJECT-TOSMALLROWS algorithm, expanding the subspace $V$ iteratively, requires to compute $\mathrm{SVD}(B_t^\top B_t)$. Since exact SVD is too costly for us, we wish to maintain an approximate SVD instead. Even though this may result in substantially different eigen-spectrum, we observe that a spectral-approximation of $\mathrm{SVD}(B_t^\top B_t)$ suffices for this subroutine, since we only need eigenvectors to be: (i) orthogonal to the row space of $A(I - V_t^\top V^t)$; (ii) orthonormal to each other. Our correctness analysis shows that an $\epsilon_B = \Theta(1)$ spectral approximation will preserve (up to constant factor) the row-norm guarantee of PROJECTTOSMALLROWS.

### 2.2.2. OVERCOMING THE BARRIERS

**Speeding-up the "hereditary-projection" step** In the PROJECTTOSMALLROWS subroutine in Larsen (2023), the matrix $B_t$ is used for (i) detecting rows with largest norms; (ii) extracting the largest rows to generate a matrix $\overline{B}$; and (iii) computing the eigenvectors of $\overline{B}^\top \overline{B}$. To optimize this process and reduce computational overhead, we avoid the explicit representation of matrix $B_t$ by substituting it with a product of appropriately-chosen sketching matrices. Given the aforementioned robustness-guarantees, Barriers in the first step can be bypassed straight-forwardly by using a JL sketch. Specifically, we utilize a random matrix $R \in \mathbb{R}^{n \times \epsilon_0^{-2}\log(1/\delta_0)}$ to obtain the compressed matrix $\widehat{B}_t :=  A(I - V_t^\top V_t)R$. Similarly, detecting rows with large $\ell_2$ norm can be done in the sketched subspace since we can preserve norms up to constant by choosing $\epsilon_0 = \Theta(1)$ and $\delta_0 = \delta/\mathrm{poly}(m,n)$, which reduces the time for querying row-norms from $O(mn^2)$ to $\widetilde{O}(\mathrm{nnz}(A) + n^\omega)$.

**Implicit leverage-score sampling** To address the hardness of computation of $\overline{B}_t^\top \overline{B}_t$ (overcoming Barrier 3), we use robust analysis to ensure that approximating the Top-$k$ SVD of $\overline{B}_t^\top \overline{B}_t$ (where $k \approx n/\log(m/n)$) using leverage-score sampling (Drineas et al., 2012; Clarkson & Woodruff, 2013; Nelson & Nguyên, 2013) preserves algorithm correctness. However, we lack explicit access to the input matrix $B_t$, so we must perform implicit leverage-score sampling w.r.t $B_t$ in $\sim \mathrm{nnz}(A)$ time. We propose IMPLICITLEVERAGESCORE (Algorithm 9), which takes $A$ and an orthonormal basis $V$ and generates a sparse embedding matrix $S_1$

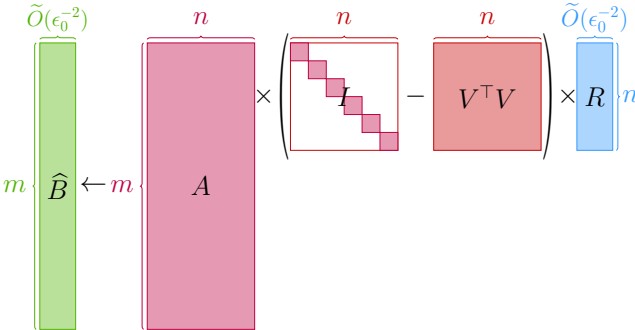

The original matrix $B$ by Larsen

*Figure 1.* We use the above sketch technique to reduce the time cost when computing the row norms. $\widehat{B} \in \mathbb{R}^{m \times \widetilde{O}(\epsilon_0^{-2})}$ is our sketched matrix. $A \in \mathbb{R}^{m \times n}$ is the original data matrix. $I \in \mathbb{R}^{n \times n}$ is an $n \times n$ unit matrix. $V^\top V \in \mathbb{R}^{n \times n}$ is the projection matrix onto the row span of $V$. And $R \in \mathbb{R}^{n \times \widetilde{O}(\epsilon_0^{-2})}$ is our JL sketching matrix. After sketched, we are able to fast query row norms of $\widehat{B}$, which are close to row norm of $B$ with an accuracy $\epsilon_0$. We select the rows from $B$ using this approximated norms. For the details of the selection operation, see Figure 2 and Figure 3.

to produce a compressed matrix $M$, whose QR factorization gives $R$. Using another sparse embedding matrix $S_2$, we calculate the compressed matrix $N$, which is used to compute the approximate leverage scores. This allows us to carry out the LSS lemma without computing $B_t$. With IMPLICITLEVERAGESCORE, we can generate a diagonal sampling matrix $\widetilde{D}$ in $\widetilde{O}(\mathrm{nnz}(A) + n^\omega)$ time. Using this subroutine to calculate $\widetilde{B}_t = \widetilde{D}_t B_t$ approximates the SVD of $\overline{B}_t^\top \overline{B}_t$. Finally, we prove that adding the eigenvectors of $\widetilde{B}_t^\top \widetilde{B}_t$ instead of $\overline{B}_t^\top \overline{B}_t$ to $V$ will still satisfy the required "oversampling" prerequisite for each row in $B$.

### 2.2.3. BEATING THE CUBIC BARRIER

**Where does the $n^3$ barrier arise from?** Recall that, in order to simulate the Edge-Walk process, the PARTIALCOL-ORING algorithm generates, in each iteration, a Gaussian vector g and projects it to $\mathrm{Span}(V)$. This requires a generic Mat-Vec product $(I - V_t^\top V_t)\mathrm{g}$, which takes $n^2$ time. Since $V_t$ is *dynamically changing* throughout iterates (depending on whether $|x_i + v_i + g_{V,i}| = 1$ and $|x_i + v_i| < 1$ is satisfied or not for each $i$), and there are $N = O(n)$ iterations, the OMv Conjecture (Henzinger et al., 2015) generally implies an $\Omega(n^3)$ runtime for implementing this iterative loop (Note that each $e_i$ can be added to $V$ at most once, $O(n)$ iterations indeed suffice). Nonetheless, we show how to re-implement PARTIALCOLORING using a "looka-head" (guess-and-correct) data structure, which combines FMM with low-rank corrections. We now describe the main ingredients of this data structure.

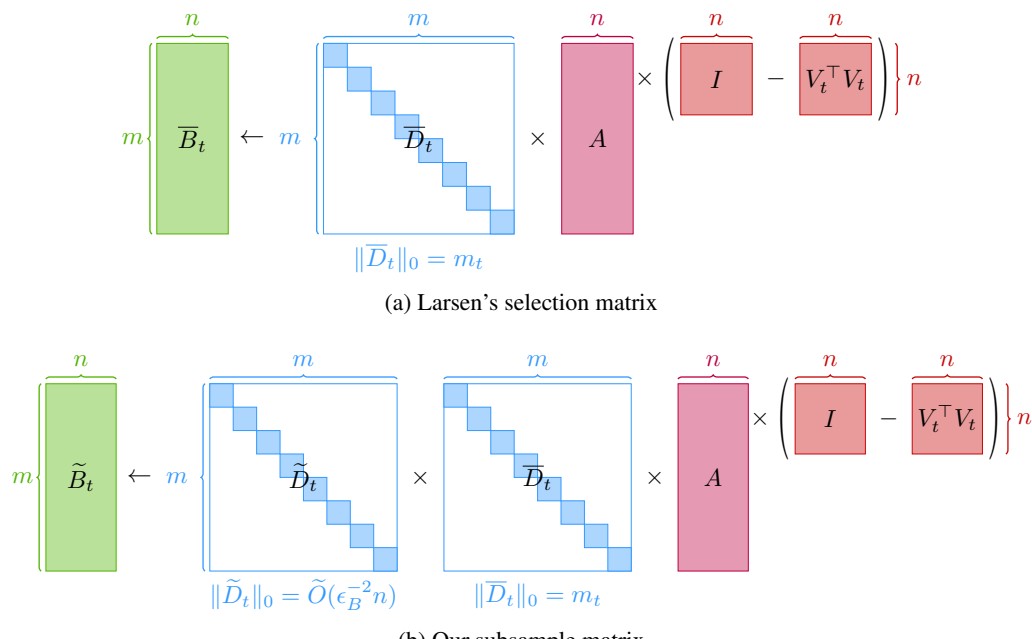

$$\|\overline{D}_t\|_0 = m_t$$

(a) Larsen's selection matrix

$$\|\widetilde{D}_t\|_0 = \widetilde{O}(\epsilon_B^{-2} n) \qquad \|\overline{D}_t\|_0 = m_t$$

(b) Our subsample matrix

*Figure 2.* This figure shows the difference of the row selection. Figure (a): Larsen's algorithm explicitly selected the $m_t$ largest rows from $B_t = A(I - V_t^\top V_t)$, and compute the eigenvectors of $\overline{B}_t^\top \overline{B}_t$, whose time cost is expensive. Figure (b): In our design, we use a subsample matrix $\widetilde{D}_t$ to generate the matrix $\widetilde{B}_t$, who has the eigenvalues close to $\overline{B}_t$. We use the above subsamping technique to reduce the time cost when computing the SVD decomposition. By using this, we can fast compute the eigenvectors.

**Precomputing Gaussian projections.** To overcome the $\sim n^2$ running time for computing the Gaussian projections for g, we add a preprocessing phase (INIT subroutine) to the PARTIALCOLORING iterative algorithm, which generates—in advance—a Gaussian matrix $\mathsf{G} \in \mathbb{R}^{n \times N}$ and stores the projection of *every* column of $\mathsf{G}$ to the row space of $V$, denoted $\{g_{V_t}\}_{g \in \mathsf{G}}$. We also design a QUERY procedure, which outputs any desired output vector $g = (I - V_t^\top V_t)\mathsf{g}$ on-demand. Since we do not know apriori if the subspace $V_t$ will change (and which coordinates $e_i$ will be added to $V_t$), we use a "Guess-and-Correct" approach: We guess the batch Gaussian projections, and then in iteration $t$, we perform a *low-rank* corrections via our update and query procedure in data structure. This idea is elaborated in more detail below.

**Lazy updates for the past rank-1 sequence.** Updating *all* the projections $\widetilde{g}$ stored in the data structure would result in prohibitively expensive runtime. Instead, we use the idea of *lazy updates*: We divide the columns of $\mathsf{G}$ into different batches, each batch having the size $K$. We also use a counter $\tau_u$ to denote which batch is being used currently, and initialize two counters $k_q$ and $k_u$ to record the times QUERY and UPDATE were called, respectively. Every time we call QUERY or UPDATE, we increment the counter by 1. When either $k_q$ or $k_u$ reaches the threshold $K$, we RESTART the process, and "accumulate" the current updates as well

as some clean-up operations for future iterations, adding up the vectors which are not present yet, and computes a new batch of $\widetilde{g}$'s for future use). This procedure runs in time $O(\mathcal{T}_{\mathrm{mat}}(n, K, n))$, and contributes $\mathcal{T}_{\mathrm{mat}}(n, K, n)$ time to the RESTART procedure.

Now, Recall that at the beginning of the PARTIALCOLORING algorithm, when we initialize the data structure, we compute the first $K$ projections $\widetilde{g}_1, \ldots, \widetilde{g}_k$. Then we enter the iteration. Recall in INIT, we generate a matrix $P$ which is defined to be $V^\top V$ for the input matrix $V$. And we precompute a batch of projections onto the row space of $V$. ($\widetilde{G} = P \cdot \mathsf{G}_{*,S}$, where $S = [K]$ at the beginning.) Besides, if there is some rows are added to $V$ during the running of the algorithm, we first find its factor that is vertical to the row space of $V$. Then we rescale this vector to unit and name it $w$. We maintain at most $K$ $w$'s. Then through the running of the algorithm, when we call the QUERY, we just simply select the corresponding row in $\widetilde{G}$, denoted as $\widetilde{g}$, and output the vector $g - \widetilde{g} - \sum_{i=1}^{k_u} w_i w_i^\top g$. Recall that the number of $g$ we pre-computed and the number of $w$ we maintained are both limited to be less than $K$, when one of them reach the limit, we call the RESTART procedure. When this condition happens, QUERY and UPDATE will take $\mathcal{T}_{\mathrm{mat}}(K, n, n)$. Thus by applying this *lazy update* idea, we finally reach the subcubic running time. The final runtime of PARTIALCOLORING using our data structure is therefore a tradeoff based on the choice of batch size $K$. Denoting

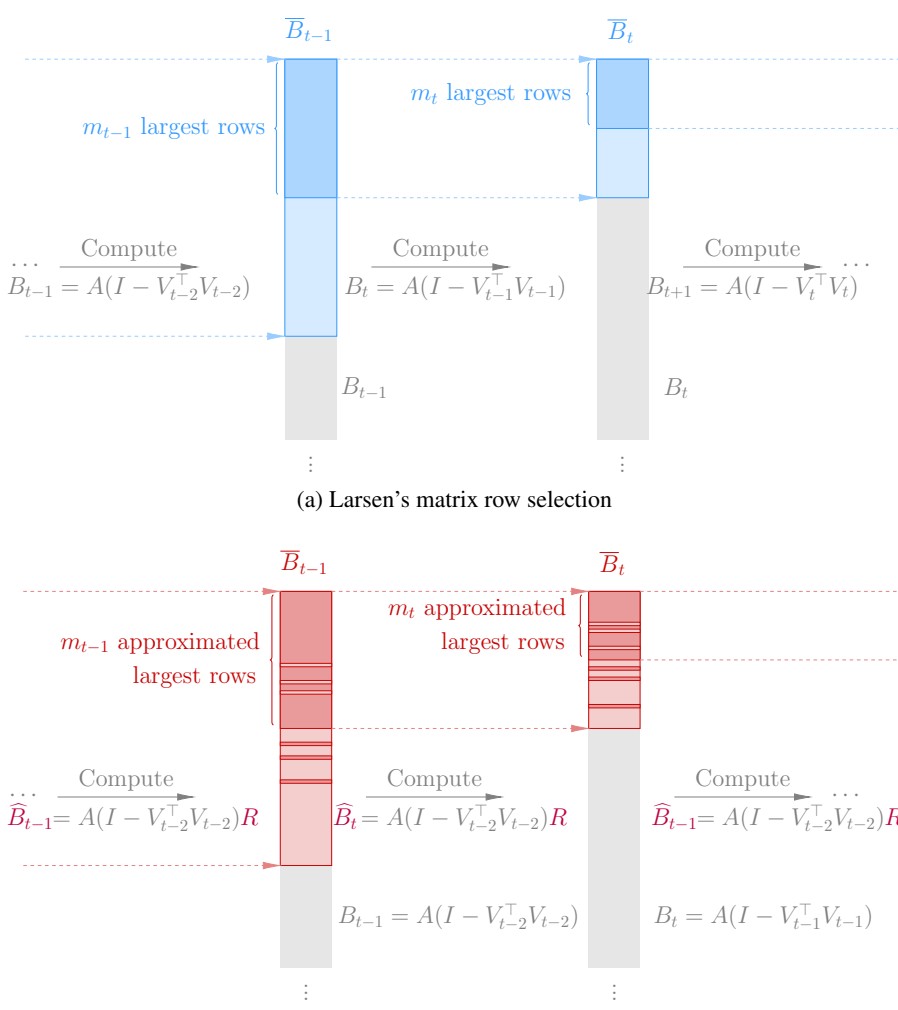

(a) Larsen's matrix row selection

(b) Our matrix row selection

*Figure 3.* This figure shows the idea of the operation at the $t$-th iteration in the projection algorithm. The figure (a) shows the idea of Larsen's, and the figure (b) shows the idea of ours. Figure (a): Larsen's algorithm selects the $m_t$ largest rows from the matrix $B_t = A(I - V_{t-1}^\top V_{t-1})$, then the rest of the rows not selected will have norm less than the threshold $C_0 \cdot T \cdot \mathrm{herdisc}(A)$. But the time cost of row norm computation is expensive. Figure (b): we compute the sketched matrix $\widetilde{B}_t = A(I - V_{t-1}^\top V_{t-1})R$, where $R$ is an JL matrix. Then we select the rows based on the approximated norms. Thus we significantly reduce the time cost of norm computation. We show that, under our setting, the norm of the rows not selected will have another guarantee, that is, $(1 + \epsilon_0) \cdot C_0 \cdot T \cdot \mathrm{herdisc}(A)$. This constant loss will still make our algorithm correct. (Since our row norm computation is approximated, there is a constant loss effecting the selecting operation. To demonstrate this, we use the darker red to demonstrate the real largest rows without being selected.)

$K = n^a$, we get a total runtime of

$$\widetilde{O}(\mathrm{nnz}(A) + n^\omega + n^{2+a} + n^{1+\omega(1,1,a)-a}).$$

For the current upper bound on the fast rectangular matrix multiplication function $\omega(\cdot, \cdot, \cdot)$ (Alman et al., 2025), setting $a = 0.53$ yields an optimal overall runtime time of $\widetilde{O}(\mathrm{nnz}(A) + n^{2.53})$. We note that even an ideal value of $\omega$ would not improve this result by much, as it would merely enable setting $a = 0.5$ which in turn would translate into an $n^{2.5}$ time algorithm, and this is the limit of our approach.

**Faster Iterative Coloring**   Recall that with the approximate small-projection matrix in hand, we still need to overcome Barriers 4 and 5 above. Computing the heaviest row $\eta := \max_{j \in [m]} \|a_j^\top (I - V^\top V)\|_2$ (Barrier 1) can again be done (approximately) via the JL-sketch (as in the first step), by computing

$$\widehat{\eta} := \max_{j \in [m]} \|a_j^\top (I - V^\top V)R\|_2$$

using the JL matrix $R$. Since $R$ has only $O(\epsilon_1^{-2} \log(1/\delta_1))$ columns (where choosing $\epsilon_1 = \Theta(1)$ and $\delta_1 = \delta/\mathrm{poly}(mn)$ is sufficient ) this step reduces from

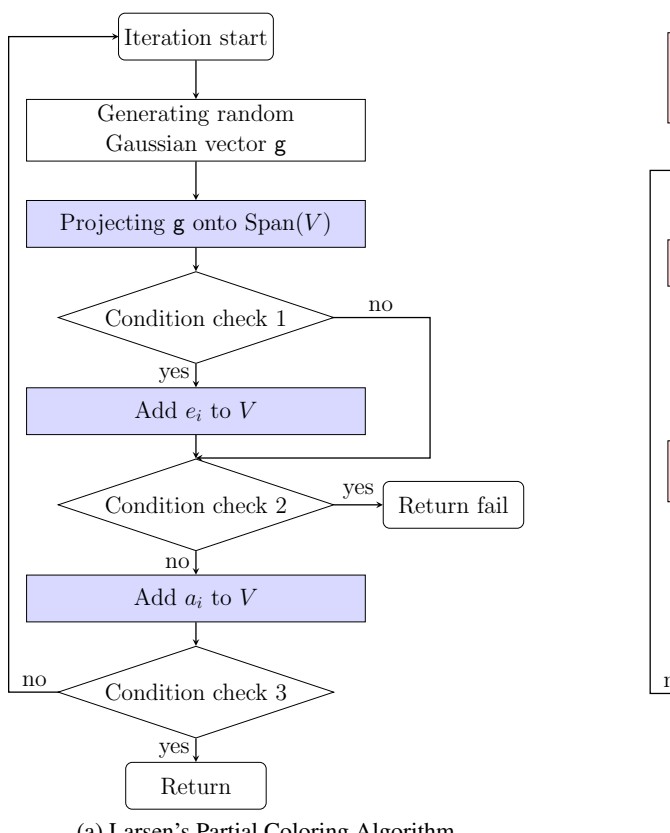

(a) Larsen's Partial Coloring Algorithm

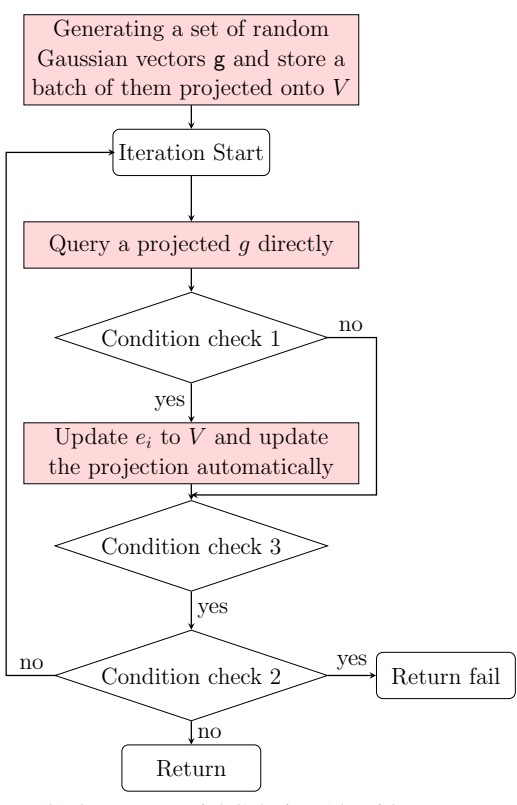

(b) Our Fast Partial Coloring Algorithm

*Figure 4.* The flowchart shows different design of the partial coloring algorithm of Larsen (2023) and ours. In figure (a), the blue blocks are the steps causing the mainly time cost. In figure (b), the red blocks are the steps we design to overcome those time costs. The $\mathbf{g} \in \mathbb{R}^n$ is the new-generated Gaussian vector and $g$ is the projected vector. That is, $g := (I - V^\top V)\mathbf{g}$.

$\mathcal{T}_{\mathrm{mat}}(m, n, n)$ time to $\widetilde{O}(n^2)$.

As discussed earlier, Barrier 5 is a different ballgame and major obstruction to speeding up Larsen's algorithm, since the verification step

$$\mathcal{E}_\tau := \bigwedge_{j=1}^{m} (|\langle a_j, v + g\rangle| \geq \tau \ \wedge \ |\langle a_j, v\rangle| < \tau) \quad (1)$$

needs to be performed *adaptively* in each of the $n$ iterations, which appears impossible to perform in $\ll mn^2$ time given the OMv Conjecture (Henzinger et al., 2015). Fortunately, it turns out we can *avoid* this verification step using a slight change in the analysis of Larsen (2023): Larsen's analysis shows the event $\mathcal{E}_\tau$ in Eq. (1) happens with constant probability. By slightly increasing the threshold to $\tau' = \tau(\delta)$, we can ensure the event $\mathcal{E}_{\tau'}$ happens with probability $1 - \delta$. Setting $\delta$ to a small enough constant so as not to affect the other parts of the algorithm, we can avoid this verification step altogether. At this point, the entire algorithm can be boosted to ensure high-probability of success. combining these ideas yields a (combinatorial) coloring algorithm that runs in $\widetilde{O}(\mathrm{nnz}(A) + n^3)$. Next, we turn to explain the new idea required to overcome the cubic term $n^3$, which is important for the near-square case ($m \approx n$).

## 3. Conclusion

In this work, we address the longstanding challenge of discrepancy minimization for real-valued matrices with a focus on achieving input-sparsity time algorithms. By introducing novel algorithmic components such as implicit leverage-score sampling and lazy update, our combinatorial algorithm achieves a runtime of $\widetilde{O}(\mathrm{nnz}(A) + n^3)$ and a FMM-based variant breaks the cubic barrier to reach $\widetilde{O}(\mathrm{nnz}(A) + n^{2.53})$. We significantly improve the computational efficiency while matching the approximation guarantees of existing methods. Our approach not only introduces novel algorithmic components but also demonstrates the potential of combining sketching and fast matrix multiplication in advancing computational geometry and optimization. We believe these techniques are of independent interest. Future work could extend these ideas to reduce the runtime of other problems in combinatorial optimization, or adapt techniques to tackle the problems in streaming and distributed models.

## Acknowledgements

The author would like to thank the anonymous reviewer of ICML 2025 for their highly insightful suggestions.

## Impact Statement

This paper presents work whose goal is to advance the field of Machine Learning and Optimization. There are many potential societal consequences of our work, but none of which we feel must be specifically highlighted here.

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

# Appendix

**Rodamap.** We organize the appendix as follows. In Section A, we give the preliminaries of the paper, including the notations we use, the definitions and some useful results from prior works. In Section B, we provide previous tools and results on sketching. In Section C, we present our first result, a fast algorithm of projecting matrix into small rows. In Section D, we present the final algorithm of hereditary minimize. In Section E, we present the second main result, the fast partial coloring algorithm. In Section F, we discuss the fast maintaining algorithm we use to boost the running time. In Section G, we provide our implicit leverage score sampling algorithm. In Section H, we give a sparsification tool for the matrix that has pattern $A^\top A$. In Section I, we provide the algorithm to find proper step size in Partial Coloring algorithm. In Section J, we provide empirical validation of the efficiency of our algorithm. In Section K, we discuss more related work on linear programming, semidefinite programming, and applications of discrepancy theory in machine leraning.

# A. Preliminaries

## A.1. Notations

For a vector $x \in \mathbb{R}^n$, we use $\|x\|_1$ to denote its entrywise $\ell_1$ norm. We use $\|x\|_\infty$ to denote its entrywise $\ell_\infty$ norm. For a matrix $X$, we use $\|X\|$ to denote the spectral norm. We use $e_i$ to denote the vector where $i$-th location is 1 and everywhere else is 0. For a matrix $A$, we say matrix $A$ positive semi-definite if $A \succeq 0$, i.e., $x^\top A x \geq 0$ for all vectors $x$. We say $A \succeq B$, if for all $x \in \mathbb{R}^d$, $x^\top A x \geq x^\top B x$. For a positive semi-definite matrix $A$, let $U\Sigma U^\top$ be the SVD decomposition of $A$. We use $A^{1/2}$ to denote $U\Sigma^{1/2}U^\top$. We use $A^{-1/2}$ to denote $U\Sigma^{-1/2}U^\top$ where $\Sigma^{-1/2}$ is diagonal matrix that $i, i$-entry is $\Sigma_{i,i}^{-1/2}$ if $\Sigma_{i,i} \neq 0$ and $i, i$-th entry is 0 if $\Sigma_{i,i} = 0$. For a matrix $A$, we use $\mathrm{nnz}(A)$ to denote the number of non-zeros in matrix $A$.

## A.2. Definitions

Here, we define disc and herdisc.

**Definition A.1** (Discrepancy). *For a real matrix $A \in \mathbb{R}^{m \times n}$, and a vector $x \in \{1, -1\}^n$. We define the discrepancy of $A$ to be*

$$\mathrm{disc}(A, x) := \|Ax\|_\infty = \max_{i \in [m]} |(Ax)_i|.$$

*And we define the discrepancy of the matrix $A$ as*

$$\mathrm{disc}(A) := \min_{x \in \{1, -1\}^n} \mathrm{disc}(A, x).$$

**Definition A.2** (Hereditary Discrepancy). *For a real matrix $A \in \mathbb{R}^{m \times n}$. Let $\mathcal{A}$ be defined as the set of matrices obtained from A by deleting some columns from A. We define the hereditary discrepancy of A to be*

$$\mathrm{herdisc}(A) := \max_{B \in \mathcal{A}} \mathrm{disc}(B),$$

*where* disc *is defined as Definition A.1.*

## A.3. Basic Linear Algebra

**Lemma A.3.** *Given two psd matrices $A \in \mathbb{R}^{n \times n}$ and $B \in \mathbb{R}^{n \times n}$. If*

$$(1 - \epsilon)A \preceq B \preceq (1 + \epsilon)A$$

*then, we have*

$$(1 - \epsilon)\lambda_i(A) \preceq \lambda_i(B) \preceq (1 + \epsilon)\lambda_i(A).$$

*Note that, $\lambda_i(A)$ is the $i$-the largest eigenvalue of A.*

### A.4. Prior Results on Herdisc

We state a tool from previous work.

**Theorem A.4** (Larsen (2017)). *Given a real matrix $A \in \mathbb{R}^{m \times n}$, let $\lambda_1 \geq \lambda_2 \geq \cdots \geq \lambda_n \geq 0$ be the eigenvalues of $A^\top A$. For all positive integers $k \leq \min\{m, n\}$, we have*

$$\text{herdisc}(A) \geq \frac{k}{2e} \sqrt{\frac{\lambda_k}{mn}}.$$

### A.5. Properties of Gaussians

We state two statements about the Gaussian variables.

**Claim A.5.** *Let $t$ be a random variable such that $t \sim \mathcal{N}(0, 1)$. Then for any $\lambda > 0$, we have that*

$$\Pr[|t| > \lambda] \leq 2 \exp(-\lambda^2/2).$$

**Claim A.6** (Claim 2 in Larsen (2023)). *For an arbitrary matrix $V \in \mathbb{R}^{l \times n}$ such that $l \leq n$ and all rows of $V$ form an orthogonal basis. Let $g \in \mathbb{R}$ be a vector which is sampled with $n$ i.i.d. $\mathcal{N}(0, 1)$ distributed coordinates. Then for any arbitrary vector $a \in \mathbb{R}^n$ we have that*

$$\langle a, (I - V^\top V)g \rangle \sim \mathcal{N}(0, a\|I - V^\top V\|^2).$$

### A.6. Azuma for Martingales with Subgaussian Tails

Then for martingales with subgaussian tails, we introduce the following Azuma's inequality. We first define the following martingale difference sequence.

**Definition A.7** (Martingale Difference Sequence). *For two sequences of random variables $\mathbb{A} = \{A_1, A_2, \ldots\}$ and $\mathbb{B} = \{B_1, B_2, \ldots\}$. If for any integer $t \in \mathbb{Z}_+$, $A_{t+1}$ is a function of $\{B_1, \ldots, B_t\}$ and the following holds,*

$$\Pr[\mathbb{E}[A_{t+1} \mid B_1, \ldots, B_t] = 0] = 1,$$

*then we say that $\mathbb{A}$ is a* martingale difference sequence *with respect to $\mathbb{B}$.*

Then we introduce the following Azuma's inequality for martingales with Subgaussian tails.

**Theorem A.8** (Shamir (2011)). *Let $\delta \in (0, 1)$ be an arbitrary failure probability. Let $\mathbb{A} = \{A_1, \ldots, A_T\}$ be a martingale difference sequence with respect to a sequence $\mathbb{B} = \{B_1, \ldots, B_T\}$. Let $b > 1$ and $c > 0$ be two constants. If for any integer $t$ and all $a \in \mathbb{R}_+$, the following holds,*

$$\max\{\Pr[A_t > a \mid B_1, \ldots, B_{t-1}], \Pr[A_t < -a \mid B_1, \ldots, B_{t-1}]\} \leq b \exp(-ca^2).$$

*Then it holds that*

$$|\frac{1}{T} \sum_{t=1}^{T} A_t| \leq 2\sqrt{\frac{28b \lg(1/\delta)}{cT}}$$

*with probability at least $1 - \delta$.*

### A.7. Concentration Inequality

We state the matrix Bernstein bound as follows:

**Lemma A.9** (Matrix Bernstein Bound (Tropp, 2011; Tropp et al., 2015)). *Let $X_1, \ldots, X_s$ be independent copies of a symmetric random matrix $X \in \mathbb{R}^{n \times n}$ with $\mathbb{E}[X] = 0$, $\|X\| \leq \gamma$ almost surely and $\|\mathbb{E}[X^\top X]\| \leq \sigma^2$. Let $W = \frac{1}{s} \sum_{i \in [s]} X_i$. For any $\epsilon \in (0, 1)$,*

$$\Pr[\|W\| \geq \epsilon] \leq 2n \cdot \exp\left(-\frac{s\epsilon^2}{\sigma^2 + \gamma\epsilon/3}\right).$$

*Note the $\|W\|$ is the spectral norm of matrix $W$.*

## A.8. Fast Matrix Multiplication

We describe several basic backgrounds about fast matrix multiplication.

**Definition A.10.** *Given three integers $n_1, n_2, n_3$, we use $\mathcal{T}_{\mathrm{mat}}(n_1, n_2, n_3)$ to denote time of multiplying a $n_1 \times n_2$ matrix with another $n_2 \times n_3$ matrix.*

It is known that the ordering of $n_1, n_2, n_3$ only affects the constant factor in the time complexity of matrix multiplication.

**Fact A.11** (Bürgisser et al. (2013); Bläser (2013)).

$$\mathcal{T}_{\mathrm{mat}}(n_1, n_2, n_3) = O(\mathcal{T}_{\mathrm{mat}}(n_1, n_3, n_2)) = O(\mathcal{T}_{\mathrm{mat}}(n_2, n_1, n_3)).$$

For convenience, we also define the $\omega(\cdot, \cdot, \cdot)$ function and the exponent $\omega$ of matrix multiplication (Williams, 2012; Gall & Urrutia, 2018; Alman & Williams, 2021; Williams et al., 2024; Alman et al., 2025).

**Definition A.12.** *For $a, b, c > 0$, we use $n^{\omega(a,b,c)}$ to denote the time of multiplying an $n^a \times n^b$ matrix with another $n^b \times n^c$ matrix. We denote $\omega := \omega(1, 1, 1)$ as the exponent of matrix multiplication. we denote $\alpha$ as the dual exponent of matrix multiplication, i.e., $2 = \omega(1, 1, \alpha)$.*

By the state-of-the-art fast matrix multiplication result in Alman et al. (2025), we have

**Lemma A.13** (Alman et al. (2025)). *We have*

- $\omega = \omega(1, 1, 1) = 2.371$.

- $\omega(1, 1, 0.5275) = 2.055$.

# B. Previous Tools and Results on Sketching

Here in this section, we provide tools and previous results on sketching matrices. In Section B.1 we introduce some sketching matrices. In Section B.2 we provide traditional JL transform. In Section B.3 we give some previous results on subspace embedding.

## B.1. Sketching Matrices

**Definition B.1** (Random Gaussian matrix or Gaussian transform, folklore ). *Let $S = \sigma \cdot G \in \mathbb{R}^{s \times m}$ where $\sigma$ is a scalar, and each entry of $G \in \mathbb{R}^{s \times m}$ is chosen independently from the standard Gaussian distribution. For any matrix $A \in \mathbb{R}^{m \times n}$, $SA$ can be computed in $O(s \cdot \mathrm{nnz}(A))$ time.*

**Definition B.2** (AMS (Alon et al., 1996)). *Let $h_1, h_2, \ldots, h_b$ be $b$ random hash functions picking from a $4$-wise independent hash family $\mathcal{H} = \{h : [n] \to \{-\frac{1}{\sqrt{b}}, +\frac{1}{\sqrt{b}}\}\}$. Then $R \in \mathbb{R}^{b \times n}$ is a AMS sketch matrix if we set $R_{i,j} = h_i(j)$.*

**Definition B.3** (CountSketch (Charikar et al., 2002)). *Let $h : [n] \to [b]$ be a random $2$-wise independent hash function and $\sigma : [n] \to \{+1, -1\}$ be a random $4$-wise independent hash function. Then $R \in \mathbb{R}^{b \times n}$ is a count-sketch matrix if we set $R_{h(i),i} = \sigma(i)$ for all $i \in [n]$ and other entries to zero.*

**Definition B.4** (Sparse Embedding Matrix I (Nelson & Nguyên, 2013)). *We say $R \in \mathbb{R}^{b \times n}$ is a sparse embedding matrix with parameter $s$ if each column has exactly $s$ non-zero elements being $\pm 1/\sqrt{s}$ uniformly at random, whose locations are picked uniformly at random without replacement (and independent across columns).*

**Definition B.5** (Sparse Embedding Matrix II (Nelson & Nguyên, 2013)). *Let $h : [n] \times [s] \to [b/s]$ be a random $2$-wise independent hash function and $\sigma : [n] \times [s] \to \{-1, +1\}$ be a $4$-wise independent. Then $R \in \mathbb{R}^{b \times n}$ is a sparse embedding matrix II with parameter $s$ if we set $R_{(j-1)b/s+h(i,j),i} = \sigma(i,j)/\sqrt{s}$ for all $(i,j) \in [n] \times [s]$ and all other entries to zero.*

**Definition B.6** (CountSketch + Gaussian transform, Definition B.18 in Song et al. (2019a)). *Let $S' = S\Pi$, where $\Pi \in \mathbb{R}^{t \times m}$ is the COUNTSKETCH transform (defined in Definition B.3) and $S \in \mathbb{R}^{s \times t}$ is the Gaussian transform (defined in Definition B.1). For any matrix $A \in \mathbb{R}^{m \times n}$, $S'A$ can be computed in $O(\mathrm{nnz}(A) + nts^{\omega-2})$ time, where $\omega$ is the matrix multiplication exponent.*

## B.2. JL Transform

We define Johnson–Lindenstrauss transform (Johnson & Lindenstrauss, 1984) as follows:

**Definition B.7** (Johnson & Lindenstrauss (1984))**.** *Let $\epsilon \in (0, 1)$ be the precision parameter. Let $\delta \in (0, 1)$ be the failure probability. Let $A \in \mathbb{R}^{m \times n}$ be a fixed matrix. Let $a_i^\top$ denote the $i$-th row of matrix $A$, for all $i \in [m]$. We say $R$ is an $\epsilon, \delta$-JL transform if with probability at least $1 - \delta$,*

$$(1 - \epsilon)\|a_i\|_2 \le \|Ra_i\|_2 \le (1 + \epsilon)\|a_i\|_2, \ \forall i \in [m].$$

It is well-known that random Gaussian matrices and AMS matrices give JL-transform property.

**Lemma B.8** (Johnson–Lindenstrauss transform (Johnson & Lindenstrauss, 1984))**.** *Let $\epsilon \in (0, 1)$ be the precision parameter. Let $\delta \in (0, 1)$ be the failure probability. Let $A \in \mathbb{R}^{m \times n}$ be a real matrix . Then there exists an sketching matrix $R \in \mathbb{R}^{\epsilon^{-2} \log(mn/\delta) \times n}$ (defined as Definition B.1 or Definition B.2), such that the following holds with probability at least $1 - \delta$,*

$$(1 - \epsilon)\|a_i\|_2 \le \|Ra_i\|_2 \le (1 + \epsilon)\|a_i\|_2, \ \forall i \in [m],$$

*where for a matrix $A$, $a_i^\top$ denotes the $i$-th row of matrix $A \in \mathbb{R}^{m \times n}$.*

The JL Lemma is known to be tight due to Larsen & Nelson (2017).

## B.3. Subspace Embedding

We first define a well-known property called subspace embedding.

**Definition B.9** (Subspace embedding (Sarlos, 2006))**.** *A $(1 \pm \epsilon)$ $\ell_2$-subspace embedding for the column space of an $m \times n$ matrix $A$ is a matrix $S$ for which for all $x \in \mathbb{R}^n$*

$$\|SAx\|_2^2 = (1 \pm \epsilon)\|Ax\|_2^2.$$

*Let $U$ denote the orthonormal basis of $A$, then it is equivalent to the all the singular values of $SU$ are within $[1 - \epsilon, 1 + \epsilon]$ with probability $1 - \delta$.*

Nelson & Nguyên (2013) shows that $r = O(\epsilon^{-2} n \log^8(n/\delta))$ and column sparsity $O(\epsilon^{-1} \log^3(n/\delta))$ suffices for subspace embedding (See Theorem 9 in Nelson & Nguyên (2013)). Later Cohen (2016a) improves the result to $r = O(\epsilon^{-2} n \log(n/\delta))$ and column sparsity is $O(\epsilon^{-1} \log(n/\delta))$ (see Theorem 4.2 in Cohen (2016a)).

**Lemma B.10** (Nelson & Nguyên (2013); Cohen (2016a))**.** *Let $A \in \mathbb{R}^{m \times n}$ be a matrix. Let $S \in \mathbb{R}^{r \times m}$ denote the sketching matrix (defined as Definition B.4). If $r = O(\epsilon^{-2} n \log(n/\delta))$ and the column sparsity of $S$ is $s = O(\epsilon^{-1} \log(n/\delta))$, then $S$ satisfies that with probability $1 - \delta$*

$$\|SAx\|_2^2 = (1 \pm \epsilon)\|Ax\|_2^2.$$

*Further, $SA$ can be done in $(s \cdot \mathrm{nnz}(A))$ time.*

# C. Small Row Projection via Implicit Leverage-Score Sampling

Here in this section, we build our algorithm of projecting a matrix to small rows. In Section C.1 we present the orghogonalize subroutine which is useful in later algorithms. In Section C.2 we present our projection algorithm together with its analysis.

## C.1. Orthogonalize Subroutine

Here in this section, we present the following algorithm named ORTHOGONALIZE, which is just the Gram-Schmidt process. This subroutine is very useful in the construction of the following algorithms.

The following lemma gives the running time of the above algorithm, with input vector sparsity time.

**Lemma C.1.** *The algorithm (ORTHOGONALIZE in Algorithm 1) runs $O(\|s\|_0 \cdot n)$ time.*

---

**Algorithm 1** Algorithm 1 in Larsen (2023)

---

1: **procedure** ORTHOGONALIZE($s \in \mathbb{R}^n, V \in \mathbb{R}^{l \times n}$)                    ▷ Lemma C.1
2:     $s' \leftarrow (I - V^\top V)s$
3:     **if** $s' \neq 0$ **then**
4:         add $s'/\|s'\|_2$ as a row of $V$
5:     **end if**
6:     **return** $V$
7: **end procedure**

---

*Proof.* The running time of ORTHOGONALIZE can be divided as follows,

- Line 2 takes time $O(\|s\|_0 \cdot n)$ to compute $s'$, where for a vector $x \in \mathbb{R}^d$, $\|x\|_0$ denotes the $\ell_0$ norm of $x$.

- Line 4 takes time $O(n)$ to compute $s'/\|s'\|_2$.

Thus we have the total running time of $O(\|s\|_0 \cdot n)$.                                    □

## C.2. Fast Projecting to Small Rows

Here in this section, we present our first main result, i.e., the faster algorithm for projecting a matrix to small rows. We first show the algorithm as follows.

### C.2.1. RUNNING TIME

The goal of this section is to prove running time of Algorithm 2. We have the following lemma.

**Theorem C.2** (Running time of Algorithm 2). *The algorithm (FASTPROJECTTOSMALLROWS in Algorithm 2) runs $\widetilde{O}(\mathrm{nnz}(A) + n^\omega)$ time[7]. The succeed probability is $1 - \delta$. Note that $\widetilde{O}$ hides $\log(n/\delta)$.*

*Proof.* The running time of FASTPROJECTTOSMALLROWS can be divided as follows,

- Run the following lines for $i \in [T]$ times, where $T = O(\log(m/n))$:
  - Line 14 needs to compute $\widehat{B}_t$, it takes the following time
  $$O(\mathrm{nnz}(A)r + n^\omega + n^2 r) = \widetilde{O}(\mathrm{nnz}(A) + n^\omega)$$
  - Line 18 takes time $\widetilde{O}(m)$ to construct $\overline{D}_t$ by computing the norms, and takes time $\widetilde{O}(m)$ to find the largest $m_{t-1}$ of them.
  - Using Lemma G.4, we can compute $\widetilde{D}_t$ in $\widetilde{O}(\mathrm{nnz}(A) + n^\omega)$ time

- Line 31 runs ORTHOGONIZE for $O(n)$ times, and takes $O(n)$ each time.

Adding these together, we have the total running time of

$$\widetilde{O}(\mathrm{nnz}(A) + n^\omega).$$

□

### C.2.2. CORRECTNESS

Here we present the correctness theorem of the Algorithm 2, and together with its proof.

**Theorem C.3** (Correctness of Algorithm 2 ). *For any given $m \times n$ matrix $A$, there is an algorithm that takes $A$ as input and output a matrix $V \in \mathbb{R}^{n/4 \times n}$ such that*

---

[7]Note that $\omega$ denotes the exponent of matrix multiplication, currently $\omega \approx 2.373$ (Williams, 2012; Alman & Williams, 2021).

---

**Algorithm 2** Our input sparsity projection algorithm (this improves the Algorithm 2 in (Larsen, 2023))

---

1: **procedure** FASTPROJECTTOSMALLROWS($A \in \mathbb{R}^{m \times n}, \delta \in (0, 1)$)          ▷ Theorem C.3
2:      $V_1 \in 0^{0 \times n}$          ▷ Initialize an empty matrix
3:      $T \leftarrow \log(8m/n)$
4:      $\delta_0 \leftarrow \Theta(\delta/(mT))$
5:      $\epsilon_0 \leftarrow 0.01$
6:      $r \leftarrow \epsilon_0^{-2} \log(1/\delta_0)$
7:      $\delta_B \leftarrow \Theta(\delta/T)$
8:      $\epsilon_B \leftarrow 0.1$
9:      **for** $t = 1 \rightarrow T$ **do**
10:          $l_t \leftarrow t \cdot \frac{n}{8T}$
11:          $m_t \leftarrow m/2^t$
12:          /*Ideally, we need to compute $B_t \leftarrow A(I - V_t^\top V_t) \in \mathbb{R}^{m \times n}$, but we have no time to do that.*/
13:          Let $R \in \mathbb{R}^{n \times r}$ denote JL sketching matrix (Either Definition B.2 or Definition B.1)
14:          Compute $\widehat{B}_t \leftarrow A(I - V_t^\top V_t)R$          ▷ $\|\widehat{B}_{t,j}\|_2 \in (1 \pm \epsilon_0)\|B_{t,j}\|_2, \forall j \in [m]$, Lemma B.8
15:          /* Ideally, we need to compute $\|B_j\|_2$, but we had no time to do that.*/
16:          Compute $\|\widehat{B}_{t,j}\|_2$ for all $j \in [m]$
17:          $\overline{S}_t \subset [m]$ denote the indices of $\widehat{B}_t$ such that it contains $m_{t-1}$ rows of the largest norm
18:          Let $\overline{D}_t$ denote a sparse diagonal matrix where $(i, i)$-th location is 1 if $i \in \overline{S}_t$ and 0 otherwise
19:          Let $\overline{B}_t = \overline{D}_t A(I - V_t^\top V_t) \in \mathbb{R}^{m_{t-1} \times n}$ be the submatrix obtained from $B_t$
20:          /* Ideally, we need to compute $\overline{B}_t^\top \overline{B}_t$, but in algorithm we had no time to do that.*/
21:          $\widetilde{D}_t \leftarrow$ SUBSAMPLE($\overline{D}_t A, V_t, \epsilon_B, \delta_B$)          ▷ Algorithm 10
22:          $\widetilde{B}_t \leftarrow \widetilde{D}_t A(I - V_t^\top V_t)$
23:                                         ▷ Note that $\overline{\sigma}_i = \lambda_i(\overline{B}_t^\top \overline{B}_t)$ and $\widetilde{\sigma}_i = \lambda_i(\widetilde{B}_t^\top \widetilde{B}_t)$
24:          /* Ideally, we need to compute the eigenvalues $\overline{\sigma}_1 \geq \cdots \geq \overline{\sigma}_n \geq 0$ and corresponding eigenvectors $\overline{u}_1, \cdots, \overline{u}_n$ of $\overline{B}_t^\top \overline{B}_t$, but in algorithm we had no time to that*/
25:          Let $\widetilde{u}_1, \cdots \widetilde{u}_n \in \mathbb{R}^n$ denote the eigenvectors of SVD decomposition of $\widetilde{B}_t^\top \widetilde{B}_t = \widetilde{U}\widetilde{\Sigma}\widetilde{U}^\top$
26:          Construct matrix $V_{t+1} \in \mathbb{R}^{l_{t+1} \times n}$ by adding rows vectors $\{\widetilde{u}_1, \cdots, \widetilde{u}_{n/(8T)}\}$ to the bottom $V_t \in \mathbb{R}^{l_t \times n}$
27:      **end for**
28:      $B_T \leftarrow A \cdot (I - V_{T+1}^\top V_{T+1})$
29:      Let $r_1, \cdots, r_{n/8}$ be the $n/8$ rows of $B_T$ with largest norm
30:      **for** $j = 1 \rightarrow n/8$ **do**
31:          $V \leftarrow$ ORTHOGONALIZE($r_j, V$)          ▷ Algorithm 1
32:      **end for**
33:      **return** $V$          ▷ $V \in \mathbb{R}^{\ell \times n}$ and $\ell \leq n/4$
34: **end procedure**

---

- *having unit length orthogonal rows*

- *all rows of $A(I - V^\top V)$ have norm at most $O(\text{herdisc}(A)\log(m/n))$,*

*holds with probability at least $1 - \delta$.*

*Proof.* We define

$$T := \log(8m/n),$$

which is the time of the main loop of Algorithm 2 execute.

We define

$$C_0 := 1000$$

to be a constant used later.

And we define

$$m_t := m/2^t$$

to denote the number of rows in $\overline{B}$ at each iteration. We then continue the proof in the following paragraphs.

**The rows of $V$ form an orthonormal basis.** To start with, let us show that, the output matrix $V$ form an orthonormal basis with its rows. Since $V_0$ has no rows, the claim is true initially.

We first fix $t \in T$, consider at $t$-th iteration, after constructing $V_{t+1}$ by adding row vectors

$$\{\widetilde{u}_1, \ldots, \widetilde{u}_{n/(8T)}\}$$

to the bottom of $V_t$ in Line 26.

Note that these are eigenvectors, so they must be orthogonal to each other and for any $u$ of them, $\|u\|_2 = 1$. Moreover, we have that

$$\{\widetilde{u}_1, \ldots, \widetilde{u}_{n/(8T)}\} \subseteq \mathrm{span}(\widetilde{B}_t).$$

We note that, since we define $\widetilde{B}_t$ as

$$\widetilde{B}_t = \widetilde{D}_t A(I - V_t^\top V_t),$$

all rows of $\widetilde{B}_t$ muse be orthogonal to any rows of $V_t \in \mathbb{R}^{l_t \times n}$, and it follows that adding the above set of eigenvectors

$$\{\widetilde{u}_1, \ldots, \widetilde{u}_{n/(8T)}\}$$

as rows of $V_t \in \mathbb{R}^{l_t \times n}$ maintains the rows of $V_{t+1} \in \mathbb{R}^{l_{t+1} \times n}$ form an orthonormal basis.

After the first for-loop, we claim obviously that, the last for-loop (Line 31) preserves the property of $V$ such that, $V$ form an orthonormal basis with its rows.

**Row norm guarantee, proved with induction.** We now claim that, after the $t$-th iteration of the first for-loop (Line 9 to Line 27), we have that

$$|\{i \in [m] \mid \|b_i\|_2 \geq (1 + \epsilon_0) \cdot C_0 \cdot T \cdot \mathrm{herdisc}(A)\}| \leq m_t,$$

i.e., there are at most $m_t$ rows in $B_t = A(I - V_t^\top V_t)$ having norm larger than $(1 + \epsilon_0) \cdot C_0 \cdot T \cdot \mathrm{herdisc}(A)$. We first note that, for $t = 0$ this holds obviously.

Then have the following inductive assumption:

*After iteration $t - 1$, there are at most $m_{t-1}$ rows have norm larger than $(1 - \epsilon_0) \cdot C_0 \cdot T \cdot \mathrm{herdisc}(A)$.*

We split the inductive step into the following two parts:

**Norm of rows not in $\overline{B}_t$ will not increase.** We first fix $t \in [T]$. Then by the inductive assumption and our approximated norm computation, we have that for the $t$'th iteration, all rows not in $\overline{B}_t$ have norm at most

$$(1 + \epsilon_0) \cdot C_0 \cdot T \cdot \mathrm{herdisc}(A).$$

Since we have

$$\mathrm{span}(AV_t^\top V_t) \subseteq \mathrm{span}(V),$$

then the norm of each row in matrix $A(I - V_t^\top V_t)$ will never increase after we add new rows to $V$. Thus we have the claim proved.

**At most $m_t$ rows in $\overline{B}_t$ reach the threshold after adding rows.** Now in this paragraph we prove that, after adding

$$\{\widetilde{u}_1, \ldots, \widetilde{u}_{n/(8T)}\}$$

the bottom of $V_t$ (Then we name the new matrix to be $v_{t+1}$), there are no less than $m_t$ rows in $\overline{B}_t(I - V_{t+1}^\top V_{t+1}) \in \mathbb{R}^{m_{t-1} \times n}$ having norm larger than

$$(1 - \epsilon_0) \cdot C_0 \cdot T \cdot \mathrm{herdisc}(A).$$

Here we note that, before adding the new rows to $V_t$, since all rows of $\overline{B}_t$ are already orthogonal to all rows of $V_t$, we have the following

$$\overline{B}_t = \overline{B}_t(I - V_t^\top V_t)^8.$$

We now prove it by contradiction. For the sake of contradiction, we assume that, more than $m_t$ rows in $\overline{B}_t(I - V_{t+1}^\top V_{t+1})$ have norm larger than $C_0 \cdot T \cdot \mathrm{herdisc}(A)$ after adding

$$\{\widetilde{u}_1, \ldots, \widetilde{u}_{n/(8T)}\}$$

to the bottom of $V_t$. We assume that $V_{t+1}$ has $l_{t+1}$ rows. Select an arbitrary set of unit vectors $\{w_1, \ldots, w_{n-l_{t+1}}\} \subseteq \mathbb{R}^n$ such that,

$$\mathrm{span}(w_1, \ldots, w_{n-l_{t+1}}) = \mathbb{R}^d \backslash \mathrm{span}(V_{t+1}).$$

Using row norms, we can generate

$$\overline{B}_t = \overline{D}_t A(I - V_t^\top V_t).$$

Using SUBSAMPLE (Algorithm 10), we can generate a matrix

$$\widetilde{B}_t = \widetilde{D}_t A(I - V_t^\top V_t).$$

By Lemma H.2, we know that $\widetilde{B}_t$ only has roughly $\|\widetilde{D}_t\|_0 = O(\epsilon_B^{-2} n \log(n/\delta_B))$ rows so that

$$\Pr\left[(1 - \epsilon_B)\overline{B}_t^\top \overline{B}_t \preceq \widetilde{B}_t^\top \widetilde{B}_t \preceq (1 + \epsilon_B)\overline{B}_t^\top \overline{B}_t\right] \geq 1 - \delta_B$$

Using Lemma A.3, the above approximation implies that

$$\Pr\left[(1 - \epsilon_B)\lambda_i(\overline{B}_t^\top \overline{B}_t) \preceq \lambda_i(\widetilde{B}_t^\top \widetilde{B}_t) \preceq (1 + \epsilon_B)\lambda_i(\overline{B}_t^\top \overline{B}_t), \forall i \in [n]\right] \geq 1 - \delta_B.$$

We define

$$\overline{Q} := \overline{B}_t(I - V_{t+1}^\top V_{t+1}).$$

And for the approximated metrics, we define

$$\widetilde{Q} := \widetilde{B}_t(I - V_{t+1}^\top V_{t+1}).$$

We first note that, all rows of $\overline{Q}$ lie in $\mathrm{span}(w_1, \ldots, w_{n-l_{t+1}})$. Then by the construction of $\widetilde{B}$, we have that, all rows of $\widetilde{Q}$ also lie in $\mathrm{span}(w_1, \ldots, w_{n-l_{t+1}})$. It follows that

$$\sum_{k \in [m_{t-1}]} \sum_{j \in [n-l_{t+1}]} \langle \overline{q}_k, w_j \rangle^2 = \sum_{k \in [m_{t-1}]} \|\overline{q}_k\|_2^2$$
$$> m_t \cdot ((1 - \epsilon_0) \cdot C_0 \cdot T \cdot \mathrm{herdisc}(A))^2,$$

where the second step follows from the contradiction assumption.

Taking the average over the vectors $w_j \in \mathbb{R}^n$, there has to be a $j \in [n - l_{t+1}]$ such that

$$\sum_{k \in [m_{t-1}]} \langle \overline{q}_k, w_j \rangle^2 > m_t \cdot ((1 - \epsilon_0) \cdot C_0 \cdot T \cdot \mathrm{herdisc}(A))^2 / n.$$

We denote $v = w_j$. Thus we have that,

$$\begin{aligned}
\|\widetilde{B}_t v\|_2 &\geq (1 - \epsilon_B) \cdot \|\overline{B}_t v\|^2 \\
&> (1 - \epsilon_B) \cdot m_t \cdot ((1 - \epsilon_0) \cdot C_0 \cdot T \cdot \mathrm{herdisc}(A))^2 / n,
\end{aligned} \tag{2}$$

where the first step follows from Lemma H.2. and the second step follows from the contradiction assumption that there are more than $m_l$ rows in $\overline{B}_t(I - V_{t+1}^\top V_{t+1})$ with norm more than $(1 - \epsilon_0) \cdot C_0 \cdot T \cdot \mathrm{herdisc}(A)$ and $v$ is a unit vector.

We can upper bound $\|\widetilde{B}_t v\|_2^2$,

$$\begin{aligned}
\|\widetilde{B}_t v\|_2^2 &= v^\top \widetilde{B}_t^\top \widetilde{B}_t v \\
&\leq \widetilde{\sigma}_{n/(8T)} \\
&\leq (1 + \epsilon_B) \cdot \overline{\sigma}_{n/(8T)},
\end{aligned} \tag{3}$$

where the first step follows from the definition of $\ell_2$ norm , the second step follows from that $v$ is orthogonal to $\widetilde{u}_1, \ldots, \widetilde{u}_{n/(8T)} \in \mathbb{R}^n$ and, the third step follows from the way we generate $\widetilde{B}_t$.

Thus we have that,

$$\begin{aligned}
\overline{\sigma}_{n/(8T)} &> \frac{1 - \epsilon_B}{1 + \epsilon_B} \cdot m_t \cdot ((1 - \epsilon_0) \cdot C_0 \cdot T \cdot \mathrm{herdisc}(A))^2 / n \\
&\geq \frac{1}{4} \cdot m_t \cdot (C_0 \cdot T \cdot \mathrm{herdisc}(A))^2 / n,
\end{aligned}$$

where the first step follows from Eq. (2) and Eq. (3), the second step follows from that we set $\epsilon_B \in (0, 0.5)$ and $\epsilon_0 = 0.01$.

But at the same time we have the following,

$$\begin{aligned}
\mathrm{herdisc}(A) &\geq \mathrm{herdisc}(\overline{B}_t) \\
&> \frac{n}{16eT} \cdot \sqrt{\frac{1}{4} \cdot \frac{m_t \cdot (C_0 \cdot T \cdot \mathrm{herdisc}(A))^2 / n}{m_{t-1} \cdot n}} \\
&= 2 \cdot \mathrm{herdisc}(A),
\end{aligned}$$

where the first step follows from that $\overline{B}_t$ is a matrix obtained from $A \in \mathbb{R}^{m \times n}$ by deleting a subset of the rows of $A$, the second step follows from Theorem A.4 and $k = n/(8T)$ here, and the last step follows from simplify the above step.

Thus, we get a contradiction.

By the proof above, we have the claim that after the first for-loop (Line 9 to Line 27 in Algorithm 2), there are at most $m_T = n/8$ rows in $B_T = A(I - V_t^\top V_t)$ having norm larger than

$$C_0 \cdot T \cdot \mathrm{herdisc}(A).$$

Then we do the second for-loop (Line 31 in Algorithm 2). We notice after that, all these rows lie in $\mathrm{Span}(V_t)$. Thus we have the desired guarantee of the algorithm.

And by union bound of the above steps, we have the result holds with probability at least $1 - \delta$. Thus we complete the proof. $\qquad\square$

## D. Discrepancy-Minimization Algorithm

### D.1. Correctness

The goal of this section is to prove Theorem D.1.

**Theorem D.1.** *The algorithm (*FASTHEREDITARYMINIMIZE *in Algorithm 3) takes a matrix* $A \in \mathbb{R}^{m \times n}$ *as input and outpus a coloring* $x \in \{-1, +1\}^n$ *such that*

$$\mathrm{disc}(A, x) = O(\mathrm{herdisc}(A) \cdot \log n \cdot \log^{1.5} m).$$

*holds with probability* $1 - \delta_{\mathsf{final}}$.

*Proof.* We divide the proof into the following two parts.

**Proof of Quality of Answer.** By the condition of the algorithm stops, we have that, there is no $i \in [n]$ such that $|x_i| < 1$. And by Lemma E.2, every time after we call FASTPARTIALCOLORING (Algorithm 4), $\mathrm{herdisc}(A, x)$ changes at most $O(\mathrm{herdisc}(A) \cdot \log^{1.5} m)$. Note that, the FASTPARTIALCOLORING is called $O(\log n)$ times, thus we have the guarantee. Thus we complete the proof.

**Proof of Success Probability.** For the FASTPARTIALCOLORING Algorithm, each time we call it, we have that

- with probability at most $19/20$, we know that it outputs fail.

- with $1/20 - \delta_{\mathsf{final}}^3/n^3$, it outputs a good $x^{\mathrm{new}}$.

- with probability at most $\delta_{\mathsf{final}}^3/n^3$ it outputs a $x^{\mathrm{new}}$ with no guarantee.

As long as we repeat second while loop more than $k = \Theta(\log(n/\delta_{\mathsf{final}}))$ times, then we can promise that with probability $1 - \delta_{\mathsf{final}}^2/n^2$, we obtain a non-final $x^{\mathrm{new}}$.

Then by union bound we have the final succeed probability is

$$1 - \underbrace{(\delta_{\mathsf{final}}^2/n^2) \cdot \log n}_{\text{all } \log n \text{ iterations get non-fail}} - \underbrace{(\delta_{\mathsf{final}}^3/n^3) \cdot k \log n}_{\text{all } V \text{ and } \eta \text{ are good}}$$

$$\geq 1 - \delta_{\mathsf{final}}.$$

$\square$

### D.2. Running Time

The goal of this section is to prove Theorem D.2.

**Theorem D.2.** *For any parameter* $a \in [0, 1]$*, the expected running time of algorithm (*FASTHEREDITARYMINIMIZE *in Algorithm 3) is*[9]

$$\widetilde{O}(\mathrm{nnz}(A) + n^\omega + n^{2+a} + n^{1+\omega(1,1,a)-a}).$$

*Note that* $\omega(\cdot, \cdot, \cdot)$ *function is defined as Definition A.12.*

*Proof.* For the running time of FASTHEREDITARYMINIMIZE, we analyze it as follows.

First we notice that, $|S|$ halves for each iteration of the outer while loop. Thus for each iteration, we can only examine the $x_i$'s for $i \in S$, where $S$ is from the previous iteration. Then, we can maintain $S$ in $O(n)$ time.

By the same argument, Line 6 takes a total time of $O(\mathrm{nnz}(A))$ to generate the submatrices $\overline{A}$.

Recall we have the $S$ halved each iteration, so $|S| \leq n/2^i$ at $i$'th iteration. Thus we have by Lemma E.12, Line 12 takes time

$$\mathcal{T}_{\mathsf{FPC}} = \widetilde{O}(\mathrm{nnz}(A) + n^\omega + n^{2+a} + n^{1+\omega(1,1,a)-a})$$

to call FASTPARTIALCOLORING.

---

[9]Note that $\omega$ is the exponent of matrix multiplication, currently $\omega \approx 2.373$ (Williams, 2012; Alman & Williams, 2021).

**Algorithm 3**

---

1: **procedure** FASTHEREDITARYMINIMIZE($A \in \mathbb{R}^{m \times n}, \delta_{\text{final}} \in (0, 0.001)$)  ▷ Lemma D.1, Lemma D.2
2:     $x \leftarrow \mathbf{0}_n$
3:     **while** $x \notin \{-1, +1\}^n$ **do**
4:         $S \leftarrow \{i \in [n] \mid |x_i| < 1\}$
5:         Let $\overline{x} \leftarrow x_S$ be the coordinates of $x$ indexed by $S$
6:         Let $\overline{A} \leftarrow A_{*,S} \in \mathbb{R}^{m \times |S|}$ be the columns of $A$ indexed by $S$
7:         finished $\leftarrow$ false
8:         counter $\leftarrow 0$
9:         $k \leftarrow \Theta(\log(n/\delta_{\text{final}}))$
10:         **while** finished $=$ false and counter $< k$ **do**
11:             counter $\leftarrow$ counter $+ 1$
12:             $x^{\text{new}} \leftarrow$ FASTPARTIALCOLORING($\overline{A}, \overline{x}, \delta_{\text{final}}^3/n^3$)  ▷ Algorithm 4
13:             **if** $x^{\text{new}} \neq$ fail **then**
14:                 $x_S \leftarrow x^{\text{new}}$
15:                 finished $\leftarrow$ true
16:             **end if**
17:         **end while**
18:         **if** finished $=$ false **then**
19:             **return** fail
20:         **end if**
21:     **end while**
22:     **return** $x$
23: **end procedure**

---

Let $p$ denote the succeed probability of FASTPARTIALCOLORING, each time, it takes $T_{\text{FPC}}$ time, so total

$$\text{Expected time} = p \cdot T_{\text{FPC}} + (1-p) \cdot p \cdot 2T_{\text{FPC}} + (1-p)^2 \cdot p \cdot 3T_{\text{FPC}} + \cdots$$

$$= \sum_{i=1}^{\infty} (1-p)^{i-1} p \cdot i \cdot T_{\text{FPC}}$$

Let $f(p) = \sum_{i=1}^{\infty} (1-p)^{i-1} p \cdot i$. Then we have

$$f(p) - (1-p)f(p) = \sum_{i=1}^{\infty} (1-p)^{i-1} pi - \sum_{i=1}^{\infty} (1-p)^i pi$$

$$= \sum_{i=0}^{\infty} (1-p)^i p(i+1) - \sum_{i=1}^{\infty} (1-p)^i pi$$

$$= p + \sum_{i=1}^{\infty} (1-p)^i p$$

$$\leq p + p \cdot 1/p$$

$$= 1 + p$$

Thus,

$$f(p) \leq \frac{1+p}{p}$$

Thus the expected time is $O(T_{\text{FPC}}/p)$.

Since there are at most $\log n$ iterations, so the expected time is

$$O(p^{-1} T_{\text{FPC}} \cdot \log n).$$

$\square$

---

**Algorithm 4** Input sparsity time partial coloring algorithm

---

1: **procedure** FASTPARTIALCOLORING($A \in \mathbb{R}^{m \times n}, x \in (-1, +1)^n, a \in [0, 1], \delta_{\text{final}} \in (0, 0.01), \delta \in (0, 0.01)$) ▷ Lemma E.2, Lemma E.12
2:      $u_1 \leftarrow \mathbf{0}_n$
3:      $\delta_1 \leftarrow \delta_{\text{final}}^2/n^2$
4:      $\delta_2 \leftarrow \delta_{\text{final}}^2/n^2$
5:      $V \leftarrow$ FASTPROJECTTOSMALLROWS($A, \delta_1$)     ▷ Algorithm 2
6:      $\eta \leftarrow$ FASTAPPROXMAXNORM($A, V, \delta_2$)     ▷ Algorithm 5
7:          ▷ $\eta \leq O(\text{herdisc}(A) \log(m/n))$
8:      $\epsilon \leftarrow \Theta((\log(mn) + n)^{-1/2})$
9:      $N \leftarrow \Theta(\epsilon^{-2} + n)$
10:      $\beta \leftarrow \Theta(\epsilon \eta \sqrt{N \log(m/\delta)})$     ▷ $\beta \leq O(\text{herdisc}(A) \log^{1.5}(m))$
11:      Generate a Gaussian matrix $\mathsf{G} \in \mathbb{R}^{n \times N}$, where each column is sampled from $\mathcal{N}(0, I_n)$
12:      MAINTAIN ds     ▷ Algorithm 6
13:      ds.INIT($V, \mathsf{G}, n, n^a$)     ▷ Algorithm 6
14:      **for** $t = 1 \to N$ **do**
15:          $g_{V_t} \leftarrow$ ds.QUERY()     ▷ Algorithm 6, $g_{V_t} = (I - V_t^\top V_t)\mathsf{G}_{*,t}$
16:          ▷ Let $\mu > 0$ be maximal such that $\max\{\|x + u_t + \mu \cdot g_{V_t}\|_\infty, \|x + u_t - \mu \cdot g_{V_t}\|_\infty\} = 1$
17:          $\mu \leftarrow$ FINDBOUNDARY($x + u_t, g_{V_t}$)     ▷ Algorithm 11
18:          $\widehat{g}_t \leftarrow \min\{\epsilon, \mu\} \cdot g_{V_t}$     ▷ $\widehat{g}_t \in \mathbb{R}^n$
19:          **for** $i = 1 \to n$ **do**
20:             **if** $|x_i + u_{t,i} + \widehat{g}_{t,i}| = 1$ and $|x_i + u_{t,i}| < 1$ **then**
21:               We will add $e_i$ into $V$ in an implicit way via data structure ds     ▷ The explicit way can be found here Algorithm 1
22:               ds.UPDATE($e_i$)     ▷ Algorithm 7
23:             **end if**
24:          **end for**
25:          $u_{t+1} \leftarrow u_t + \widehat{g}_t$
26:          **if** $|\{i \in [n] : |x_i + u_{t+1,i}| = 1\}| \geq n/2$ **then**
27:             **if** $\|Au_{t+1}\|_\infty > \beta$ **then**
28:               **return** fail     ▷ Case 2 of Lemma E.2
29:             **else**
30:               **return** $x^{\text{new}} \leftarrow x + u_{t+1}$     ▷ Case 1 of Lemma E.2
31:             **end if**
32:          **end if**
33:      **end for**
34:      **return** fail     ▷ Case 3 of Lemma E.2
35: **end procedure**

---

# E. Partial Coloring

## E.1. Fast Algorithm for Approximating Norms

---

**Algorithm 5**

---

1: **procedure** FASTAPPROXNORM($A \in \mathbb{R}^{m \times n}, V \in \mathbb{R}^{\ell \times n}, \delta_2 \in (0, 0.01)$)     ▷ Lemma E.1
2:      Let $R \in \mathbb{R}^{n \times r}$ denote a random JL matrix with $r = \Theta(\epsilon_1^{-2} \log(m/\delta_2))$    ▷ Either Definition B.2 or Definition B.1.
3:      Compute $(I - V^\top V)R$     ▷ This takes $O(n^2 r)$ time
4:      $\eta \leftarrow \max_{j \in [m]} \|a_j^\top (I - V^\top V)R\|_2$     ▷ This takes $O(\text{nnz}(A)r)$ time
5:      **return** $\eta$
6: **end procedure**

---

Here in this section, we present an subroutine used as a tool to get fast approximation to the row norm of a matrix, based on

the JL lemma. The goal of this section is to prove Lemma E.1

**Lemma E.1.** *Let $V \in \mathbb{R}^{\ell \times n}$ where $\ell \leq n$ and $A \in \mathbb{R}^{m \times n}$. For any accuracy parameter $\epsilon_1 \in (0, 0.1)$, failure probability $\delta_1 \in (0, 0.1)$, let $r = O(\epsilon_1^{-2} \log(m/\delta_1))$, there is an algorithm (procedure FASTAPPROXMAXNORM in Algorithm 5) that runs in $O((\mathrm{nnz}(A) + n^2) \cdot r)$ time and outputs a number $\eta \in \mathbb{R}$ such that*

$$\eta \geq (1 - \epsilon_1) \cdot \max_{j \in [m]} \|a_j^\top (I - V^\top V)\|_2$$

*holds with probability $1 - \delta_1$.*

*Proof.* Let $R \in \mathbb{R}^{n \times r}$ denote a random JL matrix. The running time is from computing

- Computing $VR$ takes $n^2 r$ time.

- Computing $V^\top(VR)$ takes $n^2 r$ time.

- Computing $AR$ takes $\mathrm{nnz}(A)r$ time.

- Computing $A \cdot (V^\top \cdot (VR))$ takes $\mathrm{nnz}(A)r$ time.

- Computing $\|(AR - AA \cdot (V^\top \cdot (VR)))_{j,*}\|_2$ for all $j \in [m]$. This takes $O(mr)$ time.

- Taking the max over all the $j \in [m]$. This step takes $O(m)$ time.

The correctness follows from JL Lemma (Lemma B.8) directly. □

### E.2. Correctness

The goal of this section is to prove Lemma E.2, which gives the correctness guarantee of the Algorithm 4.

**Lemma E.2** (Output Gurantees of Algorithm 4)**.** *Let $A \in \mathbb{R}^{m \times n}$ be an input matrix and $x \in (-1, 1)^n$ be an input partial coloring. Let $\delta \in (0, 0.01)$ denote a parameter. Conditioning on the event that $V$ is good and $\eta$ is good. We have:*

- **Case 1.** *The Algorithm 4 outputs a vector $x^{\mathrm{new}} \in \mathbb{R}^n$ such that with probability at least $1/10 - \delta$ the following holds,*

  - $\|x^{\mathrm{new}}\|_\infty = 1$;
  - $|\{i \in [n] \mid |x_i^{\mathrm{new}}| = 1\}| \geq n/2$;
  - *Let $a_j^\top$ denote the $j$-th row of matrix $A \in \mathbb{R}^{m \times n}$, for each $j \in [m]$. For all $j \in [m]$, the following holds*

  $$|\langle a_j, x^{\mathrm{new}} \rangle - \langle a_j, x \rangle| \leq O(\mathrm{herdisc}(A) \cdot \log(m) \cdot \log^{1/2}(m/\delta)).$$

- **Case 2.** *With probability at most $\delta$, the Algorithm 4 will output* fail *at Line 28.*

- **Case 3.** *With probability at most $9/10$, the Algorithm 4 will output* fail *at Line 34.*

*Proof.* We start with choosing the parameters:

$$N := O(n + \max\{\log(mn), n\}) = O(n + \log(m)),$$

and

$$\epsilon := O(\min\{\log^{-1/2}(mn), n^{-1/2}\}).$$

**Proof of Case 1.** We first prove that for any iteration $t \in [N]$, no entry of $x + v_t + \widehat{g}_t$ has absolute value larger than 1. Note that in $t$-th iteration, if we find $i$-th entry of $x + v_t + \widehat{g}_t$ reaches absolute value of 1, we add $e_i$ to the matrix $V_t$ (Line 22). Then in the following iterations, after we get the query $g$ (We omit the $t$ here for $g$ and $V$), we have by Lemma F.4 that, $g$ is the distance of original Gaussian vector to the row space of $V$, thus the $i$-th entry of $g$ will be 0. Hence in the following iterations, the entry that we add to $i$-th entry of $v$ will stay 0. Thus we complete the claim.

By the algorithm design, the number entries reach the absolute value 1 keeps increasing, and by the condition check of Line 20, we have that

$$|\{i \in [n] \mid |x_i^{\mathrm{new}}| = 1\}| \geq n/2.$$

Then by Lemma E.7 we have that

$$\forall j \in [m], \ |\langle a_j, g \rangle - \langle a_j, x \rangle| \leq \beta$$

with probability at least $1/10 - \delta$, where we notice that

$$\beta = O(\eta \min\{\log^{-1/2}(mn), n^{-1/2}\}\sqrt{n + \log(m)}\sqrt{\log(m/\delta)}) = O(\eta \log^{1/2}(m)).$$

Thus we have the guarantee.

**Proof of Case 2.** From Lemma E.7 we know that, there is a failure probability at most $\delta$ that,

$$\exists j \in [m], \ |\langle a_j, g \rangle - \langle a_j, x \rangle| > \beta.$$

Thus we have proved the claim.

**Proof of Case 3.** By Lemma E.8, the Algorithm 4 returns fail with probability at most $4/5$.

Thus we complete the proof.

$\square$

Now if we consider the probability of the returning $V$ and $\eta$ are not good, we have the following corollary.

**Corollary E.3.** *Let $A \in \mathbb{R}^{m \times n}$ be an input matrix and $x \in (-1, 1)^n$ be an input partial coloring. Let $\delta_{\mathrm{final}} \in (0, 0.01)$ denote a parameter. We assume $\delta = 1/100$ here in the algorithm. Then we have*

- **Case 1.** *The Algorithm 4 outputs a vector $x^{\mathrm{new}} \in \mathbb{R}^n$ such that with probability at least $1/10 - \delta - \delta_{\mathrm{final}}$ the following holds,*
  - *$\|x^{\mathrm{new}}\|_\infty = 1$;*
  - *$|\{i \in [n] \mid |x_i^{\mathrm{new}}| = 1\}| \geq n/2$;*
  - *Let $a_j^\top$ denote the $j$-th row of matrix $A \in \mathbb{R}^{m \times n}$, for each $j \in [m]$. For all $j \in [m]$, the following holds*

  $$|\langle a_j, x^{\mathrm{new}} \rangle - \langle a_j, x \rangle| \leq O(\mathrm{herdisc}(A) \cdot \log(m) \cdot \log^{1/2}(m)).$$

- **Case 2.** *With probability at most $\delta + \delta_{\mathrm{final}}$, the Algorithm 4 will output* fail *at Line 28.*

- **Case 3.** *With probability at most $9/10 + \delta_{\mathrm{final}}$, the Algorithm 4 will output* fail *at Line 34.*

- **Case 4.** *With probability at most $\delta_{\mathrm{final}}$, the Algorithm 4 will output $x^{\mathrm{new}}$ with no guarantee.*

*Proof.* Recall Lemma E.2 is conditioning on $V$ and $\eta$ are good, here we mention the failure probability of these two operations.

By the construction of Algorithm 4, when we call

$$\textsc{FastProjectoSmallRows}(A, \delta_1)$$

at Line 5, we set $\delta_1 = \delta_{\mathsf{final}}/2$.

Also when we call $\textsc{FastApproxMaxNorm}(A, V, \delta_2)$, we set $\delta_2 = \delta_{\mathsf{final}}/2$.

Thus, we know

$$\Pr[V, \eta \text{ good}] \geq 1 - \delta_{\mathsf{final}}. \tag{4}$$

and we use "$V, \eta$ not good" to denote the complement event of "$V, \eta$ good", then we have

$$\Pr[V, \eta \text{ not good}] \leq \delta_{\mathsf{final}}. \tag{5}$$

**Proof of Case 1.** By Lemma E.2, we have

$$\Pr[\text{case 1 happens} \mid V, \eta \text{ good}] \geq 1/10 - \delta. \tag{6}$$

By basic probability rule,

$$
\begin{aligned}
\Pr[\text{case 1 happens}] =\ & \Pr[\text{case 1 happens} \mid V, \eta \text{ good}] \cdot \Pr[V, \eta \text{ good}] \\
& + \Pr[\text{case 1 happens} \mid V, \eta \text{ not good}] \cdot \Pr[V, \eta \text{ not good}] \\
\geq\ & \Pr[\text{case 1 happens} \mid V, \eta \text{ good}] \cdot \Pr[V, \eta \text{ good}] \\
\geq\ & (1/10 - \delta) \cdot \Pr[V, \eta \text{ good}] \\
\geq\ & (1/10 - \delta) \cdot (1 - \delta_{\mathsf{final}}) \\
\geq\ & 1/10 - \delta - \delta_{\mathsf{final}}.
\end{aligned}
$$

where the second step follows from $\Pr[] \geq 0$, the third step follows from Eq. (6), the forth step follows from Eq. (4).

**Proof of Case 2.** Then by Lemma E.2, we have the case happens with probability at most $\delta$ when conditioning on $V, \eta$ are good, i.e.,

$$\Pr[\text{case 2 happens} \mid V, \eta \text{ good}] \leq \delta. \tag{7}$$

By basic probability rule,

$$
\begin{aligned}
& \Pr[\text{case 2 happens}] \\
=\ & \Pr[\text{case 2 happens} \mid V, \eta \text{ good}] \cdot \Pr[V, \eta \text{ good}] \\
& + \Pr[\text{case 2 happens} \mid V, \eta \text{ not good}] \cdot \Pr[V, \eta \text{ not good}] \\
\leq\ & \Pr[\text{case 2 happens} \mid V, \eta \text{ good}] + \Pr[V, \eta \text{ not good}] \\
\leq\ & \delta + \Pr[V, \eta \text{ not good}] \\
\leq\ & \delta + \delta_{\mathsf{final}}
\end{aligned}
$$

where the second step follows from $\Pr[] \leq 1$, the third step follows from Eq. (7), and the forth step follows from Eq. (5).

**Proof of Case 3.** By Lemma E.2, we have the case happens with probability at most $9/10$ conditioning on $V$ and $\eta$ is good, i.e.,

$$\Pr[\text{case 3 happens} \mid V, \eta \text{ good}] \leq 9/10. \tag{8}$$

By basic probability rule,

$$
\begin{aligned}
\Pr[\text{case 3 happens}] =\ & \Pr[\text{case 3 happens} \mid V, \eta \text{ good}] \cdot \Pr[V, \eta \text{ good}] \\
& + \Pr[\text{case 3 happens} \mid V, \eta \text{ not good}] \cdot \Pr[V, \eta \text{ not good}] \\
\leq\ & \Pr[\text{case 3 happens} \mid V, \eta \text{ good}] + \Pr[V, \eta \text{ not good}] \\
\leq\ & 9/10 + \Pr[V, \eta \text{ not good}] \\
\leq\ & 9/10 + \delta_{\mathsf{final}}.
\end{aligned}
$$

where the second step follows from $\Pr[] \leq 1$, the third step follows from Eq. (8), and the last step follows from Eq. (5).

**Proof of Case 4.** This happens when at least one of the returning $V$ and $\eta$ is not good, and the algorithm returns $x^{\text{new}}$ at Line 30. By union bound we have this happens with probability at most

$$\Pr[\text{case 4 happens}] \leq \delta_{\text{final}}.$$

$\square$

### E.3. Iterative Notations

Here in this section, we introduce some notations to be used in later proofs. We first define the matrix $V_t$ we maintained as follows.

**Definition E.4** (Matrix $V$ of each iteration). *For all $t \in [T]$, we define $V_t$ to be the matrix forming an orthonormal basis which implicitly maintained in the* MAINTAIN *data-structure at the start of $t$-th iteration. For $t = 1$, we define $V_1$ to be generated from* FASTPROJECTTOSMALLROWS *at Line 5 in Algorithm 4.*

We then define the Gaussian random vectors related to the above $V_t$.

**Definition E.5** (Random Gaussian vectors). *For all $t \in [T]$, in the $t$-th iteration, We define $\mathsf{g}_t \in \mathbb{R}^n$ to be a random Gaussian sampled from $\mathcal{N}(0,1)$. We define vector $g_{V_t}$ to be generated by query the* MAINTAIN *data-structure at Line 15 in Algorithm 4. By Lemma F.4, we have*

$$g_{V_t} = (I - V_t^\top V_t) \cdot \mathsf{g}_t.$$

*And we define $\widehat{g}_t$ to be the vector rescaled from $g_{V_t}$ at Line 18 in Algorithm 4, that is,*

$$\widehat{g}_t := \min\{\epsilon, \mu\} \cdot g_{V_t}.$$

Through the whole Algorithm 4, we iteratively maintain a vector $u \in \mathbb{R}^n$ to accumulate the random Gaussian vectors of each iteration. We formally define it here as follows,

**Definition E.6** (Accumulated maintained vector). *For all $t \in [T]$, we define the accumulated maintained vector $u_t \in \mathbb{R}^n$ to be as follows*

$$u_{t+1} := u_t + \widehat{g}_t.$$

*For the case that $t = 1$, we define $u_1 = \mathbf{0}_n$.*

### E.4. Upper Bound for the Inner Products

Here in this section, the goal is to prove Lemma E.7, which gives the upper bound of the inner products of the output vector with the rows of input matrix.

**Lemma E.7** (A variation of Lemma 4 in (Larsen, 2023)). *Let $\beta = \Theta(\epsilon\eta\sqrt{N\log(m/\delta)})$. For any input matrix $A \in \mathbb{R}^{m \times n}$ and any vector $x \in (-1,1)^n$, let $a_i^\top$ denote the $i$-th row of $A$. Let $u_{N+1} \in \mathbb{R}^n$ denote the vector $v \in \mathbb{R}^n$ in the last iteration of Algorithm 4. Then we have that,*

$$\Pr[\forall i \in [m], |\langle a_i, u_{N+1}\rangle| < \beta] \geq 1 - \delta.$$

*Proof.* To prove the lemma, we first fix $i \in [m]$ here. Then for $t$-th iteration of the outer for-loop (Line 14 in Algorithm 4).

We let $g_{V_t}$ be defined as Definition E.5. For simplicity, sometimes we use $g_t$ to denote $g_{V_t}$ if it is clear from text.

For the $t$'s that Algorithm 4 returns fail before the $t$-th iteration, we define $g_t = 0 \in \mathbb{R}^n$. We also define a cumulative vector $u_t := \sum_{i=1}^{t} g_i$. Then we have that

$$\langle a_i, u_t \rangle = \sum_{j=1}^{t} \langle a_i, g_j \rangle.$$

Now we condition on the previous $t-1$ iterations and look into the $t$-th iteration, we generate a vector $\mathbf{g}_t \in \mathbb{R}^d$ such that every entry is drawn i.i.d. from $\mathcal{N}(0,1)$. Then we have for any $\xi > 0$, we have

$$\Pr[|\langle a_i, (I - V_t^\top V_t)\mathbf{g}\rangle| > \xi] \geq \Pr[|\min\{\epsilon, \mu\} \cdot \langle a_i, (I - V_t^\top V_t)\mathbf{g}\rangle| > \xi]$$
$$\geq \Pr[|\langle a_i, g_t\rangle| > \xi],$$

where the first step follows from $\min\{\epsilon, \mu\} \leq 1$, the second step follows from Lemma F.4. We also have the following

$$\epsilon \cdot \langle a_i, (I - V_t^\top V_t)\mathbf{g}\rangle \sim \mathcal{N}(0, \epsilon^2 \|a_i(I - V_t^\top V_t)\|_2^2)$$

by the linearity of the distribution.

We can bound the variance in the following sense,

$$\epsilon^2 \|a_i(I - V_t^\top V_t)\|_2^2 \leq \epsilon^2 \eta^2$$

by the definition of $\eta$.

Let $\beta > 0$ denote some parameter. Thus using Claim A.5, we have the following

$$\Pr[|\langle a_i, g_t\rangle| > \beta] \leq \Pr[|\epsilon \cdot \langle a_i, (I - V_t^\top V_t)\mathbf{g}\rangle| > \beta]$$
$$\leq 2\exp(-(\epsilon\eta)^{-2}\beta^2/2)$$

Finally, since the $g_t$ is independent from $u_i$'s for $i \in [t-1]$, we have that

$$\mathbb{E}[\langle a_i, g_t\rangle \mid u_{t-1}, \ldots, u_1] = 0.$$

This follows by the fact that we define $\mu$ in Line 17 to be

$$\mu := \arg\max_{\mu \geq 0}\{\|x + u_t + \mu g_t\|_\infty, \|x + u_t - \mu g_t\|_\infty\} = 1.$$

Thus we have that, the sequence $\langle a_i, g_1\rangle, \ldots, \langle a_i, g_t\rangle$ becomes a martingale difference sequence (Definition A.7) with respect to $u_1, \ldots, u_t$. Then by Theorem A.8 we have that with probability at least $1 - \delta$, the following holds

$$|\langle a_i, u_{N+1}\rangle| = |\sum_{t=1}^N \langle a_i, g_t\rangle|$$
$$\leq 2\epsilon\eta\sqrt{112N\lg(1/\delta)}$$
$$\leq 100\epsilon\eta\sqrt{N\lg(1/\delta)},$$

where we set the parameters in Theorem A.8 to be

$$b = 2 \text{ and } c \leq (\epsilon\eta)^{-2}/2.$$

Then we set $\delta = \delta/m$ and let $\beta = 100\epsilon\eta\sqrt{N\log(m/\delta)}$, we have

$$\Pr[|\langle a_i, u_{N+1}\rangle| \leq \beta] \geq 1 - \delta/m.$$

The above implies that

$$\Pr[|\langle a_i, u_{N+1}\rangle| > \beta] \leq \delta/m$$

Applying a union bound over all the $i \in [m]$, then we complete the proof.

$\square$

### E.5. Upper Bound for the Failure Probability

Here in this section, the goal is to prove Lemma E.8, which gives the upper bound of the probability that the algorithm returns fail.

**Lemma E.8** (A variation of Lemma 5 in (Larsen, 2023)). *For any input matrix $A \in \mathbb{R}^{m \times n}$ and any vector $x \in [0, 1]^n$, the probability that Algorithm 4 returns* fail *at Line 34 is at most* $4/5$.

*Proof.* The basic idea of the proof is that, after many iterations ($N$ in Algorithm 4), we have the expectation of $\|x + u\|_2^2$ large enough. Hence lots of entries of $x + u$ will reach the absolute value of 1. We denote the number of $e_i$'s added to $V_t$ in Line 22 through the running of Algorithm 4 to be $R$.

Recall we define $g_t$ to be generated in Line 15 of Algorithm 4 (Definition E.5). We notice that, we add $\epsilon \cdot g_t$ or $\mu \cdot g_t$ to $u_t$ in Line 25. Here in this paragraph, we first assume that, we add $\mu \cdot g_t$, i.e, $\mu < \epsilon$. Recall we generate $\mu$ in Line 17 of Algorithm 4 to be

$$\mu := \arg\max_{\mu > 0} \{\max\{\|x + u_t + \mu \cdot g_t\|_\infty, \|x + u_t - \mu \cdot g_t\|_\infty\} = 1\}.$$

Thus we have that in Line 20 of Algorithm 4, there must be at least one entry $i \in [n]$ satisfying that

$$|x_i + u_{t,i} + g_{t,i}| = 1 \text{ and } |x_i + u_{t,i}| < 1.$$

We define $\rho_t$ to denote the probability such that the algorithm never return fail before reaching $t$-th iteration and $\mu < \epsilon$ in this iteration. For any $i \in [n]$ and $t \in [N]$, we define the following event by $E_{i,t}$: the index $i$ satisfies

$$|x_i + u_{t,i} + \mu g_{t,i}| = 1 \text{ and } |x_i + u_{t,i}| < 1$$

in Line 20 of the $t$-th iteration.

By the symmetry of the vector $\mathbf{g}_t$ (Since $\mathbf{g}_t$ is Gaussian, $\mathbf{g}_t$ and $-\mathbf{g}_t$ are equally likely conditioned on $V_t$), we have that

$$\{i \in [n] \mid E_{i,t} \text{ holds}\} \geq \rho_t/2.$$

We denote the number of $e_i$'s added to $V_t$ in Line 22 through the running of Algorithm 4 to be $R$. And we denote the number of rows added to $V_t$ by $r_t$. By the analysis above, we have that

$$\mathbb{E}[R] = \mathbb{E}[\sum_{t=1}^{N} r_t] \geq \sum_{t=1}^{N} \rho_t/2.$$

By the construction of our algorithm, no more than $n$ $e_i$'s will be added to $V$ through the algorithm running, thus we have

$$\sum_{t=1}^{N} \rho_t \leq 2 \mathbb{E}[R] \leq 2n.$$

Now for each iteration $t \in [N]$, we define $g_t$ to be the vector added to $u_t$ at Line 25 of Algorithm 4. Here we note that after $N$ iterations,

$$u_{N+1} = \sum_{t=1}^{N} g_t.$$

And this is the final vector output by the algorithm (if it doesn't return fail).

We have that:

$$\mathbb{E}[\|x + u_{N+1}\|_2^2] = \mathbb{E}[\|x + \sum_{t=1}^{N} g_t\|_2^2]$$

$$= \|x\|_2^2 + \sum_{i=1}^{N} \mathbb{E}[\|g_t\|_2^2]. \tag{9}$$

where the first step follows from the definition of $v$, and the second step follows from $\mathbb{E}[g_t | g_1, \ldots, g_{t-1}] = 0$.

Recall we define the $V_t$ to be the matrix $V$ at the start of $t$-th iteration[10] (Definition E.4). We denote the original Gaussian before projection by $g_t$ (Definition E.5).

By Claim E.9, we have

$$\mathbb{E}[\|g_t\|_2^2] \geq (1 - \delta) \cdot \epsilon^2 (n - n/4 - \mathbb{E}[R]) - \epsilon^2 \sqrt{2\rho_t} n.$$

By Claim E.10 we have the lower bound of the expectation of $R$,

$$\mathbb{E}[R] \geq 0.63n.$$

Using Claim E.11, we obtain

$$\Pr[R \geq n/2] > 0.2.$$

When $R \geq n/2$, Algorithm 4 must terminate and return the $x^{\text{new}}$ in Line 30. Thus we have that, the probability of terminating and returning fail in Line 34 is at most $4/5$.

Thus we complete the proof. $\qquad\qquad\square$

### E.6. Lower Bound for the Vector after Projection

Here in this section, we prove the following claim, giving the lower bound of the projected vector $g_t$.

**Claim E.9** (Lower Bound for the projection of projected vector $g_t$)**.** *We define $R$ to be the total number of $e_i$'s added to $V_t$ in Line 22 through the running of Algorithm 4. We define $\rho_t$ to denote the probability such that the algorithm never return* fail *before reaching $t$-th iteration and $\mu < \epsilon$ in this iteration. Then we have that*

$$\mathbb{E}[\|g_t\|_2^2] \geq (1 - \delta) \cdot \epsilon^2 (n - n/4 - \mathbb{E}[R]) - \epsilon^2 \sqrt{2\rho_t} n.$$

*Proof.* For any $t \in [N]$, we define the indicators $F_t$ to be

$$F_t = \mathbf{1}\{\forall i \in [m] \mid |\langle a_i, u_N \rangle| < \beta\},$$

and $Y_t$ to be

$$Y_t = \mathbf{1}\{\mu < \epsilon\}.$$

Then we have that

$$
\begin{aligned}
\mathbb{E}[\|g_t\|_2^2] &= \mathbb{E}[F_t Y_t \mu^2 \cdot \|(I - V_t^\top V_t) \cdot g_t\|_2^2 + F_t(1 - Y_t)\epsilon^2 \cdot \|(I - V_t^\top V_t) \cdot g_t\|_2^2] \\
&\geq \mathbb{E}[F_t(1 - Y_t)\epsilon^2 \cdot \|(I - V_t^\top V_t) \cdot g_t\|_2^2] \\
&\geq \epsilon^2 \cdot \mathbb{E}[F_t \|(I - V_t^\top V_t) \cdot g_t\|_2^2] - \epsilon^2 \cdot \mathbb{E}[Y_t \|(I - V_t^\top V_t) \cdot g_t\|_2^2].
\end{aligned}
$$

where the first step follows from the definition of $F_t$ and $G_t$, the second step follows from $F_t Y_t \mu^2 \|(I - V_t^\top V_t) \cdot g_t\|_2^2 \leq 0$, and the last step follows from splitting the terms.

Using Cauchy-Schwartz inequality, we have that

$$\mathbb{E}[Y_t \cdot \|(I - V_t^\top V_t) \cdot g_t\|_2^2] \leq \sqrt{\mathbb{E}[Y_t^2] \cdot \mathbb{E}[\|(I - V_t^\top V_t) \cdot g_t\|_2^4]}.$$

---

[10]Note that, in the design of our algortithm, we maintain this matrix $V$ implicitly in the MAINTAIN data structure.

Recall we define $Y_t$ to be an indicator, thus we have that $\mathbb{E}[Y_t^2] = \mathbb{E}[Y_t]$. Since $(I - V_t^\top V_t)$ is a projection matrix, we have

$$\|g_t\|_2 \geq \|(I - V_t^\top V_t) \cdot g_t\|_2. \tag{10}$$

Thus, the following holds

$$\mathbb{E}[Y_t \cdot \|(I - V_t^\top V_t) \cdot g_t\|_2^2] \leq \sqrt{\mathbb{E}[Y_t] \cdot \mathbb{E}[\|g_t\|_2^4]}$$
$$= \sqrt{\rho_t \cdot \mathbb{E}[\|g_t\|_2^4]}.$$

where the first step follows from Eq. (10), the second step follows from the definition of $\rho_t$.

For any $i \in [n]$, we denote the $i$-th entry of $g_t$ by $g_{t,i}$. Note that $g_{t,i} \sim \mathcal{N}(0,1)$ for all $i \in [n]$ and $t \in [N]$. And all $g_{t,i}$'s are i.i.d. Thus we have that

$$\mathbb{E}[\|g_t\|_2^4] = \mathbb{E}[(\sum_{i=1}^n g_{t,i}^2)^2]$$
$$\leq \sum_{i=1}^n \sum_{j=1}^n \mathbb{E}[g_{t,i}^2 g_{t,j}^2].$$

where the first step follows from the definition of $g_t$, the second step follows from that $g_{t,i}$'s are i.i.d. For all the terms that $i \neq j$, we have that

$$\mathbb{E}[g_{t,i}^2 g_{t,j}^2] = \mathbb{E}[g_{t,i}^2] \cdot \mathbb{E}[g_{t,j}^2] = 1.$$

For the terms that $i = j$, by the 4-th moment of the normal distribution, we have

$$\mathbb{E}[g_{t,i}^4] = 3.$$

Thus we have

$$\mathbb{E}[\|g_t\|_2^4] \leq n(n-1) + 3n \leq 2n^2.$$

Hence

$$\mathbb{E}[Y_t \cdot \|(I - V_t^\top V_t) \cdot g_t\|_2^2] \leq \sqrt{2\rho_t} n.$$

Also we have that

$$\mathbb{E}[F_t \cdot \|(I - V_t^\top V_t) \cdot g_t\|_2^2] = \Pr[F_t = 1] \cdot \mathbb{E}[\|(I - V_t^\top V_t) \cdot g_t\|_2^2]$$
$$= \Pr[F_t = 1] \cdot (\mathbb{E}[n - \dim(V_t)])$$
$$= \Pr[F_t = 1] \cdot (n - \mathbb{E}[\dim(V_t)]).$$

where the first step follows from the definition of expectation, the second step follows from the property of the vector $(I - V_t^\top V_t) \cdot g_t$ (Since g is Gaussian), and the last step follows from the definition of expectation.

Note that for any $t \in [N]$, we have that

$$\dim(V_t) \leq R + n/4$$

Thus we have

$$\mathbb{E}[\dim(V_t)] \leq \mathbb{E}[R] + n/4.$$

Hence we conclude that

$$\mathbb{E}[\|g_t\|_2^2] \geq \Pr[F_t = 1] \cdot \epsilon^2(n - n/4 - \mathbb{E}[R]) - \epsilon^2\sqrt{2\rho_t} n.$$

We have by Lemma E.7 that

$$\Pr[F_t = 1] \geq 1 - \delta.$$

Hence we have

$$\mathbb{E}[\|g_t\|_2^2] \geq (1 - \delta) \cdot \epsilon^2(n - n/4 - \mathbb{E}[R]) - \epsilon^2\sqrt{2\rho_t} n.$$

$\square$

### E.7. Lower Bound on Expectation

Here we prove the following claim, which gives the lower bound of expectation of the unit vectors added to $V$ through the algorithm running.

**Claim E.10.** *We denote the number of $e_i$'s added to $V_t$ in Line 22 through the running of Algorithm 4 to be R. We define $\rho_t$ to denote that Algorithm 4 never return* fail *before $i$-th iteration and $\mu < \epsilon$ in this iteration. We assume $\rho_t \leq 0.005$ and $\delta \leq 0.01$.*

*If*

$$\mathbb{E}[\|g_t\|_2^2] \geq (1 - \delta) \cdot \epsilon^2(n - n/4 - \mathbb{E}[R]) - \epsilon^2\sqrt{2\rho_t}n.$$

*Then we have that*

$$\mathbb{E}[R] \geq 0.63n.$$

*Proof.* By Assumption this claim, we have

$$\mathbb{E}[\|g_t\|_2^2] \geq (1 - \delta) \cdot \epsilon^2(n - n/4 - \mathbb{E}[R]) - \epsilon^2\sqrt{2\rho_t}n.$$

Note that we assume that $\rho_t \leq 0.005$ and $\delta \leq 0.01$. Then by the above step we have that

$$\begin{aligned}
&\mathbb{E}[\|g_t\|_2^2] \\
&\geq 0.99 \cdot \epsilon^2(n - n/4 - \mathbb{E}[R] - 0.102n) \\
&= 0.99 \cdot \epsilon^2(0.698n - \mathbb{E}[R]).
\end{aligned}$$

Note here that if $\rho_t > 0.005$, we have $\mathbb{E}[\|g_t\|_2^2]$ is lower bounded by 0. Recall that we have $\sum_{t=1}^{N} \rho_t \leq 2n$, thus the number of iterations that $\rho_t \geq 0.005$ is at most $400n$.

Now considering the rest $N - 400n$ iterations, it also holds that

$$\mathbb{E}[\|g_t\|_2^2] \geq 0.99 \cdot \epsilon^2(0.698n - \mathbb{E}[R]). \tag{11}$$

Hence we have

$$\begin{aligned}
\mathbb{E}[\|x + u_{N+1}\|_2^2] &\geq \|x\|_2^2 + (N - 400n) \cdot 0.99 \cdot \epsilon^2(0.698n - \mathbb{E}[R]) \\
&\geq (N - 400n) \cdot 0.99 \cdot \epsilon^2(0.698n - \mathbb{E}[R]) \\
&= 16 \cdot 0.99 \cdot (0.698n - \mathbb{E}[R]),
\end{aligned}$$

where the first step follows from Eq. (9) and Eq. (11), the second step follows from $\|x\|_2^2 \leq 0$, and the last step follows by setting $N = 16\epsilon^{-2} + 400n$.

Notice now that Algorithm 4 will only produce an output $v$ such that $\|x + u_{N+1}\|_\infty \leq 1$. Thus we conclude it happens when $\mathbb{E}[\|x + u_{N+1}\|_2^2] \leq n$. Thus we have

$$n \geq 16 \cdot 0.99 \cdot (0.698n - \mathbb{E}[R]).$$

By the above inequality, it holds that

$$\mathbb{E}[R] \geq (0.698 - 1.02(1/16))n > 0.634n.$$

$\square$

### E.8. From Expectation to Probability

**Claim E.11.** *If $\mathbb{E}[R] \geq 0.62n$ then*

$$\Pr[R \geq n/2] > 0.2.$$

*Proof.* Here we define $Z := n - R$. Then we have the expectation of $Z$

$$\mathbb{E}[Z] = n - \mathbb{E}[R] \leq 0.38n.$$

where the first step follows from the definition of $Z$, the second step follows from $\mathbb{E}[R] \geq 0.62n$.

Then by Markov's inequality we have that

$$\Pr[Z > a] \leq \frac{\mathbb{E}[Z]}{a}.$$

Thus we have

$$\Pr[Z > n/2] < 2 \cdot 0.38$$
$$= 0.76$$
$$< 0.8,$$

which implies that,

$$\Pr[Z \leq n/2] > 0.2.$$

Finally we have by $Z := n - R$ that

$$\Pr[R \geq n/2] > 0.2.$$

$\square$

### E.9. Running Time

The goal of this section is to prove Lemma E.12.

**Lemma E.12.** *For any parameter $a \in [0, 1]$, the algorithm (FASTPARTIALCOLORING in Algorithm 4) runs in*

$$\widetilde{O}(\mathrm{nnz}(A) + n^{\omega} + n^{2+a} + n^{1+\omega(1,1,a)-a})$$

*time. Note that $\omega$ is the exponent of matrix multiplication.*

*Proof.* The running time of FASTPARTIALCOLORING can be divided as follows,

- Line 5 takes time $\widetilde{O}(\mathrm{nnz}(A) + n^{\omega})$, by Lemma C.2.

- Line 6 takes time $O(\mathrm{nnz}(A) + n^2)$, by Lemma E.1.

- Line 13 takes time $O(n^{\omega} + n^2)$ to initialize the data structure, by Lemma F.7.

- Run the following for $N = O(\epsilon^{-2} + n) = O(n + \log m)$ times,

  - Line 15 takes time $O(n^{1+a})$ to query $g$, by Lemma F.7).
  - Line 17 takes time $O(n)$ to find $\mu$ due to Lemma I.1 (via Algorithm 11).
  - Line 22 takes time $O(n^{1+a})$ to update the data structure, by Lemma F.7.

- Over the entire, algorithm we will enter Line 28 at most once. To check the satisfied condition for that only takes $\mathrm{nnz}(A)$ time.

Adding them together and by the tighter time for operations of FASTMAINTAIN data structure (Lemma F.10), we have the total running time of FASTPARTIALCOLORING of

$$\mathcal{T}_{\text{FASTPARTIALCOLORING}} = \widetilde{O}(\mathrm{nnz}(A) + n^{\omega} + n^{2+a} + n^{1+\omega(1,1,a)-a}).$$

Therefore, we complete the proof. $\square$

# F. Fast Maintaining Lazy Data Structure

In this section, we provide a faster maintaining data structure with the idea of lazy update for the Partial Coloring algorithm. In Section F.1 we provide main theorem of the data structure. In Section F.2, Section F.3, Section F.4, Section F.5 we give analysis for the INITIALIZATION, QUERY, UPDATE, RESTART procedures respectively. In Section F.6 we provide the running time analysis. In Section F.7 we give the correctness analysis. In Section F.8 we give a tighter running time analysis.

---

**Algorithm 6**

---

1: **data structure** MAINTAIN          ▷ Theorem F.1
2: **members**
3:      $P \in \mathbb{R}^{n \times n}$
4:      $V \in \mathbb{R}^{n \times n}$
5:      $\mathsf{G} \in \mathbb{R}^{n \times N}$
6:      $\ell$
7:      $\tau_q$          ▷ total counter for query
8:      $k_u$          ▷ $k_u$ is a counter for update process
9:      $k_q$          ▷ $k_q$ is a counter for query process
10:      $w_1, \cdots, w_K \in \mathbb{R}^n$          ▷ This is a list
11: **end members**
12:
13: **public:**
14: **procedure** INIT($V \in \mathbb{R}^{\ell \times n}, \mathsf{G} \in \mathbb{R}^{n \times N}, K$)          ▷ Lemma F.2
15:      $k_u \leftarrow 0, k_q \leftarrow 0$
16:      $\tau_q \leftarrow 0$
17:      $P \leftarrow V^\top V$          ▷ This takes $\mathcal{T}_{\mathrm{mat}}(n, n, n)$
18:      Let $S = \{1, 2, \cdots, K\}$
19:      Let $G \in \mathbb{R}^{n \times N}$ denote $\mathsf{G}_{*,S}$
20:      Compute $\widetilde{G} = P \cdot \mathsf{G}_{*,S}$ and let $\widetilde{g}_1, \cdots, \widetilde{g}_K$ denote those new $K$ vectors          ▷ This takes $\mathcal{T}_{\mathrm{mat}}(K, n, n)$
21:      **for** $i = 1 \to K$ **do**
22:          $w_i \leftarrow \mathbf{0}_n$
23:      **end for**
24: **end procedure**
25:
26: **public:**
27: **procedure** QUERY()          ▷ Lemma F.3, F.4
28:      $k_q \leftarrow k_q + 1$
29:      $\tau_q \leftarrow \tau_q + 1$
30:      **if** $k_q \geq K$ **then**
31:          RESTART()
32:      **end if**
33:      **return** $g_{k_q} - \widetilde{g}_{k_q} - \sum_{i=1}^{k_u} w_i w_i^\top g_{k_q}$          ▷ This takes $O(nK)$ time
34: **end procedure**
35: **data structure**

---

## F.1. Main Data Structure

The goal of this section is to prove Theorem F.1.

**Theorem F.1.** *For a input matrix $V \in \mathbb{R}^{\ell \times n}$ with $\ell \leq n$, a matrix $\mathsf{G} \in \mathbb{R}^{n \times N}$ and a integer $K > 0$, $N = O(n)$, there exists a data structure* MAINTAIN*(Algorithm 6, 7) uses $O(n^2)$ space and supports the following operations:*

- INIT($V \in \mathbb{R}^{\ell \times n}, \mathsf{G} \in \mathbb{R}^{n \times N}, K \in \mathbb{N}_+$): *It initializes the projection matrix $P = V^\top V \in \mathbb{R}^{n \times n}$. It stores the matrix $\mathsf{G}$. For the first $K$ column vectors in $\mathsf{G}$, it computes their projection when apply $P$. It sets counter $k_q$ and $k_u$ to be zero. It*

---

**Algorithm 7** Update

---

1: **data structure** MAINTAIN      ▷ Theorem F.1
2: **public:**
3: **procedure** UPDATE($u \in \mathbb{R}^n$)      ▷ Lemma F.5
4:      ▷ The input vector $u$ is 1-sparse and the nonzero entry is 1
5:      $k_u \leftarrow k_u + 1$
6:      $\widetilde{w} \leftarrow ((I - P) - \sum_{i=1}^{k_u-1} w_i w_i^\top)u$      ▷ This takes $O(nK)$
7:      $w_{k_u} \leftarrow \widetilde{w}/\|\widetilde{w}\|_2$
8:      **if** $k_u \geq K$ **then**
9:          RESTART()
10:      **end if**
11: **end procedure**
12:
13: **private:**
14: **procedure** RESTART()      ▷ Lemma F.6
15:      $P \leftarrow P + \sum_{i=1}^{K} w_i w_i^\top$      ▷ This takes $\mathcal{T}_{\mathrm{mat}}(n, K, n)$
16:      Let $S = \{1 + \tau_q, 2 + \tau_q, \cdots, K + \tau_q\}$
17:      Let $G \in \mathbb{R}^{n \times K}$ denote $\mathsf{G}_{*,S}$
18:      Compute $\widetilde{G} = P \cdot G$ and let $\widetilde{g}_1, \cdots, \widetilde{g}_K$ denote those new $K$ vectors      ▷ This takes $\mathcal{T}_{\mathrm{mat}}(K, n, n)$
19:      $k_u \leftarrow 0, k_q \leftarrow 0$
20:      **for** $i = 1 \rightarrow K$ **do**
21:          $w_i \leftarrow \mathbf{0}_n$      ▷ Reset $w_i$ to an all zero vector
22:      **end for**
23: **end procedure**
24: **end data structure**

---

*sets $\tau_q$ to be zero. This procedure runs in time*

$$\mathcal{T}_{\mathrm{mat}}(n, n, n) + \mathcal{T}_{\mathrm{mat}}(K, n, n).$$

- QUERY()*: It output $(I - V^\top V)\mathsf{g}$ where $\mathsf{g} \in \mathbb{R}^n$ is the next column vector in $\mathsf{G}$ to be used. The running time of this procedure is*

  - *$O(nK)$ time, if $k_q \in [K-1]$*
  - *$\mathcal{T}_{\mathrm{mat}}(n, k, n)$ time, if $k_u = K$*

- UPDATE($u \in \mathbb{R}^n$)*: It takes an 1-sparse vector $u \in \mathbb{R}^n$ as input and maintains $V$ by adding $w$ into next row of $V$ (according to Algorithm 1, $w = \widetilde{w}/\|\widetilde{w}\|_2$ where $\widetilde{w} = (I - V^\top V)u$. It is obvious that $w \perp V$). The running time of this procedure is*

  - *$O(nK)$ time, if $k_u \in [K-1]$;*
  - *$\mathcal{T}_{\mathrm{mat}}(n, k, n)$ time, if $k_u = K$.*

- RESTART()*: It updates the projection $P = V^\top V$, generates $K$ fresh Gaussian vectors by switching the batch of Gaussian vectors we use by $\tau_q$, and compute their projections when apply $P$. It reset the counter $k_q$ and $k_u$ to be zero. The running time of this procedure*

$$\mathcal{T}_{\mathrm{mat}}(n, K, n).$$

*Proof.* It follows from combining Lemma F.2, Lemma F.3, Lemma F.4, Lemma F.5 and Lemma F.6.

$\square$

### F.2. Initialization

The goal of this section is to prove Lemma F.2.

**Lemma F.2** (Running time for INIT). *The procedure* INIT($V \in \mathbb{R}^{\ell \times n}, \mathsf{G} \in \mathbb{R}^{n \times N}, K \in \mathbb{N}_+$) *initializes the projection matrix* $P = V^\top V \in \mathbb{R}^{n \times n}$. *It stores the matrix* $\mathsf{G}$. *For the first $K$ column vectors in $\mathsf{G}$, it computes their projection when apply $P$. It sets counter $k_q$ and $k_u$ to be zero. It sets $\tau_q$ to be zero. This procedure runs in time*

$$\mathcal{T}_{\mathrm{mat}}(n, n, n) + \mathcal{T}_{\mathrm{mat}}(K, n, n).$$

*Proof.* The running time of INIT can be divided into the following lines,

- Line 17 takes time $\mathcal{T}_{\mathrm{mat}}(n, n, n)$ to compute matrix $P$.

- Line 17 takes time $O(nK)$ to generate matrix $G$.

- Line 20 takes time $\mathcal{T}_{\mathrm{mat}}(K, n, n)$ to compute matrix $\widetilde{G}$.

Taking these together, we have the total running time is

$$\mathcal{T}_{\mathrm{mat}}(n, n, n) + \mathcal{T}_{\mathrm{mat}}(K, n, n).$$

Thus, we complete the proof. □

### F.3. Query

The goal of this section is to prove Lemma F.3 and Lemma F.4.

**Lemma F.3** (Running time for QUERY). *The running time of procedure* QUERY *is*

- $O(nK)$ *time, if $k_q \in [K - 1]$.*

- $\mathcal{T}_{\mathrm{mat}}(n, k, n)$ *time, if $k_u = K$.*

*Proof.* The proof can be splitted into two cases.

For the case that $k_q \in [K - 1]$, Line 33 takes time $O(nk_u) = O(nK)$ obviously.

For the case that $k_q = K$, Line 31 runs in time $\mathcal{T}_{\mathrm{mat}}(n, k, n)$. Thus we complete the proof. □

**Lemma F.4** (Correctness for QUERY). *the procedure* QUERY *outputs* $(I - V^\top V)\mathsf{g}$ *where* $\mathsf{g} \in \mathbb{R}^n$ *is the next vector in $G$ to be used.*

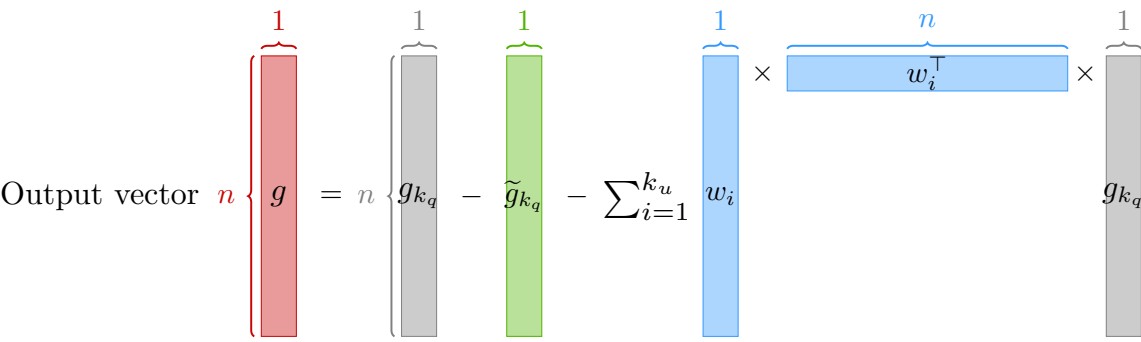

**Figure 5.** The decomposition of the output vector by QUERY. The green composition is the pre-computed factor and the blue composition is the new added ones, these two together form the projection of $g$ onto the row span of $V$.

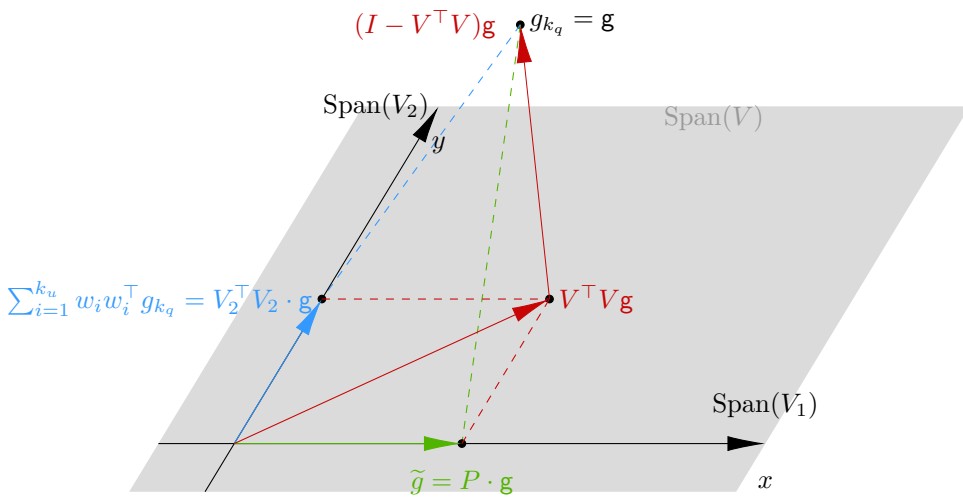

*Figure 6.* The Visualization of the decomposition. Here we denote the matrix of rows encoded to $P$ by $V_1$, and the matrix of rows not encoded yet by $V_2$. Thus we have that, $P = V_1^\top V_1$. And we assume that, $\text{Span}(V_1) = \text{Span}(e_x)$, $\text{Span}(V_2) = \text{Span}(e_y)$. Thus we visualize the idea of the decomposition as above. We divide the projection onto $\text{Span}(V)$ into the projection onto $\text{Span}(V_1)$ and the projection onto $\text{Span}(V_2)$. That is, we have $V^\top V g = V_1^\top V_1 \cdot g + V_2^\top V_2 \cdot g = \widetilde{g} + \sum_{i=1}^{k_u} w_i w_i^\top g$.

*Proof.* We divide the rows of $V \in \mathbb{R}^{\ell \times n}$, into two parts: the rows that have been encoded into $P$, and the ones that have not. Without loss of generality, we note the first $\ell - k_u$ rows are encoded, and the last $k_u$ rows are not. Thus we have that

$$(I - V^\top V)g$$

$$= g - \sum_{i=1}^{\ell} v_i v_i^\top g$$

$$= g - \sum_{i=1}^{\ell - k_u} v_i v_i^\top g - \sum_{i=1}^{k_u} w_i w_i^\top g$$

$$= g - \widetilde{g} - \sum_{i=1}^{k_u} w_i w_i^\top g,$$

where the first step follows from split the multiplication, the second step follows from split the summation, and the last step follows from the definition of $\widetilde{g}$ and the construction $P$. Thus we complete the proof.

$\square$

### F.4. Update

The goal of this section is to prove Lemma F.5.

**Lemma F.5** (Running time for UPDATE). *The running time of procedure* UPDATE *is*

- $O(nK)$ *time, if* $k_q \in [K - 1]$.
- $\mathcal{T}_{\mathrm{mat}}(n, k, n)$ *time, if* $k_u = K$.

*Proof.* For the UPDATE procedure, the running time can be divided into the following lines,

- Line 6 takes time $O(nK)$.
- Line 7 takes time $O(n)$.

- If $k$ reaches $K$, Line 9 takes time $\mathcal{T}_{\mathrm{mat}}(n, K, n) + \mathcal{T}_{\mathrm{mat}}(K, n, n)$.

Thus we have that, if $k$ doesn't reach $K$, the running time is

$$O(nK).$$

If $k$ reaches $K$, the running time is

$$\mathcal{T}_{\mathrm{mat}}(n, K, n) + \mathcal{T}_{\mathrm{mat}}(K, n, n).$$

$\square$

### F.5. Restart

The goal of this section is to prove Lemma F.6.

**Lemma F.6** (Running time for RESTART). *The running time of procedure* UPDATE *is*

$$O(\mathcal{T}_{\mathrm{mat}}(K, n, n)).$$

*Proof.* For the RESTART procedure, the running time can be divided as the following lines

- Line 15 takes time $\mathcal{T}_{\mathrm{mat}}(n, K, n)$.

- Line 17 takes time $O(nK)$.

- Line 18 takes time $\mathcal{T}_{\mathrm{mat}}(K, n, n)$.

Adding them together, we have the runnning time is

$$\mathcal{T}_{\mathrm{mat}}(n, K, n) + \mathcal{T}_{\mathrm{mat}}(K, n, n) = O(\mathcal{T}_{\mathrm{mat}}(K, n, n)).$$

$\square$

### F.6. Running Time of the Maintain algorithm

The goal of this section is to prove Lemma F.7.

**Lemma F.7** (Running time of FASTMAINTAIN). *Let $N = O(n)$. For any parameter $a \in [0, \alpha]$, let $K = n^a$ where $\alpha$ is the dual exponent of matrix multiplication. For any input $G \in \mathbb{R}^{n \times N}$, and an input $b \in \{0, 1\}^{n \times N}$ such that only one entry of each row of $b$ is $1$, the procedure* FASTMAINTAIN *(Algorithm 8) runs in time*

$$O(n^\omega + n^{2+a} + n^{3-a}).$$

*For the current $\omega \approx 2.373$ and $\alpha \approx 0.31$. Due to the above equation, if we choose the $a = \min\{\alpha, 1/2\}$, then the running time can be simplified to*

$$O(n^{2.5} + n^{3-\alpha}).$$

**Remark F.8.** *Instead of using $\alpha$, we can use the $\omega(\cdot, \cdot, \cdot)$ function in Table 3 in (Gall & Urrutia, 2018). In Lemma F.10, we provide a tighter analysis of Lemma F.7.*

*Proof.* The running time can be divided as follows.

- Line 2 takes time $\mathcal{T}_{\mathrm{mat}}(n, n, n) + \mathcal{T}_{\mathrm{mat}}(K, n, n) = O(n^\omega + n^2)$ to initialize the data structure.

- Run the following lines for $N = O(n)$ times (We ignore the condition of restart here):

    - Line 4 takes time $O(nK) = O(n^{1+a})$ to query for $g$.

– Line 7 takes time $O(nK) = O(n^{1+a})$ time update the data structure.

The above runs in $O(n^{2+a})$ time in total.

- For the restart condition, we note that, UPDATE is called for $O(n)$ times, so we will call RESTART for $O(n/K) = O(n^{1-a})$ times, each time it will take $O(n^2)$ time for restarting. The condition that QUERY calls RESTART is as the same. Thus we have the restart time bounded by $O(n^{3-a})$.

Combining the above together, we have the running time of FASTMAINTAIN is

$$O(n^\omega + n^{2+a} + n^{3-a}).$$

$\square$

## F.7. Correctness

**Algorithm 8** The purpose of writing down SLOWMAINTAIN is proving the correctness of FASTMAINTAIN. See Lemma F.9.

```
1:  procedure FASTMAINTAIN(b ∈ {0,1}^{n×N}, G ∈ ℝ^{n×N}, n, N)
2:      ds.INIT(V, G, K)                                                    ▷ Algorithm 6
3:      for t = 1 → N do
4:          g_t ← ds.QUERY()                                               ▷ Algorithm 6
5:          for i = 1 → n do
6:              if b_{i,t} = 1 then
7:                  ds.UPDATE(e_i)                                         ▷ Algorithm 7
8:              end if
9:          end for
10:     end for
11:     return g_1, ⋯, g_N
12: end procedure
13:
14: procedure SLOWMAINTAIN(b ∈ {0,1}^{n×N}, G ∈ ℝ^{n×N}, n, N)
15:     for t = 1 → N do
16:         g_t ← (I − V^⊤ V)G_{*,t}                              ▷ G_{*,t} is the t-th column of G
17:         for i = 1 → n do
18:             if b_{i,t} = 1 then
19:                 Add e_i to V according to Algorithm 1 and update V
20:             end if
21:         end for
22:     end for
23:     return g_1, ⋯, g_N
24: end procedure
```

The goal of this section is to prove Lemma F.9

**Lemma F.9.** *For any input $b \in \{0,1\}^{n \times N}$ and $G \in \mathbb{R}^{n \times N}$, the $N$ vectors outputted by procedure* FASTMAINTAIN *(Algorithm 8) are exactly same as the $N$ vectors outputted by procedure* SLOWMAINTAIN *(Algorithm 8).*

*Proof.* For all $i \in [N]$, we denote vectors output by the two algorithms by $g_{i,F}$ for FASTMAINTAIN and $g_{i,S}$ for SLOW-MAINTAIN.

**Induction Hypothesis.** For any $t \in [N]$, we have that $g_{i,F} = g_{i,S}$ for all $i \in [t-1]$.

**Proof of Base Case.** For the case that $i = 1$, we have by Lemma F.4 that

$$g_{1,F} = (I - V^\top V)G_{*,1} = g_{1,S}.$$

Thus the base case holds.

**Proof of Inductive case.** For the inductive case, we prove it by the following steps. We first notice that, with the binary matrix $b$, we have added some vectors into $V$ through the procedure when it reaches $t$-th iteration. We divide it into the following two circumstances:

**No vectors added to $V$ after the last query of $g_{t-1}$.** Since no vectors are added to $V$ after query of $g_{t-1}$, the $V$ stays the same when querying the $g_t$. Thus by Lemma F.4, we have that

$$g_{i,F} = (I - V^\top V)\mathsf{G}_{*,1} = g_{i,S}.$$

Hence we have the correctness.

**There are vectors added to $V$ after the last query of $g_{t-1}$.** If there's vectors added to the matrix $V$, then the FASTMAINTAIN calls the UPDATE for some times, and added some vectors into ds. Without loss of generality, we assume the UPDATE never calls RESTART. Then we have that, new vectors are encoded as $w$ in the ds. We divide the rows into

- Rows added before querying $g_{t-1}$, denoted the rows as the first $W_0$ ones.

- Rows added after querying $g_{t-1}$ and before querying $g_t$, denote the set as the last $W_1$ ones.

We denote the Gaussian vector as g, Thus we have that query $(I - V^\top V)\widetilde{g}_t$ as follows,

$$g_{t,S} = (I - V^\top V)\mathsf{g}_{t,S} = \mathsf{g}_{t,S} - \sum_{i \in [W_0]} w_{i,S} w_{i,S}^\top \mathsf{g}_{t,S} - \sum_{i=W_0+1}^{W_0+W_1} w_{i,S} w_{i,S}^\top \mathsf{g}_{t,S}.$$

And in the FASTMAINTAIN (Algorithm 8), when we call QUERY, we divide the rows as the ones encoded into $P$ and the ones not, that is

$$g_{t,F} = \mathsf{g}_{t,F} - \widetilde{g}_{t,F} - \sum_{i \in [k_u]} w_{i,F} w_{i,F}^\top \mathsf{g}_{t,F}.$$

If we denote the rows have been encoded into $P$ as the first $\ell$ rows. We have that

$$\widetilde{g}_{t,F} = \sum_{i \in [\ell]} w_{i,F} w_{i,F}^\top \mathsf{g}_{t,F}.$$

By the construction of the procedures, we have that

$$\ell + k_u = W_0 + W_1,$$

and

$$w_{i,F} = w_{i,S}$$

for each $i \in [\ell + k_u]$. And by the using of the counter $\tau_q$ in the ds, we have that

$$\mathsf{g}_{t,F} = \mathsf{g}_{t,S}.$$

Then by Lemma F.4, we have the $t$-th query

$$\text{ds.QUERY}() = (I - V^\top V)G_{*,t}.$$

Thus we complete the proof.

$\square$

### F.8. Tighter Running Time Analysis

The time analysis of Lemma F.7 is not tight. We can further improve it by using function $\omega(\cdot, \cdot, \cdot)$.

**Lemma F.10** (A Tighter Analysis of Running time of FASTMAINTAIN). *Let $N = O(n)$. For any parameter $a \in [0,1]$, let $K = n^a$. For any input $G \in \mathbb{R}^{n \times N}$, and an input $b \in \{0,1\}^{n \times N}$ such that only one entry of each row of $b$ is 1, the procedure* FASTMAINTAIN *(Algorithm 8) runs in time*

$$O(n^{\omega(1,1,1)} + n^{2+a} + n^{1+\omega(1,1,a)-a}).$$

*For the current $\omega(\cdot,\cdot,\cdot)$ function (see Table 3 in (Gall & Urrutia, 2018)). We can choose $a = 0.529$, then the running time becomes*

$$O(n^{2.53}).$$

*For the ideal $\omega$, we can choose $a = 0.5$, then the running time becomes $n^{2.5}$.*

# G. Implicit Leverage Score Sampling Algorithm

In this section, we present two crucial subroutines. That is, leverage score approximation and a fast sampling algorithm based on it.

## G.1. Leverage Score Approximation

Here we present the following algorithm, which providing a fast leverage score approximation.

---
**Algorithm 9** Implicit Leverage Score Computation

---
1: **procedure** IMPLICITLEVERAGESCORE($A \in \mathbb{R}^{m \times n}, V \in \mathbb{R}^{n \times n}, \epsilon_\sigma = \Theta(1), \delta_\sigma$)           ▷ Lemma G.2
2:                                                           ▷ The goal of this procedure is to compute a constant approximation
3:      $s_1 \leftarrow \widetilde{O}(\epsilon_\sigma^{-2} n)$
4:      Choose $S_1 \in \mathbb{R}^{s_1 \times m}$ to be sparse embedding matrix                                        ▷ Definition B.4
5:      Compute $(S_1 \cdot A)$                                                       ▷ It takes $\widetilde{O}(\epsilon_\sigma^{-1} \mathrm{nnz}(A))$ time.
6:      Compute $M \leftarrow (S_1 \cdot A) \cdot (I - V^\top V)$                                    ▷ It takes $\widetilde{O}(\epsilon_\sigma^{-2} n^\omega)$ time
7:      Let $R \in \mathbb{R}^{n \times n}$ denote the R of QR factorization of $M$
8:      $s_2 \leftarrow \Theta(\log(m/\delta_\sigma))$
9:      Choose $S_2 \in \mathbb{R}^{n \times s_2}$ to be a JL matrix                                  ▷ Either Definition B.1 or Definition B.2
10:     Compute $N \leftarrow (I - V^\top V)R^{-1}S_2$                                             ▷ $N \in \mathbb{R}^{n \times s_2}$
11:     **for** $j = 1 \to m$ **do**                                                    ▷ It takes $\widetilde{O}(\mathrm{nnz}(A))$ time
12:         Compute $\widetilde{\sigma}_j \leftarrow \|(e_j^\top A)N\|_2^2$
13:     **end for**
14:     **return** $\widetilde{\sigma}$                                               ▷ $\widetilde{\sigma}_j = \Theta(\sigma_j), \forall j \in [m]$
15: **end procedure**

---

**Definition G.1** (Leverage score). *Let $A \in \mathbb{R}^{m \times n}$. The leverage score of $A$ is a vector $\sigma(A) \in \mathbb{R}^m$ satisfying*

$$\sigma(A)_i = a_i(A^\top A)^{-1}a_i^\top,$$

*where $a_i$ denote the $i$-th row of $A$.*

The above algorithm provides an approximation to all the leverage scores of matrix $A(I - V^\top V)$, see the following lemma.

**Lemma G.2** (Implicit leverage score). *Given a matrix $A \in \mathbb{R}^{m \times n}$ and a matrix $V \in \mathbb{R}^{n \times n}$, there is an algorithm (procedure* IMPLICITLEVERAGESCORE *Algorithm 9) that runs in $\widetilde{O}(\epsilon_\sigma^{-2}(\mathrm{nnz}(A) + n^\omega))$ time and output a vector $\widetilde{\sigma} \in \mathbb{R}^m$, such that, $\widetilde{\sigma}$ is an approximation of the leverage score of matrix $A(I - V^\top V)$, i.e.,*

$$\widetilde{\sigma} \in (1 \pm \epsilon_\sigma) \cdot \sigma(A(I - V^\top V)),$$

*with probability at least $1 - \delta_\sigma$. The $\widetilde{O}$ hides the $\log(n/\delta_\sigma)$ factors.*

**Remark G.3.** *The only difference between classical statement is, in classical there is one input matrix $A$. In our case, the target matrix is implicitly given in a way $A(I - V^\top V)$, and we're not allowed to formally write down $A(I - V^\top V)$. The correctness proof is similar as literature (Clarkson & Woodruff, 2013; Nelson & Nguyên, 2013).*

*For our task, we only need an overestimation of the leverage score. We believe that the two-sides error might have future applications, therefore, we provide a two-sided error proof.*

*Proof.* We follow an approach of (Woodruff, 2014), but instead we use a high precision sparse embedding matrix (Nelson & Nguyên, 2013) and prove a two-sided bound on leverage score.

**Correctness.**

Let $S_1$ be a sparse embedding matrix with $s = O(\epsilon_\sigma^{-2} n \operatorname{poly}\log(n/(\epsilon_\sigma \delta_\sigma)))$ rows and column sparsity $O(\epsilon^{-1} \log(n/\delta))$. We first compute $M := (S_1 A) \cdot (I - V^\top V)$, then compute the QR decomposition $M = QR$.

Now, let $S_2 \in \mathbb{R}^{n \times s_2}$ matrix with $s_2 = O(\epsilon_\sigma^{-2} \log(m/\delta_\sigma))$, each entry of $S_2$ is i.i.d. $\mathcal{N}(0, 1/s_2)$ random variables. Then we generate an sketched matrix $N := (I - V^\top V) R^{-1} S_2$. Set $\widetilde{\sigma}_j = \|(e_j^\top A) N\|_2^2$ for all $j \in [m]$. We argue $\widetilde{\sigma}_j$ is a good approximation to $\sigma_j$.

First, with failure probability at most $\delta_\sigma / m$, we have that $\widetilde{\sigma}_j \in (1 \pm \epsilon_\sigma) \cdot \|e_j^\top A(I - V^\top V) R^{-1}\|_2^2$ via Lemma B.8. Now, it suffices to argue that $\|e_j^\top A(I - V^\top V) R^{-1}\|_2^2$ approximates $\|e_j^\top U\|_2^2$ well, where $U \in \mathbb{R}^{m \times n}$ is the left singular vectors of $A(I - V^\top V)$. To see this, first observe that for any $x \in \mathbb{R}^n$,

$$
\begin{aligned}
\|A(I - V^\top V) R^{-1} x\|_2^2 &= (1 \pm \epsilon_\sigma) \cdot \|S_1 A(I - V^\top V) R^{-1} x\|_2^2 \\
&= (1 \pm \epsilon_\sigma) \cdot \|Qx\|_2^2 \\
&= (1 \pm \epsilon_\sigma) \cdot \|x\|_2^2,
\end{aligned}
$$

where the first step follows from Lemma B.10, and the last step is due to $Q$ has orthonormal columns.

This means that all singular values of $A(I - V^\top V) R^{-1}$ are in the range $[1 - \epsilon_\sigma, 1 + \epsilon_\sigma]$. Now, since $U$ is an orthonormal basis for the column space of $A(I - V^\top V)$, $A(I - V^\top V) R^{-1}$ and $U$ has the same column space (since $R$ is full rank). This means that there exists a change of basis matrix $T \in \mathbb{R}^{n \times n}$ with $A(I - V^\top V) R^{-1} T = U$. Our goal is to provide a bound on all singular values of $T$. For the upper bound, we claim the largest singular value is at most $1 + 2\epsilon_\sigma$, to see this, suppose for the contradiction that the largest singular is larger than $1 + 2\epsilon_\sigma$ and let $v$ be its corresponding (unit) singular vector. Since the smallest singular value of $AR^{-1}$ is at least $1 - \epsilon_\sigma$, we have

$$
\begin{aligned}
\|A(I - V^\top V) R^{-1} T v\|_2^2 &\geq (1 - \epsilon_\sigma) \|Tv\|_2^2 \\
&> (1 - \epsilon_\sigma)(1 + 2\epsilon_\sigma) \\
&> 1,
\end{aligned}
$$

however, recall $A(I - V^\top V) R^{-1} T = U$, therefore $\|A(I - V^\top V) R^{-1} T v\|_2^2 = \|Uv\|_2^2 = \|v\|_2^2 = 1$, a contradiction.

One can similarly establish a lower bound of $1 - 2\epsilon_\sigma$. Hence, the singular values of $T$ are in the range of $[1 - 2\epsilon_\sigma, 1 + 2\epsilon_\sigma]$. This means that

$$
\begin{aligned}
\|e_j^\top A(I - V^\top V) R^{-1}\|_2^2 &= \|e_i^\top U T^{-1}\|_2^2 \\
&= (1 \pm 2\epsilon_\sigma) \cdot \|e_j^\top U\|_2^2 \\
&= (1 \pm 2\epsilon_\sigma) \cdot \sigma(A(I - V^\top V))_j,
\end{aligned}
$$

as desired. Scaling $\epsilon_\sigma$ by $\epsilon_\sigma / 2$ yields the approximation result.

**Running Time.**

We divided the running time of the algorithm into the following lines.

- Line 6 takes time $\widetilde{O}(\epsilon_\sigma^{-1} \operatorname{nnz}(A))$ to compute the matrix $M$.

- Line 7 takes time $\widetilde{O}(\epsilon_\sigma^{-2} n^\omega)$ to compute the QR decomposition of $M$.

- In Line 10, we can first multiply $R^{-1}$ with $S_2$ in time $\widetilde{O}(\epsilon_\sigma^{-2} n^2)$, this gives a matrix of size $n \times s_2$. Multiplying this matrix with $(I - V^\top V)$ takes $\widetilde{O}(n^\omega)$ time. Note that $R \in \mathbb{R}^{n \times n}$ hence $R^{-1}$ can be computed in $O(n^\omega)$ time.

- Line 11 loop takes in total $\widetilde{O}(\mathrm{nnz}(A))$ time to compute the output vector $\widetilde{\sigma}$.

Hence, the overall time for computing $\widetilde{\sigma}(A(I - V^\top V)) \in \mathbb{R}^m$ is $\widetilde{O}(\epsilon_\sigma^{-2}(\mathrm{nnz}(A) + n^\omega))$. $\qquad \square$

### G.2. Fast Subsampling

Here we present the fast subsampling algorithm based on the leverage score approximation.

---

**Algorithm 10** Fast Subsample Algorithm

---

1: **procedure** SUBSAMPLE($B \in \mathbb{R}^{m \times n}, V \in \mathbb{R}^{n \times n}, \epsilon_B, \delta_B$) $\qquad\qquad\qquad\qquad\qquad \triangleright$ Lemma G.4
2: $\quad$ $\widetilde{\sigma} \leftarrow$ IMPLICITLEVERAGESCORE($B, V, \delta_B$) $\qquad \triangleright$ Compute an $O(1)$-approximation to the leverage scores of $B(I - V^\top V), \widetilde{\sigma} \in \mathbb{R}^m$
3: $\quad$ Sample a subset rows of $\overline{B} = B(I - V^\top V)$ with proper re-weighting according to approximate leverage score $\widetilde{\sigma}$
4: $\quad$ Let $\widetilde{D}$ denote the diagonal sampling matrix such that $\Pr[\overline{B}^\top \widetilde{D}\widetilde{D}\overline{B} \in (1 \pm \epsilon_B) \cdot \overline{B}^\top \overline{B}] \geq 1 - \delta_B$
5: $\quad$ **return** $\widetilde{D}$ $\qquad\qquad\qquad\qquad\qquad\qquad\qquad\qquad\qquad\qquad\qquad \triangleright \widetilde{D} \in \mathbb{R}^{m \times m}$
6: **end procedure**

---

**Lemma G.4.** *There is an algorithm (Algorithm 10) takes $B \in \mathbb{R}^{m \times n}$ and $V \in \mathbb{R}^{n \times n}$ as inputs and outputs a diagonal sampling matrix $\widetilde{D}$ with $\|\widetilde{D}\|_0 = O(\epsilon_B^{-2} n \log(n/\delta_B))$ and runs in time*

$$\widetilde{O}(\mathrm{nnz}(B) + n^\omega).$$

*Here $\widetilde{O}$ hides the $\log(n/\delta_B)$ factors. Here $\omega$ is the exponent of matrix multiplication. Currently, $\omega \approx 2.373$.*

*Proof.* We first approximately compute the leverage score, i.e., gives an $O(1)$-approximation to all leverage scores via Lemma G.2. Then run we samples a number of rows according to the leverage scores, using Lemma H.2, we can show the correctness. $\qquad \square$

## H. Sampling a Batch of Rank-$1$ Terms

In this section, we give a sparsification tool for the matrix that has pattern $A^\top A$.

We define our sampling process as follows:

**Definition H.1** (Sampling process). *Let $H = A^\top A$. Let $p_j \geq \beta \cdot \sigma_j(A)/n$, suppose we sample with replacement independently for $s$ rows of matrix $A$, with probability $p_j$ of sampling row $j$ for some $\beta \geq 1$. Let $j_t$ denote the index of the row sampled in the $t$-th trial. Define the generated sampling matrix as*

$$\widetilde{H} := \frac{1}{T} \sum_{t=1}^{T} \frac{1}{p_{j_t}} a_{j_t} a_{j_t}^\top,$$

*where $T$ denotes the number of the trials.*

For our sampling process defined as Definition H.1, we can have the following guarantees:

**Lemma H.2** (Sample using Matrix Chernoff). *Let $\epsilon_0, \delta_0 \in (0, 1)$ be precision and failure probability parameters, respectively. Suppose $\widetilde{H}$ is generated as in Definition H.1, then with probability at least $1 - \delta_0$, we have*

$$(1 - \epsilon_0) \cdot H \preceq \widetilde{H} \preceq (1 + \epsilon_0) \cdot H.$$

*Moreover, the number of rows sampled is*

$$T = \Theta(\beta \cdot \epsilon_0^{-2} n \log(n/\delta_0)).$$

*Proof.* The proof of this Lemma is follows by designing the sequence of random matrices, then applying the matrix Chernoff bound to get the desired guarantee.

We first define the following vector generated from scaling the rows of $A$,

$$y_j = (A^\top A)^{-1/2} \cdot a_j$$

for all $j \in [m]$. And for $i \in [m]$, we define the matrix $Y_j := \frac{1}{p_j} y_j y_j^\top$ generated by $y_j$, and we define $X_j := Y_j - I_n$, where $I_n$ is $n \times n$ identity matrix.

Based on the above vector and matrix definitions, we define the following distributions:

- We define a distribution y for random vector: For $y \in \mathbb{R}^n$, if $y \sim$ y, then for each $j \in [m]$, $y = y_j$ with probability $p_j$.

- We define a distribution $\mathcal{Y}$ such that, For $Y \in \mathbb{R}^{n \times n}$, if $Y \sim \mathcal{Y}$, then for each $j \in [m]$, $Y = Y_j$ with probability $p_j$.

- We define a distribution $\mathcal{X}$ such that, For $X \in \mathbb{R}^{n \times n}$, if $X \sim \mathcal{X}$, then for each $j \in [m]$, $X = X_j = Y_j - I_n$ with probability $p_j$.

Using $H = A^\top A$, we write the $y_j$ as

$$y_j = H^{-1/2} \cdot a_j.$$

We notice that,

$$
\begin{aligned}
\sum_{j=1}^m y_j y_j^\top &= \sum_{j=1}^m H^{-1/2} a_j a_j^\top H^{-1/2} \\
&= H^{-1/2} (\sum_{i=j}^m a_j a_j^\top) H^{-1/2} \\
&= H^{-1/2} (A^\top A) H^{-1/2} \\
&= I_n.
\end{aligned}
\tag{12}
$$

where the first step follows from the definition of $y_j$, the second step follows from reorganization, the third step follows from definition of $a_j$, and the last step follows from the definition of $H$.

Besides, we notice the the connection between the norm of $y_j$ and the leverage score of $A$:

$$
\begin{aligned}
\|y_j\|_2^2 &= a_j^\top (A^\top A)^{-1} a_j \\
&= \sigma_j(A).
\end{aligned}
\tag{13}
$$

Here we denote the index of the row that we sample during $t$-th trial as $j_t$, note that $j_t \in [m]$, for all $t \in [T]$.

**Unbiased Estimator.** For a matrix $X \sim \mathcal{X}$, here we show that $X$ has the expectation of $0$, we note that

$$
\begin{aligned}
\mathbb{E}_{X \sim \mathcal{X}}[X] &= \mathbb{E}_{Y \sim \mathcal{Y}}[Y] - I_n \\
&= (\sum_{j=1}^m p_j \cdot \frac{1}{p_j} y_j y_j^\top) - I_n \\
&= 0.
\end{aligned}
$$

where the first step follows from the definition of $X$, the second step follows from the definition of $Y$ and the definition of expectation, and the last step follows from Eq. (12).

**Upper Bound on $\|X\|$.** To give an upper bound of $\|X\|$, we first provide an upper bound for any $\|X_j\|$, then the upper bound of $\|X\|$ follows immediately. We note that,

$$\|X_j\| = \|Y_j - I_n\|$$

$$\leq 1 + \|Y_j\|$$

$$= 1 + \frac{\|y_j y_j^\top\|}{p_j}$$

$$\leq 1 + \frac{n \cdot \|y_j\|_2^2}{\beta \cdot \sigma_j(A)}$$

$$= 1 + \frac{n}{\beta}.$$

where the first step follows from the definition of $X_j$, the second step follows from triangle inequality and the definition of $I_n$, the third step follows from the definition of $Y_j$, the forth step follows from $p_j \geq \beta \cdot \sigma_j(A)/nd$ and the definition of $\ell_2$ norm and the last step follows from Eq. (13).

**Bound on** $\|\mathbb{E}[X^\top X]\|$. To upper bound the spectral norm of the matrix, we first provide the upper bound of the expectation of the matrix $X^\top X$. We compute the matrix expectation:

$$\mathbb{E}_{X_{j_t} \sim \mathcal{X}}[X_{j_t}^\top X_{j_t}]$$

$$= I_n + \mathbb{E}_{y_{j_t} \sim \mathbf{y}}\left[\frac{y_{j_t} y_{j_t}^\top y_{j_t} y_{j_t}^\top}{p_{j_t}^2}\right] - 2 \mathbb{E}_{y_{j_t} \sim \mathbf{y}}\left[\frac{y_{j_t} y_{j_t}^\top}{p_{j_t}}\right]$$

$$= I_n + (\sum_{j=1}^m \frac{\sigma_j(A)}{p_j} y_j y_j^\top) - 2I_n$$

$$\leq \sum_{j=1}^m \frac{n}{\beta} y_j y_j^\top - I_n$$

$$= (\frac{n}{\beta} - 1)I_n,$$

where the first step follows from definition of $X$, the second step follows from Eq. (12), Eq. (13) and the definition of expectation, the third step follows from $p_j \geq \beta \cdot \sigma_j(A)/n$, and the last step follows from Eq. (12) and factorising.

Thus we upper bound the spectral norm of $\mathbb{E}_{y_{j_t} \sim \mathbf{y}}[X_{j_t}^\top X_{j_t}]$ as

$$\|\mathbb{E}_{y_{j_t} \sim \mathbf{y}}[X_{j_t}^\top X_{j_t}]\| \leq \frac{n}{\beta} - 1.$$

**Put things together.** Here we define $W := \sum_{t=1}^T X_{j_t}$. We choose the parameter

$$\gamma = 1 + \frac{n}{\beta}, \quad \sigma^2 = \frac{n}{\beta} - 1$$

Then, we apply Matrix Chernoff Bound as in Lemma A.9:

$$\Pr[\|W\| \geq \epsilon_0]$$

$$\leq 2n \cdot \exp\left(-\frac{T\epsilon_0^2}{n/\beta - 1 + (1 + n/\beta)\epsilon_0/3}\right)$$

$$= 2n \cdot \exp(-T\epsilon_0^2 \cdot \Theta(\beta/n))$$

$$\leq \delta_0$$

where the last step follows from that we choose $T = \Theta(\beta \cdot \epsilon_0^{-2} n \log(n/\delta_0))$.

Finally, we notice that

$$W = \frac{1}{T}(\sum_{t=1}^T \frac{1}{p_{j_t}} y_{j_t} y_{j_t}^\top - I_n)$$

$$= H^{-1/2}(\frac{1}{T}\sum_{t=1}^{T}\frac{1}{p_{j_t}}a_{j_t}a_{j_t}^{\top})H^{-1/2} - I_n$$

$$= H^{-1/2}\widetilde{H}H^{-1/2} - I_n.$$

where the first step follows from definition of $W$, the second step follows from $y_{j_t} = H^{-1/2}a_{j_t}$, and the last step follows from definition of $\widetilde{H}$.

Since

$$\|H^{-1/2}\widetilde{H}H^{-1/2} - I_n\| \le \epsilon_0$$

Thus, we know that

$$(1-\epsilon_0)I_n \preceq H^{-1/2}\widetilde{H}H^{-1/2} \preceq (1+\epsilon_0)I_n$$

which implies that

$$(1-\epsilon_0)H \preceq \widetilde{H} \preceq (1+\epsilon_0)H.$$

Thus, we complete the proof. $\qquad\square$

## I. Boundary-Seeking Subroutine for Partial Coloring

In order to find the proper step size in the Partial Coloring algorithm, we propose the following subroutine.

---
**Algorithm 11**

---
1: **procedure** FINDBOUNDARY($a \in \mathbb{R}^n, b \in \mathbb{R}^n$)                    $\triangleright$ Lemma I.1
2:     Compute $n$ intervals $[x_i, y_i]$ such that $x_i, y_i$ are the two boundaries of $-1 \le a_i\mu - b_i \le 1$, e.g., $x_i, y_i \in \{\frac{b_i-1}{a_i}, \frac{b_i+1}{a_i}\}$ and $x_i \le y_i$, for all $i \in [n]$
3:     Compute $n$ intervals $[x_{n+i}, y_{n+i}]$ such that $x_i, y_i$ are the two boundaries of $-1 \le a_i\mu + b_i \le 1$, e.g., $x_i, y_i \in \{\frac{-b_i-1}{a_i}, \frac{-b_i+1}{a_i}\}$ and $x_i \le y_i$, for all $i \in [n]$
4:     Linear scan the $x_1, \cdots, x_{2n}$, find the index $u$ such that $u \leftarrow \max\{x_i\}_{i\in[2n]}$
5:     Linear scan the $x_1, \cdots, y_{2n}$, find the index $v$ such that $v \leftarrow \min\{y_i\}_{i\in[2n]}$
6:     **if** $u \le v$ **then**
7:         **if** $|u| \le |v|$ **then**
8:             **return** $v$
9:         **else**
10:            **return** $u$
11:        **end if**
12:    **else**
13:        **return** fail
14:    **end if**
15: **end procedure**

---

Here we illustrate the idea of this algorithm with an example in the following Figure 7.

**Lemma I.1.** *There is an algorithm (Algorithm 11) that takes two vectors $a, b \in \mathbb{R}^n$, and outputs an positive number $\mu \in \mathbb{R}_+$, such that $\mu$ is the number with the largest absolute value satisfying that*

$$\max\{\|\mu \cdot a + b\|_\infty, \|\mu \cdot a - b\|_\infty\} = 1.$$

*And runs in time $O(n)$.*

*Proof.* **Running Time.** The running time of Algorithm 11 can be divided into the following lines,

- Line 2 takes time $O(n)$ to compute the $x_i, y_i$'s for $i \in [n]$.

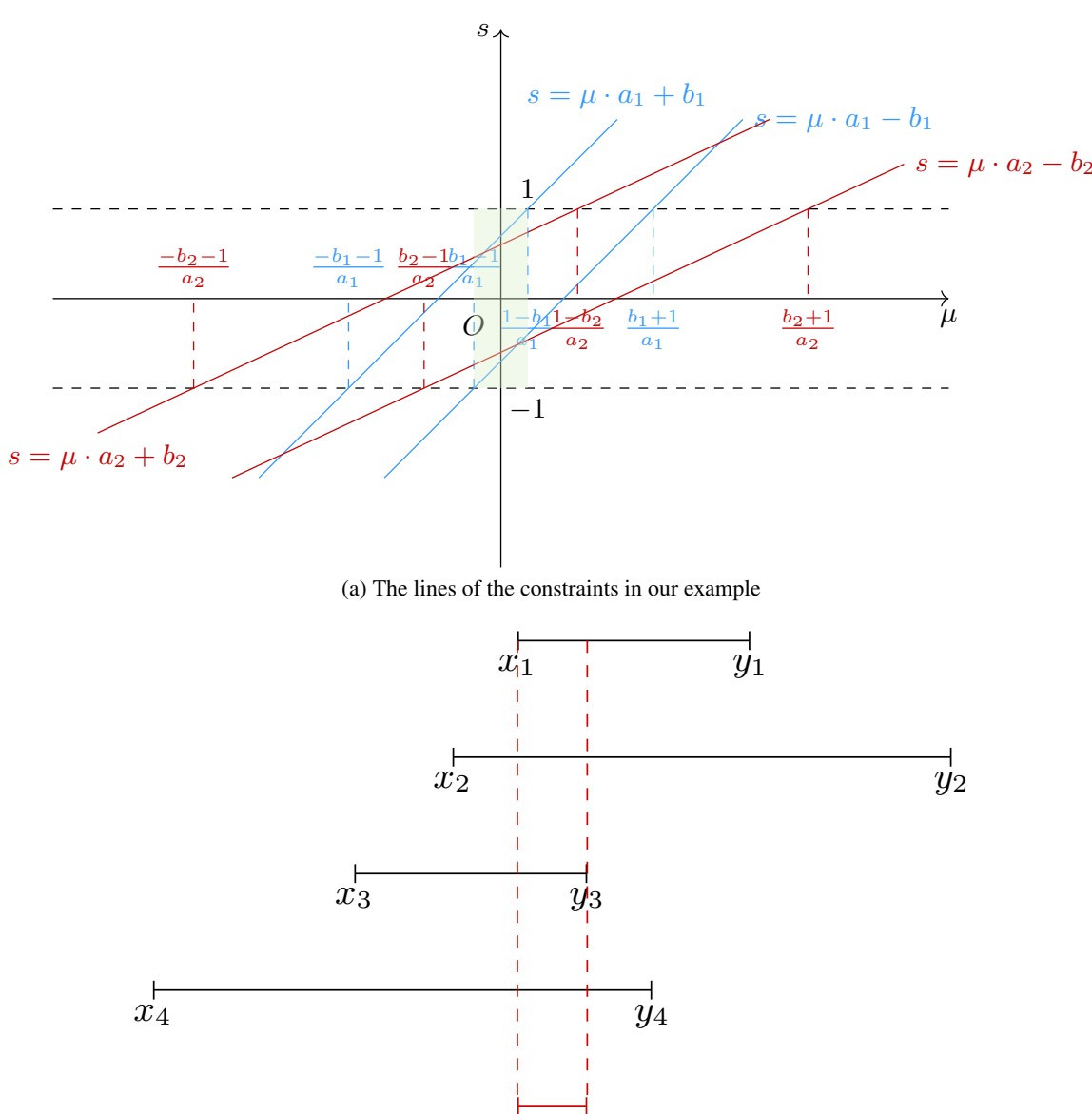

(a) The lines of the constraints in our example

(b) The feasible intervals

*Figure 7.* Here we give an example of our Algorithm 11. The algorithm is looking for the intervals to make every constraint ($|\mu \cdot a_i + b_i| \leq 1$ or $|\mu \cdot a_i - b_i| \leq 1$) hold, so we have $2n$ lines in total ($n$ is the dimension). Here in the example, we assume $n = 2$, so we have $2n = 4$ lines in total. Thus we have 4 intervals, and taking the intersection over them, we get the interval $[u, v]$. Figure (a) shows the lines of the constraints we have, the green area shows the feasible region. Figure (b) shows our idea of taking intersection over the $2n = 4$ constraints. (Here we set $a = (1, \frac{15}{7})$ and $b = (0.7, \frac{3}{5})$.)

- Line 3 takes time $O(n)$ to compute the $x_i, y_i$'s for $i \in \{n+1, \dots, 2n\}$.

- Line 4 takes time $O(n)$ to linear scan the $x_i$'s for $i \in [2n]$.

- Line 5 takes time $O(n)$ to linear scan the $y_i$'s for $i \in [2n]$.

The rest part of the algorithm all runs in an constant time. Thus we have the total running time is

$$O(n).$$

**Correctness.** We first prove that, if the algorithm outputs a number $\mu$, then for every $i \in [n]$, it holds that

$$-1 \le \mu \cdot a_i - b_i \le 1,$$
$$\text{and } -1 \le \mu \cdot a_i + b_i \le 1.$$

Here we call the feasible region of the above question to be $\mathcal{U}$. By the construction of our algorithm we have that,

$$[x_i, y_i] = \{\mu \in \mathbb{R} \mid -1 \le \mu \cdot a_i - b_i \le 1\}, \qquad\qquad \forall i \in [n]$$
$$\text{and } [x_i, y_i] = \{\mu \in \mathbb{R} \mid -1 \le \mu \cdot a_{i-n} - b_{i-n} \le 1\}, \qquad\qquad \forall i \in \{n+1, \ldots, 2n\}$$

Thus we have that

$$\mathcal{U} = \bigcap_{i \in [2n]} [x_i, y_i].$$

We note that, if the algorithm outputs a number $\mu$, then it must holds that

$$u \le v.$$

Recall that we define $u$ and $v$ to be

$$u := \max\{x_i\}_{i \in [2n]}$$
$$v := \min\{y_i\}_{i \in [2n]},$$

thus we have that

$$[u, v] = \mathcal{U}.$$

Recall the output number $\mu$ equals to either $u$ or $v$, it must stands that

$$\mu \in \mathcal{U}.$$

Thus the constraints hold for $\mu$.

Now we prove that

$$\max\{\|\mu \cdot a + b\|_\infty, \|\mu \cdot a - b\|_\infty\} = 1.$$

Without loss of generality, we assume $|u| > |v|$, that is, $\mu = u$. The case that $|u| \le |v|$ is just the same. We first define

$$j := \{i \in [2n] \mid u = x_i\}.$$

Here we first assume $j \in [n]$, that is, we have that

$$|\mu \cdot a_j - b_j| = 1.$$

(For the case that $j \in \{n+1, \ldots, 2n\}$, we have $|\mu \cdot a_j + b_j| = 1$, which is the same to analyse.) Note that, $\mu \in \mathcal{U}$, so we have that

$$|\mu \cdot a_i - b_i| \le 1, \ \forall i \in [n]$$
$$\text{and } |\mu \cdot a_i + b_i| \le 1, \ \forall i \in [n].$$

Thus we have that

$$\|\mu \cdot a - b\|_\infty = 1$$
$$\text{and } \|\mu \cdot a + b\|_\infty \le 1,$$

which is

$$\max\{\|\mu \cdot a + b\|_\infty, \|\mu \cdot a - b\|_\infty\} = 1.$$

Now for the case that $j \in \{n+1, \ldots, 2n\}$, by a similar analysis we have that

$$\|\mu \cdot a + b\|_\infty = 1$$
$$\text{and } \|\mu \cdot a - b\|_\infty \leq 1,$$

which also implies the result that

$$\max\{\|\mu \cdot a + b\|_\infty, \|\mu \cdot a - b\|_\infty\} = 1.$$

By a same analysis we can have the above result if $\mu = v$.

Now we prove that, $\mu = u$ and $\mu = v$ are the only two case that

$$\max\{\|\mu \cdot a + b\|_\infty, \|\mu \cdot a - b\|_\infty\} = 1.$$

Suppose for the contradiction that there exists an $t \in \mathbb{R}$ such that $t \neq u$ and $t \neq v$, and it holds that

$$\max\{\|t \cdot a + b\|_\infty, \|t \cdot a - b\|_\infty\} = 1. \tag{14}$$

If $t \notin \mathcal{U}$, then there must exists and $i \in [n]$ such that

$$|t \cdot a_i + b_i| > 1$$
$$\text{or } |t \cdot a_i - b_i| > 1,$$

which is a violation of the hypothesis (Eq.(14)). Then for the case that $t \in \mathcal{U}$, since $t \neq u$ and $t \neq v$, it must holds that $t \in (u, v)$. Note we define

$$u := \max\{x_i\}_{i \in [2n]}$$
$$v := \min\{y_i\}_{i \in [2n]}.$$

It must holds that

$$t > x_i, \ \forall i \in [2n], \text{and } t < y_i, \ \forall i \in [2n].$$

Then we have that

$$|t \cdot a_i + b_i| < 1$$
$$\text{or } |t \cdot a_i - b_i| < 1,$$

which implies that

$$\max\{\|t \cdot a + b\|_\infty, \|t \cdot a - b\|_\infty\} < 1,$$

which is a violation of the hypothesis (Eq.(14)). Thus we conclude there is no such $t$.

If we define $\mu$ to be one of $u$ or $v$, depending on who has larger absolute value, then $\mu$ is the number with the largest absolute value satisfying that

$$\max\{\|\mu \cdot a + b\|_\infty, \|\mu \cdot a - b\|_\infty\} = 1.$$

Thus we complete the proof. □

**Corollary I.2.** *For the case that $\|b\|_\infty \leq 1$, Algorithm 11 always return a number $\mu$.*

*Proof.* If $\|b\|_\infty \leq 1$, then we have that

$$|b_i| \leq 1, \ \forall i \in [n].$$

Without loss of generality we assume $b_i \geq 0$, then $b_i \in [0, 1]$, which implies that

$$\frac{b_i - 1}{a_i} \leq 0$$

$$\text{and } \frac{b_i + 1}{a_i} \geq 0.$$

Thus we have that

$$0 \in [x_i, y_i], \ \forall i \in [n].$$

By a similar analysis we have

$$0 \in [x_i, y_i], \ \forall i \in \{n + 1, \ldots, 2n\}.$$

Thus we have

$$0 \in \mathcal{U} = \bigcap_{i \in [2n]} [x_i, y_i],$$

which means

$$\mathcal{U} \neq \emptyset$$

So it must holds that

$$u \leq v.$$

Thus we complete the proof. $\qquad\square$

## J. Experimental Results

We conduct the experiments to empirically validate the efficiency of our algorithm. We focused our experimental evaluation on demonstrating the improvements achieved through the fast hereditary projection. This is because the lazy update scheme is primarily designed to leverage fast matrix multiplication. Although fast matrix multiplication theoretically has lower asymptotic complexity, algorithms based on FMM often suffer from significantly large constant factors, adversely affecting their practical runtime performance. Additionally, for practical considerations, the parameters used in the experiments do not strictly follow the theoretical suggestions in the algorithm. Our experimental setup follows the matrix configurations used in Larsen (2023):

- Uniform matrices: Every entry is chosen independently and uniformly from the set $\{-1, +1\}$.

- 2D corner matrices: Sample two point sets $P = \{p_1, \ldots, p_n\}$ and $Q = \{q_1, \ldots, q_m\}$ independently and uniformly from the unit square $[0, 1] \times [0, 1]$. Construct a matrix with one column per $p_j \in P$ and one row per $q_i \in Q$; set the entry $(i, j)$ to 1 if $p_j$ is dominated by $q_i$, i.e. $\langle q_i, x \rangle > \langle p_j, x \rangle$ and $\langle q_i, y \rangle > \langle p_j, y \rangle$, and to 0 otherwise.

- 2D halfspace matrices: Draw $P = \{p_1, \ldots, p_n\}$ uniformly from $[0, 1] \times [0, 1]$ and construct a set $Q$ of $m$ half-spaces as follows: pick a point $a$ uniformly on either the left or the top boundary of the unit square, a point $b$ uniformly on either the right or the bottom boundary, and then choose uniformly whether the half-space consists of all points above the line through $a$ and $b$ or all points below it. Form a matrix with one column per $p_j \in P$ and one row per half-space $h_i \in Q$; set the entry $(i, j)$ to 1 if $p_j \in h_i$ and to 0 otherwise.

Our algorithm achieves substantial speedups with only minor sacrifices in approximation guarantees. The speedup will be more significant once $m$ is much bigger than $n$, as sketching is known to work well in the regime where $m \gg n$.

Our code is available at https://github.com/magiclinux/input_sparsity_discrepancy_icml_2025.

*Table 2.* Results of experiments on uniform matrices. Here Larsen's algorithm is Larsen (2023). The runtime is measured in second.

| Matrix Size | Sparsity | Larsen's Obj. Val. | Our Obj. Val. | Larsen's Runtime | Our Runtime |
|---|---|---|---|---|---|
| $400 \times 400$ | 1.0 | 54 | 56 | 2.98 | 2.16 |
| $400 \times 400$ | 0.5 | 38 | 42 | 2.90 | 1.95 |
| $400 \times 400$ | 0.1 | 14 | 20 | 2.91 | 1.90 |
| $2000 \times 2000$ | 1.0 | 140 | 148 | 345 | 164 |
| $2000 \times 2000$ | 0.5 | 96 | 99 | 334 | 156 |
| $2000 \times 2000$ | 0.1 | 46 | 47 | 331 | 152 |
| $10000 \times 1000$ | 1.0 | 132 | 140 | 378 | 63 |
| $10000 \times 1000$ | 0.5 | 92 | 97 | 374 | 62 |
| $10000 \times 1000$ | 0.1 | 42 | 44 | 375 | 62 |

*Table 3.* Results of experiments on 2D corner matrices. Here Larsen's algorithm is Larsen (2023). The runtime is measured in second.

| Matrix Size | Sparsity | Larsen's Obj. Val. | Our Obj. Val. | Larsen's Runtime | Our Runtime |
|---|---|---|---|---|---|
| $400 \times 400$ | 1.0 | 24 | 28 | 2.80 | 2.15 |
| $400 \times 400$ | 0.5 | 30 | 40 | 2.79 | 2.18 |
| $400 \times 400$ | 0.1 | 17 | 18 | 2.75 | 1.93 |
| $2000 \times 2000$ | 1.0 | 52 | 60 | 347 | 170 |
| $2000 \times 2000$ | 0.5 | 83 | 92 | 352 | 169 |
| $2000 \times 2000$ | 0.1 | 45 | 44 | 350 | 181 |
| $10000 \times 1000$ | 1.0 | 46 | 52 | 386 | 65 |
| $10000 \times 1000$ | 0.5 | 76 | 77 | 374 | 60 |
| $10000 \times 1000$ | 0.1 | 37 | 40 | 375 | 62 |

*Table 4.* Results of experiments on 2D halfspace matrices. Here Larsen's algorithm is Larsen (2023). The runtime is measured in second.

| Matrix Size | Sparsity | Larsen's Obj. Val. | Our Obj. Val. | Larsen's Runtime | Our Runtime |
|---|---|---|---|---|---|
| $400 \times 400$ | 1.0 | 38 | 42 | 2.67 | 2.08 |
| $400 \times 400$ | 0.5 | 30 | 40 | 2.70 | 2.12 |
| $400 \times 400$ | 0.1 | 17 | 18 | 2.62 | 1.89 |
| $2000 \times 2000$ | 1.0 | 50 | 56 | 352 | 174 |
| $2000 \times 2000$ | 0.5 | 75 | 76 | 345 | 171 |
| $2000 \times 2000$ | 0.1 | 38 | 40 | 345 | 169 |
| $10000 \times 1000$ | 1.0 | 62 | 66 | 390 | 67 |
| $10000 \times 1000$ | 0.5 | 71 | 75 | 389 | 68 |
| $10000 \times 1000$ | 0.1 | 33 | 36 | 382 | 64 |

# K. More Related Work

Because sketching techniques are pivotal in both linear and semidefinite programming, we first survey this body of work. We then discuss how discrepancy theory has been leveraged in modern machine-learning applications.

**Linear programming**    Linear programming has a long and rich history in both mathematics and computer science. The simplex algorithm (Dantzig, 1947), while foundational, admits exponential worst-case running time. The ellipsoid method brought the first polynomial guarantee (Khachiyan, 1980), but was slower than simplex in practice. Karmarkar's interior-point method (Karmarkar, 1984) was a major advance, offering polynomial complexity together with strong empirical performance. When the number of constraints $d$ satisfies $d = \Omega(n)$ (where $n$ is the number of variables), Karmarkar's algorithm runs in $O(n^{3.5})$ time, later improved to $O(n^3)$(Vaidya, 1987; Renegar, 1988) and then to $O^*(n^{2.5})$(Vaidya, 1989). Cohen et al. (2019) achieved a time bound of $O(n^\omega + n^{2.5-\alpha/2} + n^{2+1/6})$, where $\omega$ is the matrix-multiplication exponent and $\alpha$ its dual. Their techniques were subsequently generalized to empirical risk minimization(Lee et al., 2019; Song, 2019; Qin et al., 2023; Gu et al., 2025), while Song & Yu (2021) reproduced the results of Cohen et al. (2019) using oblivious sketching matrices instead of non-oblivious sampling. For tall, dense constraint matrices, Brand et al. (2020) obtained a running time of $\widetilde{O}(nd) + \mathrm{poly}(d)$. There are also works studying linear programming in the streaming setting (Chen et al., 2023; Song et al., 2023c; Brand et al., 2025).

**Semidefinite programming**    Two principal families of algorithms address semidefinite programming (SDP): second-order and first-order methods. Second-order methods enjoy logarithmic dependence on the accuracy parameter $\epsilon$, whereas first-order methods typically have polynomial dependence on $1/\epsilon$. Among second-order techniques, cutting-plane approaches such as Lee et al. (2015); Jiang et al. (2020c; 2024) solve SDPs in $O(m(mn^2 + m^2 + n^\omega))$ time, where $m$ is the number of constraints. Interior-point methods based on self-concordant barriers constitute another major line: log-barrier algorithms (Nesterov & Nemirovski, 1992; Jiang et al., 2020b; Huang et al., 2022c; Liu et al., 2023), hybrid-barrier variants (Anstreicher, 2000; Huang et al., 2022c; Liu et al., 2023), and those using the Lee–Sidford barrier (Lee & Sidford, 2014; Liu et al., 2023; Gu et al., 2024a). Notably, Jiang et al. (2020b) obtains a running time of $O(\sqrt{n}(mn^2 + n^\omega))$, and Huang et al. (2022c) shows that when $m = \Omega(n^2)$, an SDP can be solved in $O(m^\omega + m^{2+1/4})$ time. First-order algorithms avoid second-order information but pay a higher cost in their dependence on $1/\epsilon$. Representative results include Arora & Kale (2007); Jain & Yao (2011); Allen Zhu et al. (2016); Garber & Hazan (2016); Allen-Zhu & Li (2017); Carmon et al. (2019); Lee & Padmanabhan (2020); Yurtsever et al. (2019); Grigorescu et al. (2022); Song et al. (2023a;c); Chen et al. (2023); Gu et al. (2024b).

**Applications of discrepancy theory in machine learning**    Matousek (1999a); Karnin & Liberty (2019) introduce the relationship of discrepancy and concepts in learning theory such as VC dimension and PAC learning. Chen et al. (2018) introduces a method to learn hash functions via discrepancy minimization. Learning hash functions in content-based image retrieval optimize binary codes to preserve similarity in high-dimensional feature spaces, enabling faster and more efficient image search. Wang et al. (2023) leverage discrepancy minimization for unsupervised graph matching by aligning predictions from classical solvers and neural models. Han et al. (2025) proposed an algorithm for compressing the KV cache recursively using a geometric correlated sampling process based on discrepancy theory. Nikolov et al. (2013) investigated the relationship between discrepancy minimization and differential privacy in the context of linear queries over histograms. Quasi-Monte Carlo methods (Lyu et al., 2020; Lyu, 2023) leverage concepts from discrepancy theory by employing low-discrepancy sequences to efficiently approximate high-dimensional integrals and expectations.

