# OpenReview forum: "Discrepancy Minimization in Input-Sparsity Time"
_ICML.cc/2025/Conference — ICML 2025 spotlightposter_

### Official Review · Reviewer_ZGyW · 2025-03-05

**Overall Recommendation:** 4

**Summary:**

The paper gives a new algorithm for discrepancy minimization over real valued matrices that nearly runs in input-sparsity time. Specifically, building on Bansal's and Larsen's previous algorithms, the authors give a

1) A combinatorial algorithm that runs in time $\tilde{O}(nnz(A) + n^3)$ time.

2) If Fast Matrix Multiplication (FMM) is allowed, then $\tilde{O}(nnz(A) +n^{2.53})$, breaking the cubic barrier for the first time.

The algorithm guarantees a coloring $x\in \set{-1,1}^n$ achieving discrepancy $O(\text{herdisc}(A) \log n \log^{1.5}m)$

They do this by introducing a host of interesting techniques, notably:

1) A new sketching method using implicit leverage‐score sampling (which is being utilised very heavily in a lot of recent papers) to quickly compute a hereditary projection (Theorem 1.3), while simultaneously avoiding explicitly calculating the entire projection matrix (the theorem is technical, this is just an informal description).

2) A “guess-and-correct” data structure that batches gaussian projections.

The paper is relatively readable, but quite frankly too technical sometimes. Intuition for the proofs of Theorems or even their utilities in the "big picture" is rarely communicated, and for a 51 pages paper, I expected significantly more handholding and intuition sharing throughout the reading journey. I've had to skip some proofs completely because they offered almost no intuition and were time sinks to verify, not to mention that they felt extremely mechanical.

Ultimately, I think the main result is interesting enough to be published in ICML since it tackles a long standing open problem, and it is clearly important, though I would've wished if the presentation was more accessible for a result of this importance.

**Claims And Evidence:**

The paper’s claims are backed by Lemmas/Theorems which are proved in details in the Appendix.

**Essential References Not Discussed:**

N/A

**Experimental Designs Or Analyses:**

N/A

**Methods And Evaluation Criteria:**

Since the contribution is theoretical, and there are no experiments, the evaluation centers on computational complexity and approximation guarantees rather than empirical performance. I think this part is adequately addressed.

**Other Comments Or Suggestions:**

N/A

**Other Strengths And Weaknesses:**

Strengths:

1) As mentioned, the theoretical improvement in runtime is substantial compared to previous methods, and is interesting.

2) The paper introduces a host of technical results that might be of independent interest.

Weaknesses:

1) The technical exposition is very dense and may be a turn-off for readers not already very familiar with the literature on discrepancy theory

2) As mentioned above, some parts of the analysis and proofs (especially regarding the correctness of the algorithm) are very subtle and could benefit from additional intuitive explanations.

3) Experiments! This could've been a very easy win, and I don't see why the authors don't include even a tiny experiment for their algorithms. I understand that there is a ton of parameters to choose (various $\varepsilon, \delta$ to choose), but looking at the algorithm, I really think they could've implemented a variant on a toy dataset.

4) The constants in the proofs are not optimised, and sometimes they're laughably huge (I get it, no one likes to optimize constants in proofs, but some constants in the papers are just absurd). I suspect if more effort was put into this, it would've easily translated into a feasible practical algorithm, but alas.

**Questions For Authors:**

N/A

**Relation To Broader Scientific Literature:**

The paper is well-situated in the existing literature on discrepancy minimisation. For example, it builds on the Bansal’s seminal SDP approach, The iterative partial-coloring method of Lovett & Meka, and Larsen's algorithm. The paper makes a significant contribution relative to these prior works.

**Theoretical Claims:**

I reviewed the outlines of the key proofs (notably for Theorems 1.1 and 1.3) and I think they are Kosher. However, I skipped several proofs (for example some of the parts in the long correctness proof of Algorithm 2).

---

> ### Author Rebuttal · Authors · 2025-03-31
>
> We sincerely thank the reviewer for the insightful comments and thorough evaluation of our paper.
> ### W1 and W2: Technical Density and Intuition
> We appreciate the reviewer's concern regarding the technical density of our presentation. Given the complex nature of discrepancy minimization and the depth of the theoretical contributions, our initial goal was to ensure rigorous correctness. However, we acknowledge that improving readability and intuition is crucial. In the final version, we will include additional intuitive explanations and clear roadmaps to guide readers through the intricate parts of our proofs, particularly emphasizing high-level insights before delving into technical details.
> ### W3: Experiments
> Thank you for pointing this out. We conduct the experiments to demonstrate the effectiveness of our algorithm. We focused our experimental evaluation on demonstrating the improvements achieved through the fast hereditary projection. This is because the lazy update scheme is primarily designed to leverage fast matrix multiplication. Although fast matrix multiplication theoretically has lower asymptotic complexity, algorithms based on FMM often suffer from significantly large constant factors, adversely affecting their practical runtime performance. Additionally, for practical considerations, the parameters used in the experiments do not strictly follow the theoretical suggestions in the paper.
> Our experimental setup follows the matrix configurations used in Larsen’s paper.
>
> **Uniform matrices:** Each entry of $A$ is uniformly chosen from $\\{-1,1\\}$
> | Matrix Size      | Sparsity | Larsen's Obj. Val. | Our Obj. Val. | Larsen's Runtime (s) | Our Runtime (s)|
> |------------------|----------|--------------------|---------------|------------------|-------------|
> | 400×400          | 1.0      | 54                 | 56            | 2.98             | 2.16        |
> | 400×400          | 0.5      | 38                 | 42            | 2.90             | 1.95        |
> | 400×400          | 0.1      | 14                 | 20            | 2.91             | 1.90        |
> | 2000×2000        | 1.0      | 140                | 148           | 345              | 164         |
> | 2000×2000        | 0.5      | 96                 | 99            | 334              | 156         |
> | 2000×2000        | 0.1      | 46                 | 47            | 331              | 152         |
> | 10000×1000       | 1.0      | 132                | 140           | 378              | 63          |
> | 10000×1000       | 0.5      | 92                 | 97            | 374              | 62          |
> | 10000×1000       | 0.1      | 42                 | 44            | 375              | 62          |
>
> **2D Corner matrices:** $A$ is constructed by choosing two sets of points in the unit square—one for rows and one for columns—and marking an entry as 1 if the row point strictly exceeds the column point in both coordinates, otherwise marking it as 0.
> | Matrix Size      | Sparsity | Larsen's Obj. Val. | Our Obj. Val. | Larsen's Runtime (s)| Our Runtime (s) |
> |------------------|----------|--------------------|---------------|------------------|-------------|
> | 400×400          | 1.0      | 30                 | 32            | 2.80             | 2.15        |
> | 400×400          | 0.5      | 36                 | 36            | 2.79             | 2.18        |
> | 400×400          | 0.1      | 17                 | 18            | 2.75             | 1.93        |
> | 2000×2000        | 1.0      | 52                 | 58            | 347              | 170         |
> | 2000×2000        | 0.5      | 83                 | 92            | 350              | 169         |
> | 2000×2000        | 0.1      | 45                | 46            | 350            | 171        |
> | 10000×1000       | 1.0      | 46                 | 52            | 386              | 65          |
> | 10000×1000       | 0.5      | 76                 | 77            | 374              | 60          |
> | 10000×1000       | 0.1      | 37                 | 40            | 375              | 62          |
>
> Our algorithm achieves substantial speedups with only minor sacrifices in approximation guarantees. The speedup will be more significant once $m$ is much largerer than $n$, as sketching is known to work well in the regime where $m \gg n$ .
>
> ### W4: Constants
> We agree with the reviewer that the constants in our current analysis primarily serve to highlight theoretical improvements. Optimizing these constants would indeed further enhance the practical applicability of our algorithm, although it would require additional effort. Nevertheless, our preliminary experiments indicate that the algorithm achieves significant improvements in runtime even without strictly adhering to the exact parameters suggested by the theoretical analysis.
>
> We appreciate the positive assessment of our work. Thank you again for your thoughtful review.

---

> > ### Comment · Reviewer_ZGyW · 2025-04-04
> >
> > Appreciate it, my concerns have been addressed, and I trust you’ll follow through on your commitment to:
> >
> > > include additional intuitive explanations and clear roadmaps to guide readers through the intricate parts of our proofs, particularly emphasizing high-level insights before delving into technical details

---

> > > ### Author Response · Authors · 2025-04-04
> > >
> > > We're glad to have addressed your concerns. Thank you for your valuable suggestions!

---

### Official Review · Reviewer_8Nfk · 2025-03-11

**Overall Recommendation:** 3

**Summary:**

The authors develop a new, faster algorithm for approximate discrepancy minimization with bounds on the computation time depending on the input-sparsity. Their algorithm is optimal for "tall" matrices, i.e. m x n matrices with m being a polynomial in n.
Additionally, the accuracy of the approximation matches a previous algorithm by Larsen.

They identify five "barriers", i.e. shortcomings, in Larsen's algorithm and they claim to improve on all of them. Some of the main ideas here are:
1) using sketching matrices and "ImplicitLeverageScore" to approximate some matrices efficiently.
2) Introducing a data structure that allows batching a series of computations.

Finally, they have a variant of their algorithm which uses "Fast Matrix Multiplication (FMM)"

**Claims And Evidence:**

The claims are supported by informal overviews of the various parts of the algorithm as well as proofs in the appendix. While I did not read the proofs in detail, the informal overview seemed convincing enough.

**Essential References Not Discussed:**

The discussion of the literature seems appropriate.

**Experimental Designs Or Analyses:**

Not relevant for this paper

**Methods And Evaluation Criteria:**

Their algorithm is compared with several existing algorithms and results in the discrepancy minimization litterature, which makes sense.

No experimental comparisons, does it actually compare in practise to existing algorithms? In particular, the previous algorithm of Larsen is accompanied by practical experiments, so it might have made sense to compare the new algorithm in practice. Of course, a purely theoretical improvement may be interesting in its own right.

**Other Comments Or Suggestions:**

List of typos and unclear phrases:

Typo p. 22: "proof of suceed probability" should be "proof of success probability"

Typo p. 15: "For convinient" should be "For convenience"


Typos p. 16, "It is well-known that random Gaussian matrices an AMS matrices gives JL-transform property" should instead be

"It is well-known that random Gaussian matrices and AMS matrices give JL-transform property".

Typo p. 20, "we now proof" should be "we now prove"
On p. 24, it says "expectated time". Expectated is not a word, should probably be "expected time".
On p. 24, there should probably be a "such that" or something similar before the inequality in the statement of Lem. E.1, right after "eta \in R".

Typo p. 26:  It says "have have" at the bottom of the page.
On page 26, it says "the number entries reach the absolute value 1 keeps increasing" which makes very little sense to the point where I'm not sure what the sentence is trying to convey.

Typo page 27: The last "We" should not be capitalized in Corollary E.3.
Typo pages 27 and 28: "fourth" should be "fourth".
Typo in definition E.4. Should be "which is implicitly maintained..." instead of "which implicitly maintained..."

On page 30, should it not be g_t in "we first assume that, we add mu \cdot g"?

On page 30, on the bottom of the page it says "there must be at least one entry i \in [n] satisfying that ..." the two equations that come after this are identical which surely must be a mistake? Additionally, there is a | too much at the beginning of the second equation here here.


On page 31 at the top, what does it mean that a set is larger than a number?

In Lem. E.2, and Cor. E.3 it says (0,1)^n - should this not be (-1,1)^n?

In definition E.5, what does it mean that a vector g_t is sampled from a univariate Gaussian N(0,1)? Are the coordinates i.i.d from this distribution?

**Other Strengths And Weaknesses:**

The results themselves seem fine. It might not be the most original work, since it seems to mostly consist of several existing algorithms stitched together. However, in terms of technical depth, it is well beyond most NeurIPS papers.

I'm unsure whether the paper fits the scope of the conference. There may be applications to machine learning, but the paper itself seems to have very little to do with machine learning and would probably fit better at a TCS venue. The authors also do not give convincing arguments that the paper belongs in a machine learning conference.

Finally, there are simply too many typos and weird, unclear phrases that don't make sense, especially in the appendices. This is very distracting and makes for a tedious reading experience. See "Other Comments or Suggestions" for some examples.

**Questions For Authors:**

- Do you think your algorithm would be competitive in practice with the previous algorithm by Larsen?

- Can you give some further examples of why discrepancy minimization is interesting for the ML community?

**Relation To Broader Scientific Literature:**

The paper provides a fairly thorough overview of existing theoretical results and previous algorithms in discrepancy minimization. Additionally, several applications of discrepancy minimzation are mentioned, although these are kept pretty vague.

The most important references are the algorithms by Larsen (2023) and Eldahn & Singh (2018).

**Theoretical Claims:**

I did not read any proofs in full as they all appear in the appendix. However, I skimmed part of the proof of Theorem C.3 (page 19) in which it is claimed that a particular set of eigenvectors are orthogonal. This is not true in general, and it was unclear to me why it should be true in their proof. It might still be true, but at least I think it requires an argument.

---

> ### Author Rebuttal · Authors · 2025-03-31
>
> We thank the reviewer for the valuable feedback.
>
> ### Theoretical Claims: The proof of Theorem C.3 (page 19) claimed that a particular set of eigenvectors are orthogonal, which is not true in general.
>
> Because any real symmetric matrix admits an orthogonal diagonalization, its eigenvectors can always be taken to be orthonormal.
>
> ### W1: Originality and connection to existing methods
>
> While our work builds upon Larsen's algorithm and Lovett–Meka random walk method, we would like to highlight the significant theoretical and algorithmic advances introduced in our work:
> -  We introduce an innovative implicit leverage-score sampling technique, enabling us to significantly reduce the computational complexity of hereditary projection from Larsen’s original $O(mn^2)$ (or $O(mn^\omega)$ with fast matrix multiplication) to $O(nnz(A)+n^{\omega})$. This improvement is crucial, particularly for sparse input matrices.
> - We provide a robust theoretical analysis that carefully manages error accumulation introduced by randomized sketching methods. Such robustness was not previously established and represents a substantial methodological advancement over existing discrepancy minimization frameworks.
> - The lazy-update mechanism we introduce to efficiently implement the iterative Edge-Walk algorithm fundamentally reduces computational complexity from Larsen’s $O(mn^2+n^3)$ to our result of $\tilde{O}(nnz(A) + n^{2.53})$. This addresses and overcomes a significant computational barrier identified in prior literature.
>
> ### W2 and Q2: Interest to the ICML community and applications in ML
> Thanks for pointing this out. We argue that the broad relevance of discrepancy theory to the ICML community stems from its strong connections to central themes in machine learning. Specifically, discrepancy minimization intersects significantly with computational learning theory [1,2], computer vision [3], unsupervised learning [4], attention KV caching [5], differential privacy [6], and sampling [7,8].
> More specifically,
> [1,2] introduces the relationship of discrepancy and concepts in learning theory such as VC dimension, PAC learning.
> [3] introduces a method to learn hash functions via discrepancy minimization. Learning hash functions in content-based image retrieval optimize binary codes to preserve similarity in high-dimensional feature spaces, enabling faster and more efficient image search.
> [4] leverage discrepancy minimization for unsupervised graph matching by aligning predictions from classical solvers and neural models.
> [5] proposed an algorithm for compressing the KV cache recursively using a geometric correlated sampling process based on discrepancy theory
> [6] investigated the relationship between discrepancy minimization and differential privacy in the context of linear queries over histograms.
> Quasi-Monte Carlo methods [7,8] leverage concepts from discrepancy theory by employing low-discrepancy sequences to efficiently approximate high-dimensional integrals and expectations.
>
> In the final manuscript, we will explicitly clarify and elaborate on these connections to further emphasize the relevance and significance of our work to the ICML audience.
>
> [1] Matousek, Jiri. Geometric discrepancy: An illustrated guide. 2009
>
> [3] Karnin, Zohar, and Edo Liberty. Discrepancy, coresets, and sketches in machine learning. COLT’19
>
> [3] Chen, Zhixiang, et al. Deep hashing via discrepancy minimization. CVPR’18
>
> [4] Wang, Runzhong, Junchi Yan, and Xiaokang Yang. Unsupervised learning of graph matching with mixture of modes via discrepancy minimization. TPAMI’23
>
> [5] Han, Insu, et al. BalanceKV: KV Cache Compression through Discrepancy Theory. arXiv preprint:2502.07861
>
> [6] Nikolov, Aleksandar, Kunal Talwar, and Li Zhang. The geometry of differential privacy: the sparse and approximate cases. STOC’13
>
> [7] Lyu, Yueming, Yuan Yuan, and Ivor Tsang. Subgroup-based rank-1 lattice quasi-monte carlo. NeurIPS’20
>
> [8] Lyu, Yueming. Fast rank-1 lattice targeted sampling for black-box optimization. NeurIPS’23
>
> ### W3: Typos
> We sincerely appreciate the detailed list of typos and suggestions provided by the reviewer. We will carefully proofread the manuscript, correcting all listed typographical errors and ensuring clarity in terminology and statements. We recognize that readability and precision are crucial, particularly in technical papers, and will make substantial efforts to improve the overall presentation quality.
> ### Q1: Competitiveness relative to Larsen’s algorithm
> Thank you for pointing this out. We conduct the experiments to demonstrate the effectiveness of our algorithm. Due to space limit, please refer to the rebuttal to the reviewer ZGyW for the experiment results.
>
> We sincerely appreciate your valuable feedback, which will significantly help us improve our paper. We hope that we have adequately addressed your concerns. Please do not hesitate to reach out if you have any further questions or comments. Thank you once again for your thoughtful review.

---

### Official Review · Reviewer_N7kM · 2025-03-13

**Overall Recommendation:** 4

**Summary:**

The paper is on discrepancy minimization for real matrices - goal is to develop constructive methods that exploit input sparsity.  Algorithmic discrepancy is a well-studied topic in TCS and there have been many breakthroughs in the last 15 years, starting with Bansal.  Recently there was a result for binary matrices based on sparsity.  This paper obtains an analogous result for real-valued matrices.

The main technique is basically dig into Larsen's algorithm, identify all the barriers to improve Larsen's algorithm for sparse real matrices.  Then, the idea is to bring in techniques from randomized linear algebra and sketching to save computation.  It is not straightforward since there are several steps involved and naive ways could lead to error accumulation.  Besides carefully using these tools, tightening/modifying the analysis of Larsen, the paper also introduces new techniques including an implicit leverage score sampler and efficiently implementing certain random-walk based rounding step.

**Claims And Evidence:**

Yes - don't see any problems.

**Essential References Not Discussed:**

Looks adequate.

**Experimental Designs Or Analyses:**

Theory work.

**Methods And Evaluation Criteria:**

Theory work.

**Other Comments Or Suggestions:**

Might be worth adding nnz(A)+n^3 combinatorial bound to Table 1

page 2 - notation V_{l,*} undefined

Could you adopt a single notation to capture both \cal{T}_mat(m,n,k) and \omega(a,b,c)?

page 5 line 223 and on & 234 and on - most of the content seems repeated

**Other Strengths And Weaknesses:**

+ves

+ Makes good progress on an important combinatorics / algorithmic question
+ The techniques are non-trivial and this area is usually challenging
+ There are some novel components such as the lazy updates and implicit leverage score sampling.  It is conceivable these might have other applications
+ Brings matrix sketching/randomized linear algebra techniques to the algorithmic discrepancy space

-ves
- Heavily built on Larsen's algorithm / Lovett--Meka random walk and other existing tools from the literature
- The problem might be of narrow interest, even to the ICML audience

**Questions For Authors:**

None

**Relation To Broader Scientific Literature:**

Improvement to the running time of real-valued matrix discrepancy.

**Theoretical Claims:**

Yes - don't see major problems.

---

> ### Author Rebuttal · Authors · 2025-03-31
>
> We thank the reviewer for the valuable feedback and for recognizing the novelty, significance, and technical contributions of our work.
>
> ### W1: Connection to existing works
> While our work indeed builds upon Larsen's algorithm and Lovett–Meka random walk method, we would like to highlight the significant theoretical and algorithmic advances in our work:
> -  We introduce an innovative implicit leverage-score sampling technique, enabling us to significantly reduce the computational complexity of hereditary projection from Larsen’s original $O(mn^2)$ (or $O(mn^\omega)$ with fast matrix multiplication) to $O(nnz(A)+n^{\omega})$. This improvement is crucial, particularly for sparse input matrices.
> - We provide a robust theoretical analysis that carefully manages error accumulation introduced by randomized sketching methods. Such robustness was not previously established and represents a substantial methodological advancement over existing discrepancy minimization frameworks.
> - The lazy-update mechanism we introduce to efficiently implement the iterative Edge-Walk algorithm fundamentally reduces computational complexity from Larsen’s $O(mn^2+n^3)$ to our result of $\tilde{O}(nnz(A) + n^{2.53})$. This addresses and overcomes a significant computational barrier identified in prior literature.
>
> ### W2: Interest to ICML community
> Thanks for pointing this out. We argue that the broad relevance of discrepancy theory to the ICML community stems from its strong connections to central themes in machine learning. Specifically, discrepancy minimization intersects significantly with computational learning theory [1,2], computer vision [3], unsupervised learning [4], attention KV caching [5], differential privacy [6], and sampling [7,8].
>
> More specifically,
> [1,2] introduces the relationship of discrepancy and concepts in learning theory such as VC dimension, PAC learning.
> [3] introduces a method to learn hash functions via discrepancy minimization. Learning hash functions in content-based image retrieval optimize binary codes to preserve similarity in high-dimensional feature spaces, enabling faster and more efficient image search.
> [4] leverage discrepancy minimization for unsupervised graph matching by aligning predictions from classical solvers and neural models.
> [5] proposed an algorithm for compressing the KV cache recursively using a geometric correlated sampling process based on discrepancy theory
> [6] investigated the relationship between discrepancy minimization and differential privacy in the context of linear queries over histograms.
> Quasi-Monte Carlo methods [7,8] leverage concepts from discrepancy theory by employing low-discrepancy sequences to efficiently approximate high-dimensional integrals and expectations.
> In the final manuscript, we will explicitly clarify and elaborate on these connections to further emphasize the relevance and significance of our work to the ICML audience.
>
> [1] Matousek, Jiri. Geometric discrepancy: An illustrated guide. 2009
>
> [2] Karnin, Zohar, and Edo Liberty. Discrepancy, coresets, and sketches in machine learning. COLT’19
>
> [3] Chen, Zhixiang, et al. Deep hashing via discrepancy minimization. CVPR’18
>
> [4] Wang, Runzhong, Junchi Yan, and Xiaokang Yang. Unsupervised learning of graph matching with mixture of modes via discrepancy minimization. TPAMI’23
>
> [5] Han, Insu, et al. BalanceKV: KV Cache Compression through Discrepancy Theory. arXiv preprint:2502.07861
>
> [6] Nikolov, Aleksandar, Kunal Talwar, and Li Zhang. The geometry of differential privacy: the sparse and approximate cases. STOC’13.
>
> [7] Lyu, Yueming, Yuan Yuan, and Ivor Tsang. Subgroup-based rank-1 lattice quasi-monte carlo. NeurIPS’20
>
> [8] Lyu, Yueming. Fast rank-1 lattice targeted sampling for black-box optimization. NeurIPS’23
>
> ### C1: Adding the combinatorial bound to Table 1
> Thank you for your nice suggestion. We will add this to Table 1.
>
> ### C2: notation $V_{l,*}$ undefined
> Thanks for pointing this out. This means the $l$-th row of the matrix $V$. We will define this in the final manuscript.
>
> ### C3: Could you adopt a single notation to capture both \cal{T}_mat(m,n,k) and \omega(a,b,c)?
> Thank you for your insightful question. We use two notations intentionally due to their different emphases and contexts:  We use $\mathcal{T}_\mathrm{mat}(m,n,k)$ to explicitly highlight absolute matrix dimensions, making it convenient for describing the algorithm's complexity at a high level; meanwhile, $\omega(a,b,c)$ concisely captures relative dimension ratios and is widely adopted in algebraic complexity theory, facilitating rigorous complexity analysis. We will clarify this distinction in our revision to ensure readers fully understand why both notations are necessary and complementary.
>
> ### C4: page 5 line 223 and on & 234 and on - most of the content seems repeated
> Thanks for pointing this out. We will remove the repeated parts.
>
> We greatly appreciate the positive evaluation of our paper and thank you for your constructive review.

---

### Official Review · Reviewer_8esX · 2025-03-14

**Overall Recommendation:** 4

**Summary:**

This paper proposes an improved randomized algorithm for discrepancy minimization problem for real-valued matrices m*nmatrices A with m = poly(n). The paper builds on top of work of Larsen and proposes an improvements to Larsen's algorithm that allow authors to achieve a combinatorial algorithm that runs in input-sparsity time $\widetilde{O}(nnz(A)+n^3)$  with the same approximation guarantee as Larsen's algorithm. The authors also demonstrate how using Fast matrix multiplication one can decrease the runtime to $\widetilde{O}(nnz(A)+n^2.53)$. Prior to this work, the best runtime guarantee in a similar setup was O(mn^2).

The paper introduces novel ideas for each subroutine of the Larsen's algorithm using sketching and score sampling. The authors use sketching techniques to speed up ProjectToSmallRows subroutine of Larsen's algorithm. To overcome $\Omega(n^3)$ barrier for the iterative Partial coloring subroutine authors propose a clever way to perform updates in batches with a new "lookahead" data structure.

**Claims And Evidence:**

The authors provide detailed well-written proofs for the claims made in the paper. They present conceptual arguments and brief summary of the key steps and new ideas compared to the Larsen algorithm in the main text, while detailed definitions and proofs, justifying the claims are presented in the appendix.

**Essential References Not Discussed:**

In my opinion, the paper provides a very nice overview of the related literature and gives a detailed comparison to the related work.

**Experimental Designs Or Analyses:**

N/A

**Methods And Evaluation Criteria:**

No experiments are conducted

**Other Comments Or Suggestions:**

-

**Other Strengths And Weaknesses:**

The problem solved in this paper is a well-known problem with multiple applications and a randomized theoretical algorithm with a sub-cubic runtime is definitely of great interest. The paper introduces several new ideas and provides a nice overview of key novelties compared to Larsen's work it builds upon. In my opinion the results are well-presented and easy to read.

**Questions For Authors:**

-

**Relation To Broader Scientific Literature:**

The discrepancy minimization problem has been extensively studied in the literature from both existence (to understand minimal possible discrepancy) and algorithmic perspective. Prior to this work best runtime for the real-valued matrices was by Larsen (2023) running in O(mn^2). This paper focuses on an important practical case of sparse matrices and achieves the first algorithm that works O(nnz(A) +n^3) time providing a significant improvement for tall sparse matrices.  This problem was earlier solved by Jain, Sah and Sawhney 2023 for binary matrices, however, their proof does not seem to translate to real-valued matrices easily.

**Theoretical Claims:**

I skimmed through the proofs and they seem to be sound, but I did not verify the details

---

> ### Author Rebuttal · Authors · 2025-03-31
>
> We greatly appreciate the reviewer's recognition of our contributions, specifically, including significant runtime improvements, novel algorithmic techniques, clear comparisons to existing literature, and rigorous theoretical justifications. We believe our contributions represent a significant step forward in improving computational efficiency and advancing algorithmic techniques for discrepancy minimization. Please let us know if you have any further comments regarding our work. Thank you again for your positive feedback.

---

### Decision · Program_Chairs · 2025-05-01

**Decision:**

Accept (spotlight poster)

**Comment:**

The (NP-hard) discrepancy of an m x n matrix A is the minimum, over all discrete x in {-1, 1}^n, of the infinity norm of Ax. This is a fundamental quantity in discrete mathematics, algorithms, and in ML. This work presents a combinatorial---not SDP or LP-based---O(nnz(A) + n^3)-time approximation algorithm for the discrepancy with the same approximation guarantee as recent work of Kasper Green Larsen's, optimal for “tall” matrices where m = poly(n). (As usual, "nnz" denotes the number of nonzeroes.) Using further analysis and modern fast matrix multiplication, the paper improves the runtime to O(nnz(A) + n^{2.53}). The paper identifies five areas of improvement in Larsen's algorithm and claim to improve on all of them. Some of the main ideas here are using sketching matrices and a data structure that allows batching a series of computations. This paper makes good progress on a fundamental algorithmic problem relevant to ML.